# SFedPO: Streaming Federated Learning with a Prediction Oracle under Temporal Shifts

## Abstract

Federated Learning (FL) enables decentralized clients to collaboratively train a global model without sharing raw data. However, most existing FL frameworks assume that clients train on static local datasets collected in advance or that the data follows a fixed underlying distribution, which limits their applicability in dynamic environments where data evolves over time. A parallel line of research, online FL, removes all assumptions and adopts an adversarial perspective, but this approach is often overly pessimistic and neglects the structured, partially predictable nature of real-world data dynamics. To bridge this gap, we propose SFedPO, a streaming federated learning framework that incorporates a prediction oracle to capture the temporal evolution of client-side data distributions. We theoretically analyze the convergence bounds of SFedPO and develop two practical strategies: a Distribution-guided Data Sampling (DDS) strategy that dynamically selects training data under limited storage by balancing historical reuse and distribution adaptation, and a Shift-aware Aggregation Weights (SAW) mechanism that modulates global aggregation based on client-specific sampling behaviors. We further establish robustness guarantees under prediction errors. Extensive experiments demonstrate that SFedPO effectively adapts to streaming scenarios with distribution shifts and significantly outperforms existing methods.

## 1 Introduction

Federated Learning (FL) (McMahan et al., 2017) has emerged as a prominent distributed learning paradigm that enables multiple clients to collaboratively train a shared global model while keeping data local. By leveraging local computation and periodically exchanging model parameters via a central server, FL eliminates the need for centralized data collection. This makes it particularly well-suited for real-world scenarios where data is inherently distributed, such as mobile sensing (Jiang et al., 2020) and edge intelligence (Mills et al., 2019).

Most existing FL frameworks assume that clients train on static local datasets collected in advance or that their data follows a fixed underlying distribution (McMahan et al., 2017; Li et al., 2020b; Wang et al., 2020b; Wang & Ji, 2022; Ye et al., 2023). In contrast, online FL (Mitra et al., 2021; M Ghari & Shen, 2022) discards such assumptions and instead adopts a potentially adversarial modeling perspective. It aims to minimize cumulative regret (Kwon et al., 2023; Patel et al., 2023) and provides robust theoretical guarantees under worst-case scenarios. While static local datasets or a fixed data distribution could be a strong assumption, the adversarial perspective represents the opposite extreme and is overly pessimistic.

Consider a streaming FL setting where clients (e.g., mobile devices or UAVs) continuously acquire new data over time. On one hand, this results in time-varying and non-stationary data distributions. On the other hand, the evolution of clients' data often follows structured and partially predictable patterns (Huynh et al., 2025). For instance, in the case of UAVs, their mobility trajectories typically follow pre-defined routes or scheduled missions. As a result, the data they collect exhibits spatial and temporal regularities. While assuming static datasets or fixed distributions could be inappropriate, the adversarial perspective may be overly pessimistic. This raises a natural question: Can we design a new FL framework that leverages partial predictions about distribution shifts to jointly optimize its two core components—client-side data sampling and server-side aggregation?

Addressing this question introduces significant challenges in both theoretical analysis and system design. Theoretically, the time-varying nature of clients' data renders the local objective functions time-dependent, complicating convergence analysis and requiring new tools to characterize the evolving optimization landscape. From a system design perspective, the key challenge lies in accommodating heterogeneous client behaviors, particularly their sampling strategies in response to distribution shifts.

Motivated by the discussions above, we propose SFedPO, a streaming federated learning framework with a prediction oracle to address temporal distribution shifts. Rather than assuming static datasets or adversarial dynamics, SFedPO operates in environments where clients continuously collect new data and incorporates a prediction oracle that provides prior knowledge about the temporal evolution of clients' data distributions. Building upon this, we introduce a distribution-guided data sampling strategy that selectively reuses and updates local data in response to distribution shifts. This allows clients to maintain a representative memory buffer under limited storage constraints. Furthermore, we develop an aggregation algorithm that adapts to heterogeneous client behaviors, including their sampling strategies. Our main contributions are summarized as follows:

- We propose SFedPO, a novel streaming FL framework incorporating a prediction oracle to model the temporal evolution of clients' data distributions. This framework provides a principled approach that bridges the gap between static-data assumptions and adversarial modeling commonly found in existing FL paradigms.
- We provide a theoretical convergence analysis of SFedPO and develop two core components: Distribution-guided Data Sampling (DDS) for local training, and Shift-aware Aggregation Weights (SAW) for client-adaptive global aggregation. In addition, we conduct a robustness analysis that quantifies the impact of oracle prediction errors on convergence guarantees.
- We perform extensive experiments, demonstrating that SFedPO effectively adapts to streaming scenarios with distribution shifts and significantly outperforms existing methods.

## 2 RELATED WORKS

We categorize existing FL literature into three main lines based on their assumptions about the data.

**Federated Learning with Static or Stationary Data.** Most traditional FL methods assume that clients train on static datasets or that their data follows a fixed distribution (McMahan et al., 2017; Li et al., 2020b; Wang et al., 2020b; Wang & Ji, 2022; Ye et al., 2023). Some recent studies extend FL to streaming scenarios. For instance, Marfoq et al. (2023) formalize FL over data streams and propose a meta-algorithm similar to vanilla FedAvg (McMahan et al., 2017) through a weighted empirical risk minimization design. ODE (Gong et al., 2023) introduces a date evaluation metric based on inference accuracy for on-device data selection under storage constraints. However, these works still assume that the data follows a stationary distribution.

**Federated Learning under Distribution Shifts.** Some studies have extended FL to streaming settings where data continuously arrive over time (Huynh et al., 2025; Marfoq et al., 2023; Gong et al., 2023; Liu et al., 2023). Wang et al. (2023a) assume the existence of a long-term distribution in the local data stream of clients and propose the cache update strategy to align the data distribution in the local cache to the underlying long-term distribution. Fed-HIST (Zhang et al., 2024) avoids the problem of raw data storage by retrieving model-based historical representations through similarity comparison. In parallel, the concept drift problem has been explored in FL. Most concept drift adaptation methods typically modify the model architecture (Chen et al., 2021), optimization strategy (Panchal et al., 2023; Canonaco et al., 2021), or client clustering (Li et al., 2024; Jothimurugesan et al., 2023; Chen et al., 2024) in response to detected shifts. Furthermore, Federated Continual Learning (FCL) (Yang et al., 2024; Guo et al., 2021; Dong et al., 2022) has been developed to mitigate catastrophic forgetting under sequential task arrivals, typically by leveraging parameter decomposition (Yoon et al., 2021), generative replay (Wuerkaixi et al., 2024; Qi et al., 2023), or knowledge distillation (Huang et al., 2022; Usmanova et al., 2021). However, existing approaches across these lines do not explicitly model or predict the temporal evolution of data distributions. In contrast, our work leverages a predictive oracle and develops a theoretically grounded, distribution-guided data sampling strategy for streaming FL. See details in Appendix B.

**Online Federated Learning.** Online FL removes assumptions on the underlying data distribution and instead aims to minimize regret under potentially arbitrary or adversarial data streams. FedOMD (Mitra et al., 2021) studies online federated optimization against adversarially revealed loss functions using online mirror descent, achieving sublinear regret. Ganguly & Aggarwal (2023) further combine FedAvg and FedOMD within a multiscale framework to adapt to non-stationary environments, establishing dynamic regret bounds under general convex losses. To address system-level challenges such as device heterogeneity and availability variations, ASO-Fed (Chen et al., 2020) introduces an asynchronous online FL framework based on continuous local updates and asynchronous aggregation.

## 3 PROBLEM FORMULATION

### 3.1 MODELING DYNAMIC DATA DISTRIBUTIONS

We consider a federated learning system consisting of $N$ clients, denoted by the set $\mathcal{N} = \{1, 2, \ldots, N\}$, and these clients are coordinated by a central server. The learning process unfolds in $R$ communication rounds, indexed by $r \in \{1, \ldots, R\}$. Each round is further divided into $T$ fine-grained time steps, indexed by $t \in \{1, \ldots, T\}$. We define a time step as the granularity at which one client's data distribution may evolve.

Instead of assuming a static dataset or a stationary distribution, clients in our system continuously receive new data generated from a dynamic distribution. We model the distributions of new data as governed by a latent state space $\mathcal{M} = \{1, 2, \ldots, M\}$. Specifically, each state $m \in \mathcal{M}$ corresponds to a stationary data distribution $\mathcal{D}_m$. At each time step $t$ in round $r$, client $n \in \mathcal{N}$ is associated with a latent state $m_{n,t}^{(r)} \in \mathcal{M}$. The ground-truth state distribution of client $n$ is denoted as $\boldsymbol{\pi}_n = (\pi_{n,1}, \ldots, \pi_{n,M})$, where $\pi_{n,m}$ denotes the probability that client $n$ is in state $m$. While the true state transition dynamics are unknown, we assume the existence of a **prediction oracle** that provides a prediction over clients' latent states. That is, the oracle for client $n$ outputs a prediction vector $\hat{\boldsymbol{\pi}}_n = (\hat{\pi}_{n,1}, \ldots, \hat{\pi}_{n,M})$, where $\hat{\pi}_{n,m}$ denotes the estimate of $\pi_{n,m}$.

In streaming settings with limited local storage, clients must manage their training data continuously by retaining only a subset (possibly empty) of newly encountered samples. Considering this issue, we consider a distribution-guided data sampling mechanism, which explicitly adjusts the composition of each client's local dataset based on the observed data distribution. Specifically, we define a sampling strategy $\boldsymbol{\alpha} = \{\alpha_{n,m}\}_{n \in \mathcal{N}, m \in \mathcal{M}}$, where $\alpha_{n,m} \in [0, 1]$ represents the ratio of new samples from state $m$ that client $n$ incorporates into its local dataset. Let $\mathcal{D}_{n,t}^{(r)}$ denote the effective training distribution maintained by client $n$ at time $t$ in round $r$, which is updated as a convex combination of the previous local distribution and the current state's distribution, i.e.,

$$\mathcal{D}_{n,t}^{(r)} = (1 - \alpha_{n,m_{n,t}^{(r)}}) \cdot \mathcal{D}_{n,t-1}^{(r)} + \alpha_{n,m_{n,t}^{(r)}} \cdot \mathcal{D}_{m_{n,t}^{(r)}}. \tag{1}$$

This update captures the trade-off between preserving historical samples and adapting to recent distribution shifts. Given the local training distribution $\mathcal{D}_{n,t}^{(r)}$, we define the local loss function of client $n$ at time step $t$ in round $r$ as

$$F_{n,t}^{(r)}(\mathbf{x}) \triangleq \mathbb{E}_{\xi \sim \mathcal{D}_{n,t}^{(r)}}[f(\mathbf{x}; \xi)], \tag{2}$$

where $f(\mathbf{x}; \xi)$ denotes the sample-wise loss function.

### 3.2 FEDERATED TRAINING PROCESS

Our learning objective is to train a global model $\mathbf{x}^* \in \mathbb{R}^d$ that generalizes well across the full space of data distributions encountered by all clients over time. Specifically, we define the global objective as a weighted sum over all $M$ possible distributions:

$$\mathbf{x}^* = \arg\min_{\mathbf{x}} F(\mathbf{x}) = \sum_{m=1}^{M} w_m \cdot F_m(\mathbf{x}), \tag{3}$$

where $F_m(\mathbf{x}) = \mathbb{E}_{\xi \sim \mathcal{D}_m}[f(\mathbf{x}; \xi)]$ denotes the expected loss under distribution $\mathcal{D}_m$, and $w_m$ reflects the population-level importance or frequency of state $m$ across all clients and time steps.

The training proceeds over $R$ communication rounds. In each round $r$, a subset of clients $S_r$ is sampled independently with availability probabilities $q_n$ (Wang & Ji, 2022; Xiang et al., 2024), and each selected client initializes its local model with the current global model $\bar{\mathbf{x}}^{(r)}$. As defined earlier, each round is divided into $T$ time steps, we align local model updates with these time steps: at each time step $t$ in round $r$, client $n \in S_r$ samples a mini-batch of data from its updated local distribution $\mathcal{D}_{n,t}^{(r)}$ and performs one step of Stochastic Gradient Descent (SGD) as follows:

$$\mathbf{x}_{n,t}^{(r)} = \mathbf{x}_{n,t-1}^{(r)} - \eta \cdot \mathbf{g}_{n,t}^{(r)}, \tag{4}$$

where $\eta$ is the learning rate, and $\mathbf{g}_{n,t}^{(r)} \triangleq \nabla f(\mathbf{x}_{n,t-1}^{(r)}; \xi_{n,t})$ is the stochastic gradient of $F_{n,t}^{(r)}(\mathbf{x}_{n,t-1}^{(r)})$ evaluated on samples $\xi_{n,t} \sim \mathcal{D}_{n,t}^{(r)}$. After completing the local updates, each client uploads its final model $\mathbf{x}_{n,T}^{(r)}$ to the server. The server then aggregates the received models using a weighted average:

$$\bar{\mathbf{x}}^{(r+1)} = \sum_{n \in S_r} p_n \cdot \mathbf{x}_{n,E}^{(r)}, \tag{5}$$

where $p_n$ is the aggregation weight for the client $n$.

To support effective learning under temporal distribution shifts, our framework incorporates two core components: (1) a distribution-guided data sampling strategy that dynamically adjusts local datasets, and (2) an aggregation algorithm that accounts for the heterogeneous client behaviors, including sampling strategies and available probabilities.

## 4 THEORETICAL ANALYSIS

In this section, we analyze the convergence bound of the proposed federated learning framework under temporal distribution shifts, which highlights the effect of sampling strategy $\boldsymbol{\alpha}$ and aggregation weights $\{p_n\}_{n \in \mathcal{N}}$. We begin by stating several assumptions on the local loss functions $F_{n,t}^{(r)}$ and the state-specific loss function $F_m$.

**Assumption 1.** *(L-smoothness). All loss functions involved in the optimization are L-smooth. That is, there exists a constant $L > 0$ such that*

$$\|\nabla F_{n,t}^{(r)}(\mathbf{x}) - \nabla F_{n,t}^{(r)}(\mathbf{y})\| \le L\|\mathbf{x} - \mathbf{y}\| \text{ and } \|\nabla F_m(\mathbf{x}) - \nabla F_m(\mathbf{y})\| \le L\|\mathbf{x} - \mathbf{y}\|,$$

*for all $\mathbf{x}, \mathbf{y} \in \mathbb{R}^d$, $n \in \mathcal{N}$, $t \in \{1, \dots, T\}$, $r \in \{1, \dots, R\}$, and $m \in \mathcal{M}$.*

**Assumption 2.** *(Unbiased gradient and bounded variance). For each client $n$, the stochastic gradient is an unbiased estimator of the full gradient and has bounded variance. Formally,*

$$\mathbb{E}_{\xi \sim \mathcal{D}_{n,t}^{(r)}}[\nabla f(\mathbf{x}; \xi)] = \nabla F_{n,t}^{(r)}(\mathbf{x}), \ \mathbb{E}_{\xi \sim \mathcal{D}_{n,t}^{(r)}}[\|\nabla f(\mathbf{x}; \xi) - \nabla F_{n,t}^{(r)}(\mathbf{x})\|^2 | \mathbf{x}] \le \sigma^2,$$

*where $\xi$ is sampled from the $n$-th client's local data distribution $\mathcal{D}_{n,t}^{(r)}$ uniformly at random.*

To characterize the heterogeneity of local objectives, many prior works adopt standard dissimilarity assumptions (Wang & Ji, 2022; Wang et al., 2020b), which bound the deviation between the local and global gradients:

$$\frac{1}{M} \sum_{m=1}^{M} \|\nabla F_m(\mathbf{x}) - \nabla F(\mathbf{x})\|^2 \le \beta^2 \|\nabla F(\mathbf{x})\|^2 + \zeta^2.$$

While this formulation captures the average heterogeneity across all latent states, it lacks granularity for state-specific characterization. Therefore, we decompose it into two assumptions: one that bounds the overall gradient heterogeneity across all latent states and another that controls the state-specific gradient heterogeneity. These can be viewed as variants of the above dissimilarity assumptions.

**Assumption 3.** *There exists a constant $G > 0$ such that $\sum_{m=1}^{M} \|\nabla F_m(\mathbf{x})\|^2 \le G$, for all $\mathbf{x} \in \mathbb{R}^d$.*

**Assumption 4.** *For each latent state $m \in \mathcal{M}$, there exists a constant $d_m > 0$ such that $\|\nabla F_m(\mathbf{x}) - \nabla F(\mathbf{x})\|^2 \le d_m$, for all $\mathbf{x} \in \mathbb{R}^d$.*

We now present the main convergence result under the assumptions stated above. The theorem provides an upper bound on the optimization error, taking into account the effects of data dynamics, sampling strategies, and aggregation weights.

Figure 1: An overview of the proposed SFedPO framework. On the client side, each client determines its DDS strategy based on a prediction oracle $\hat{\boldsymbol{\pi}}_n$ and state-wise heterogeneity bounds $\{d_m\}_{m=1}^M$. This produces sampling ratios $\hat{\alpha}_{n,m}$ for constructing the local dataset and updating the local model. On the server side, global aggregation is performed with the well-designed SAW, which adaptively adjusts the aggregation weights $\hat{p}_n$ according to each client's heterogeneity score $\hat{s}_n$.

**Theorem 1.** *Let Assumptions 1 – 4 hold. Suppose the data sampling strategy $\boldsymbol{\alpha}$ and the aggregation weights $\{p_n\}_{n \in \mathcal{N}}$ are fixed. If the learning rate satisfies $LT\eta \leq \min\{\frac{1}{2M}, \sqrt{\frac{1}{5}}\}$. Then, the optimization error will be bounded as follows:*

$$\min_r \mathbb{E}\|\nabla F(\bar{\mathbf{x}}^{(r)})\|^2 \leq \frac{18}{\sum_{n=1}^N p_n q_n}\Big[\frac{1}{T\eta R}F(\bar{\mathbf{x}}^{(1)}) + L\eta\sigma^2 \sum_{n=1}^N p_n^2 q_n + \frac{5L\eta\sigma^2}{3}\sum_{n=1}^N p_n q_n + \tag{6}$$

$$\frac{5}{3}\sum_{n=1}^N p_n q_n \Big(\frac{\beta_n}{\alpha_n}(1-\gamma_n) + \frac{2G\gamma_n\beta'_n}{1-(1-\alpha_n)^2} + 2G\gamma_n \sum_{m=1}^M (\frac{\pi_{n,m}\alpha_{n,m}}{\alpha_n}-w_m)^2\Big) + \frac{1}{R}\sum_{n=1}^N \frac{7G\gamma_n q_n}{(1-(1-\alpha_n)^{2T})}\Big],$$

*where*

$$\alpha_n = \sum_{m=1}^M \pi_{n,m}\alpha_{n,m}, \quad \beta_n = \sum_{m=1}^M \pi_{n,m}\alpha_{n,m}d_m, \quad \gamma_n = \frac{1}{T}\sum_{t=1}^T (1-\alpha_n)^t, \tag{7}$$

$$\beta'_n = 2\sum_{m=1}^M \pi_{n,m}\alpha_{n,m}^2 - \sum_{m=1}^M \pi_{n,m}^2 \alpha_{n,m}^2 + \alpha_n^2. \tag{8}$$

*Proof.* See details in Appendix E. $\square$

## 5 THE SFEDPO FRAMEWORK

Based on the convergence analysis, we develop two coordinated modules that jointly minimize the upper bound of the optimization error in streaming federated environments.

**Distribution-guided Data Sampling (DDS).** To ensure effective local training under temporally evolving data, DDS leverages client-specific latent state distributions to guide sample selection. As demonstrated in our theoretical analysis, the convergence bound is strongly influenced by the data sampling strategy, particularly through quantities such as $\alpha_n$, $\beta_n$, and $\beta'_n$. We treat $\alpha_n$ as a fixed hyperparameter that governs the average update ratio of the client buffer, referred to as the *sampling budget*. Accordingly, we formulate a constrained optimization problem to determine the optimal client-specific sampling ratios $\{\alpha_{n,m}\}_{m=1}^M$, leading to the solution (see Appendix F.1 for details):

$$\alpha_{n,m} \propto \frac{\lambda_1 \alpha_n + \frac{\nu_m - \mu_m}{\pi_{n,m}} + 4G\gamma_n w_m - (1-\gamma_n)d_m}{1 + \frac{1-\alpha_n}{\alpha_n}\pi_{n,m}}, \tag{9}$$

where $\lambda_1$, $\mu_m$ and $\nu_m$ are multipliers. To simplify the solution and derive a closed-form, interpretable expression, we relax the KKT conditions by eliminating the dual variables, setting $\mu_m = 0$ and $\nu_m = 0$, under the assumption that the optimal values of $\alpha_{n,m}$ lie strictly within the open interval $(0, 1)$. This assumption is practically reasonable. In realistic streaming environments, it is neither desirable nor feasible for a client to completely discard previously stored data ($\alpha_{n,m} = 1$) or to entirely ignore new samples from a given state ($\alpha_{n,m} = 0$). Either extreme leads to inefficient storage

usage and poor adaptability to evolving data distributions. Under this relaxed setting, we define the following score function for each state $m \in \mathcal{M}$:

$$\text{score}_n(m) = \frac{w_m - a_1 d_m + b_1}{1 + \frac{1-\alpha_n}{\alpha_n} \pi_{n,m}}, \tag{10}$$

where $a_1 = \frac{1-\gamma_n}{4G\gamma_n}$ and $b_1 = \frac{\lambda_1}{4G\gamma_n}$ are tunable constants. Intuitively, the score increases with the weight in the global objective function $w_m$, and decreases with the state-specific heterogeneity bound $d_m$ or the probability $\pi_{n,m}$ of client $n$ being in state $m$. To ensure feasibility and numerical stability, we apply a ReLU operation to filter out negative scores, and normalize the resulting values with respect to the state distribution $\boldsymbol{\pi}_n$:

$$\alpha_{n,m} = \frac{\alpha_n \cdot \text{ReLU}(\text{score}_n(m))}{\sum_{m'=1}^{M} \pi_{n,m'} \cdot \text{ReLU}(\text{score}_n(m'))} \tag{11}$$

To deal with the extreme case that $\alpha_{n,m} > 1$, we extend (11) and compute $\alpha_{n,m}$ using Algorithm 2.

**Shift-aware Aggregation Weights (SAW).** To enable global aggregation that adapts to heterogeneous client behaviors, we optimize the aggregation weights $p_n$ via a relaxed surrogate objective (see Appendix F.2 for details).

$$\min_{p_n} L\eta\sigma^2 \sum_{n=1}^{N} p_n^2 q_n + \frac{5}{3} \sum_{n=1}^{N} p_n q_n s_n - \lambda_2 \sum_{n=1}^{N} p_n q_n$$

$$\text{s.t.} \sum_{n=1}^{N} p_n = 1, \ p_n \geq 0,$$

where $\lambda_2$ is a balancing hyperparameter, and

$$s_n = L\eta\sigma^2 + \frac{\beta_n}{\alpha_n}(1-\gamma_n) + \frac{2G\gamma_n\beta_n'}{1-(1-\alpha_n)^2} + 2G\gamma_n \sum_{m=1}^{M} \left(\frac{1}{\alpha_n}\pi_{n,m}\alpha_{n,m} - w_m\right)^2. \tag{12}$$

We refer to $s_n$ as the *heterogeneity score* of client $n$, as it captures the combined effect of its latent state distribution $\boldsymbol{\pi}_n$ and data sampling strategy $\{\alpha_{n,m}\}_{m=1}^{M}$. The closed-form solution is given by:

$$p_n \propto \frac{1}{q_n} - a_2 \cdot s_n + b_2, \tag{13}$$

where $a_2$ and $b_2$ are tunable constants. Intuitively, clients with smaller $s_n$ are assigned higher aggregation weights. The term $1/q_n$ ensures fairness with respect to availability. We propose to determine more distinguishing aggregation weights for each client $n$ by leveraging available probability $q_n$ and heterogeneity score $s_n$ as follows:

$$p_n = \frac{\text{ReLU}(\frac{1}{q_n} - a_2 \cdot s_n + b_2)}{\sum_{n'=1}^{N} \text{ReLU}(\frac{1}{q_{n'}} - a_2 \cdot s_{n'} + b_2)}. \tag{14}$$

**Practical Implementation with Prediction Oracle.** The DDS and SAW modules are theoretically derived under the true latent state distribution $\boldsymbol{\pi}_n$ for each client $n$, which is not directly observable in practice. To make the framework practically implementable, we utilize a prediction oracle that provides an estimated distribution $\hat{\boldsymbol{\pi}}_n$. Accordingly, the sampling strategy is adapted as follows:

$$\widehat{\text{score}}_n(m) = \frac{w_m - a_1 d_m + b_1}{1 + \frac{\alpha_n}{1-\alpha_n}\hat{\pi}_{n,m}}, \quad \hat{\alpha}_{n,m} = \frac{\text{ReLU}(\widehat{\text{score}}_n(m))}{\sum_{m'=1}^{M} \hat{\pi}_{n,m'} \cdot \text{ReLU}(\widehat{\text{score}}_n(m'))}.$$

Similarly, the aggregation weights are computed using $\hat{\boldsymbol{\pi}}_n$ in place of $\boldsymbol{\pi}_n$ in all related terms.

$$\hat{p}_n = \frac{\text{ReLU}(\frac{1}{q_n} - a_2 \cdot \hat{s}_n + b_2)}{\sum_{n'=1}^{N} \text{ReLU}(\frac{1}{q_{n'}} - a_2 \cdot \hat{s}_{n'} + b_2)}.$$

The overall procedure of SFedPO is illustrated in Fig. 1, highlighting the key components on both the client and server sides. The complete procedure is summarized in Algorithm 1.

While the above implementation enables SFedPO to operate using a prediction oracle, it is critical to understand how prediction errors may affect the theoretical performance. To this end, we perform a robustness analysis that quantifies the impact of prediction errors on the convergence guarantee. Specifically, we define the prediction error for each client $n$ as $\delta_n := \|\hat{\boldsymbol{\pi}}_n - \boldsymbol{\pi}_n\|_1$, and analyze how this error propagates into the convergence bound through the data sampling strategy $\boldsymbol{\alpha}$ and aggregation weights $\{p_n\}_{n \in \mathcal{N}}$, both of which are functions of $\boldsymbol{\pi}_n$ in theory but implemented based on $\hat{\boldsymbol{\pi}}_n$ in practice. Recall the convergence bound in Theorem 1, we focus on the following term:

$$\mathcal{B}(\alpha_{n,m}, p_n) := \frac{1}{\sum_{n=1}^{N} p_n q_n} \Big[ L\eta\sigma^2 \sum_{n=1}^{N} p_n^2 q_n + \frac{5}{3} \sum_{n=1}^{N} p_n q_n s_n \Big]. \tag{15}$$

**Theorem 2** (Robustness to prediction errors). *Assume that all sampling scores and aggregation weights remain strictly positive under both the true and estimated state distributions. Let the data sampling and aggregation strategies be constructed using the estimated distributions $\hat{\boldsymbol{\pi}}_n$ provided by a prediction oracle. Then, the convergence degradation compared to the ideal strategy using the true distributions $\boldsymbol{\pi}_n$ is bounded as*

$$|\mathcal{B}(\hat{\alpha}_{n,m}, \hat{p}_n) - \mathcal{B}(\alpha_{n,m}, p_n)| \leq \mathcal{O}\Big( \sum_{n=1}^{N} \delta_n \Big), \tag{16}$$

*where $\delta_n := \|\hat{\boldsymbol{\pi}}_n - \boldsymbol{\pi}_n\|_1$ denotes the estimation error of the prediction oracle for client $n$. Here $\mathcal{O}$ hides absolute constants.*

*Proof.* See details in Appendix G. ∎

## 6 EXPERIMENTS

### 6.1 EXPERIMENTAL SETUP

**Datasets and models.** We conduct comprehensive experiments on four public benchmark datasets: Fashion-MNIST (Xiao et al., 2017), CIFAR-10 (Krizhevsky et al., 2009), CINIC-10 (Darlow et al., 2018), and HAM10000 (Tschandl et al., 2018). We adopt LeNet-5 (LeCun et al., 1998) for Fashion-MNIST, AlexNet (Krizhevsky et al., 2012) for CIFAR-10 and CINIC-10, and a customized CNN for HAM10000. More details and model architectures, such as ResNet, are provided in Appendix H.

**Federated settings.** We simulate a federated environment consisting of $N = 30$ clients. In each communication round, clients are selected independently based on their availability probability $\{q_n\}_{n=1}^{N}$. For each client $n$, we draw $q_n \sim \mathcal{N}(0.2, 0.01^2)$, clipped to $[0.01, 1]$ to avoid degenerate participation. To capture temporally evolving local distributions, we introduce $M = 60$ latent states, each representing a possible data distribution encountered over time. To simulate intra-state heterogeneity, we organize these states into 6 clusters (10 states per cluster), each associated with a Dirichlet partitioning strategy with distinct concentration parameters $\alpha \in \{0.05, 0.1, 0.2, 0.5, 1.0, 100.0\}$. Based on this latent state space, we consider two heterogeneity scenarios. (1) Full-access (mild heterogeneity): Each client has non-zero access probability $\pi_{n,m} > 0$ for all states $m \in \{1, \dots, M\}$. (2) Partial-access (extreme heterogeneity): Each client is restricted to a randomly sampled subset of 10 states, with $\pi_{n,m} = 0$ for others. Moreover, 50% of clients are initialized with latent states drawn from high-heterogeneity clusters ($\alpha \in \{0.05, 0.1\}$), thereby amplifying both spatial and temporal heterogeneity. These two scenarios allow us to rigorously test the robustness of SFedPO under both mild and extreme data heterogeneity. Details are provided in Appendix H.1.

**Baselines.** We compare SFedPO against representative methods from four categories: (1) Online FL: FedOGD (Kwon et al., 2023) and FedOMD (Mitra et al., 2021); (2) FL for concept drift: AdapFedAvg (Canonaco et al., 2021), FedDrift (Jothimurugesan et al., 2023), and Flash (Panchal et al., 2023); (3) FCL: FedEWC (FedOGD with EWC (Kirkpatrick et al., 2017) applied to clients) and FLwF-2T (Usmanova et al., 2021); (4) Data selection methods in FL: Importance Sampling (IS) (Li et al., 2021), ODE (Gong et al., 2023), and DRSR (Wang et al., 2023a).

### 6.2 MAIN RESULTS

**Performance: test accuracy.** Table 1 presents the test accuracy of SFedPO and all baselines on four datasets under two scenarios. The results demonstrate that SFedPO consistently outperforms

Table 1: Test accuracy (%, mean±std on 5 trials) comparison of our SFedPO framework to other baselines in full and partial scenarios on several datasets.

| Method | Fashion-MNIST | | CIFAR-10 | | CINIC-10 | | HAM10000 | |
|---|---|---|---|---|---|---|---|---|
| | full | partial | full | partial | full | partial | full | partial |
| FedOGD | 85.04±0.46 | 82.67±0.68 | 51.14±2.23 | 41.06±2.09 | 43.86±0.90 | 34.40±1.91 | 54.11±0.51 | 45.28±0.62 |
| FedOMD | 85.03±0.41 | 82.67±0.70 | 51.06±2.22 | 40.98±2.08 | 43.84±0.91 | 34.37±1.91 | 54.14±0.55 | 45.21±0.67 |
| AdapFedAvg | 84.95±0.42 | 82.61±0.74 | 48.86±2.33 | 39.37±2.25 | 42.94±0.99 | 33.22±1.96 | 52.24±0.53 | 41.65±1.77 |
| FedDrift | 85.06±0.40 | 82.62±0.72 | 51.07±2.40 | 41.23±2.14 | 43.88±0.84 | 34.57±1.73 | 54.25±0.43 | 45.35±0.67 |
| Flash | 85.65±0.83 | 83.89±1.05 | 63.87±3.44 | 56.08±0.93 | 50.80±0.92 | 44.34±2.72 | 64.70±0.74 | 54.70±1.99 |
| FedEWC | 85.12±0.46 | 82.77±0.69 | 51.18±2.23 | 41.12±2.06 | 43.88±0.92 | 34.46±1.92 | 54.38±0.53 | 45.43±0.75 |
| FLwF-2T | 86.46±0.10 | 84.92±0.64 | 49.69±0.83 | 40.01±1.09 | 43.65±0.56 | 36.51±0.69 | 56.88±0.40 | 49.28±1.27 |
| IS | 83.36±0.73 | 80.27±1.33 | 52.03±1.87 | 42.66±2.02 | 37.92±0.82 | 30.86±1.39 | 55.32±1.99 | 42.99±3.18 |
| ODE | 83.33±0.42 | 80.21±0.79 | 53.52±1.71 | 40.69±1.57 | 38.02±0.90 | 31.31±1.28 | 55.32±0.89 | 43.31±2.64 |
| DRSR | 85.75±0.21 | 83.04±0.57 | 57.64±1.11 | 48.42±0.73 | 43.49±0.59 | 37.90±1.00 | 59.72±0.55 | 52.42±1.47 |
| **SFedPO** | **87.60±0.06** | **86.77±0.42** | **67.45±0.16** | **63.53±0.80** | **51.00±0.32** | **47.71±0.59** | **68.63±0.43** | **64.23±0.31** |

Table 2: Modularity. Accuracy (%) of classic federated learning methods with SFedPO and their improvement over the originals without SFedPO.

| Method | Fashion-MNIST | | CIFAR-10 | | CINIC-10 | | HAM10000 | |
|---|---|---|---|---|---|---|---|---|
| | full | partial | full | partial | full | partial | full | partial |
| FedAvg | 87.55(**+0.29**) | 86.63(**+0.13**) | 67.72(**+1.86**) | 63.83(**+4.01**) | 50.90(**+1.40**) | 47.90(**+1.18**) | 68.88(**+1.61**) | 64.53(**+1.52**) |
| FedProx | 87.23(**+0.34**) | 86.29(**+0.22**) | 65.21(**+1.88**) | 61.69(**+4.58**) | 50.08(**+1.66**) | 46.73(**+0.96**) | 67.14(**+1.80**) | 62.68(**+1.57**) |
| FedCurv | 87.62(**+0.42**) | 86.76(**+0.25**) | 67.60(**+1.75**) | 63.48(**+3.65**) | 50.97(**+1.49**) | 48.04(**+1.20**) | 68.94(**+1.19**) | 65.03(**+2.37**) |
| FedNTD | 87.50(**+0.09**) | 87.09(**+0.40**) | 65.63(**+2.09**) | 62.43(**+3.05**) | 52.61(**+1.08**) | 50.45(**+0.80**) | 68.56(**+1.20**) | 65.82(**+0.79**) |
| FedEXP | 85.07(**+1.35**) | 85.50(**+4.17**) | 65.71(**+1.93**) | 66.57(**+7.76**) | 43.65(**+1.14**) | 48.48(**+7.82**) | 65.72(**+2.14**) | 64.71(**+6.72**) |

all baselines across different settings. For instance, on the CIFAR-10 dataset, SFedPO surpasses all other methods by at least 3.58% in the full-access scenario and 7.39% in the partial-access scenario. We further observe that the performance gain of SFedPO in the partial-access scenario is more pronounced than in the full-access scenario, indicating that our method is particularly effective under extreme heterogeneity. This validates the effectiveness of our distribution-guided data sampling and aggregation strategies, which adaptively respond to state-specific and client-specific variation.

**Modularity: improvements over FL methods.** Our proposed SFedPO exhibits strong modularity and can be easily integrated into a wide range of classical FL methods as a plug-and-play module to cope with streaming data scenarios. To evaluate its effectiveness in this setting, we apply SFedPO's data sampling and aggregation strategies to several representative FL methods, including FedAvg (McMahan et al., 2017), FedProx (Li et al., 2020a), FedCurv (Shoham et al., 2019), FedNTD (Lee et al., 2022), and FedEXP (Jhunjhunwala et al., 2023). For comparison, we consider the original versions of these methods under same streaming settings, where each client performs uniform data sampling across latent states (i.e., $\alpha_{n,m} = \alpha_n$ for all $m$), and the server performs uniform model averaging over the participating clients (i.e., $p_n = \frac{1}{|S_r|}$ for each $n \in S_r$). As shown in Table 2, integrating SFedPO consistently improves the test accuracy of all methods across different datasets and both full- and partial-access scenarios. The gains are particularly notable under partial access, where client heterogeneity is more severe, confirming that SFedPO enhances the robustness and adaptability of FL methods under dynamic data distributions.

## 6.3 ABLATION STUDY

**Effects of different configurations.** We first evaluate the robustness of SFedPO to different configuration parameters in the FL environment. Specifically, we vary three core parameters, including time step ($T \in \{2, 5, 8, 10\}$), training round ($R \in \{50, 100, 150, 200\}$), and data capacity of clients ($D \in \{250, 500, 750, 1000\}$), then respectively show the performance of our method and four baselines in Fig. 2a, Fig. 2b, and Fig. 2c. The experimental results demonstrate that our method outperforms all baseline approaches across different parameters. All the experiments in this part are conducted in the partial scenario on the CIFAR-10 dataset.

**Effectiveness and robustness of modules.** We evaluate the stability and effectiveness of the two core components in SFedPO: Distribution-guided Data Sampling (DDS) and Shift-aware Aggregation Weights (SAW). For the DDS module, we first vary the sampling budget ($\alpha_n \in$

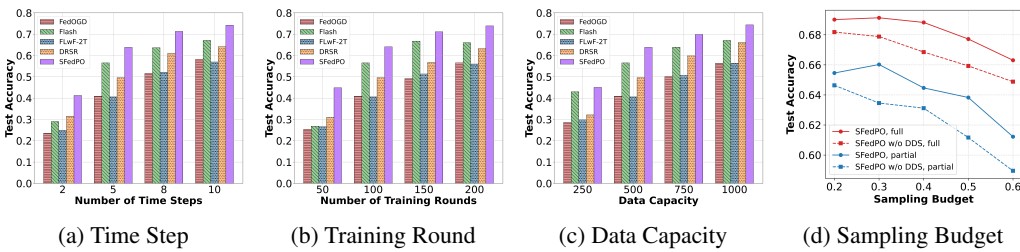

(a) Time Step     (b) Training Round     (c) Data Capacity     (d) Sampling Budget

Figure 2: The impact of four key parameters on performance.

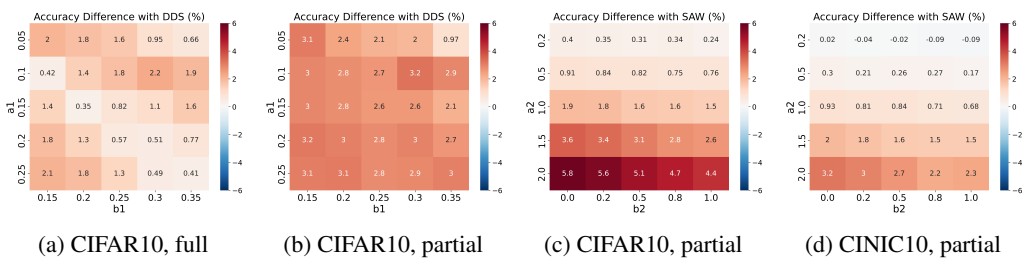

(a) CIFAR10, full     (b) CIFAR10, partial     (c) CIFAR10, partial     (d) CINIC10, partial

Figure 3: Hyperparameter sensitivity analysis for $\{a_1, b_1\}$ and $\{a_2, b_2\}$.

$\{0.2, 0.3, 0.4, 0.5, 0.6\}$) and compare the performance of the standard SFedPO with an ablated version where clients sample data uniformly across latent states (i.e., $\alpha_{n,m} = \alpha_n$ for all $m$). Fig.2d demonstrates that the DDS module consistently improves training performance across all sampling budgets. We then tune the hyperparameters of $\{a_1, b_1\}$ in (87) across diverse datasets and scenarios. Fig. 3a and Fig. 3b illustrate that DDS consistently improves performance across a wide range of hyperparameter settings ($a_1 \in [0.05, 0.25]$, $b_1 \in [0.15, 0.35]$). For the SAW module, we investigate the impact of parameters $a_2, b_2$ in (13). As shown in Fig. 3c and Fig. 3d, SAW brings noticeable accuracy gains over uniform aggregation ($p_n = 1/|S_r|$), with stable performance across a wide range of parameter values ($a_2 \in [0.5, 2.0]$, $b_2 \in [0.0, 1.0]$). These results collectively confirm that both DDS and SAW are not only effective but also resilient to hyperparameter changes.

**Effects of Prediction Error.** To evaluate the robustness of SFedPO against errors in the prediction oracle, we conduct an empirical study aligned with our theoretical analysis in Theorem 2. To simulate prediction errors, we introduce additive perturbations to the ground-truth $\boldsymbol{\pi}_n$ to obtain noisy estimates $\hat{\boldsymbol{\pi}}_n$, and apply SFedPO based on $\hat{\boldsymbol{\pi}}_n$ to perform local data sampling and global aggregation. Specifically, we perturb each state probability by a random noise uniformly drawn from $[-\epsilon, \epsilon]$, followed by renormalization to ensure $\sum_m \hat{\pi}_{n,m} = 1$. We vary $\epsilon$ from 0.00 to 0.10 to simulate increasing levels of oracle error, and report the resulting model accuracy in Table 3. We observe that SFedPO exhibits stable performance under varying degrees of perturbation on the CIFAR-10 dataset.

Table 3: Accuracy (%, mean±std on 5 trials) under different degrees of perturbation.

| Epsilon | 0.00 | 0.02 | 0.04 | 0.06 | 0.08 | 0.10 |
|---|---|---|---|---|---|---|
| full | $67.45_{\pm 0.16}$ | $66.90_{\pm 0.56}$ | $66.96_{\pm 0.45}$ | $66.97_{\pm 0.31}$ | $67.14_{\pm 0.47}$ | $66.95_{\pm 0.32}$ |
| partial | $63.52_{\pm 0.82}$ | $62.63_{\pm 0.16}$ | $62.36_{\pm 0.20}$ | $62.49_{\pm 0.14}$ | $62.48_{\pm 0.15}$ | $61.90_{\pm 0.15}$ |

## 7 CONCLUSIONS

We propose SFedPO, a streaming federated learning framework for dynamic environments with evolving local data distributions. Departing from conventional FL assumptions of static datasets or a stationary distribution, SFedPO incorporates a prediction oracle to capture the temporal evolution of client-side data distributions. Guided by theoretical convergence analysis, we develop two key components: a Distribution-guided Data Sampling (DDS) strategy that balances data reuse and distribution adaptation under storage constraints, and a Shift-aware Aggregation Weights (SAW) mechanism that adjusts global aggregation in response to client-specific sampling behavior. We further establish robustness guarantees under prediction errors. Extensive experiments demonstrate that SFedPO effectively adapts to streaming scenarios with distribution shifts and significantly outperforms existing methods.

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

# Appendix

## A  THE USE OF LLMs

This paper was proofread and linguistically polished with the assistance of a large language model (ChatGPT). All research ideas, methods, analyses, experiments, and conclusions are entirely the work of the authors.

## B  RELATED WORKS

This section provides a more detailed introduction to the second type of data stream paradigm, i.e., FL under distribution shifts, mainly from the perspectives of streaming FL, concept drift in FL, and federated continual learning.

**Streaming Federated Learning.**  To better capture real-world scenarios where data arrives continuously, recent studies have extended FL to the streaming setting. From a theoretical perspective, Marfoq et al. (2023) formalize FL over data streams and propose a general FL algorithm through a weighted empirical risk minimization, but still assume stationary distributions. Huynh et al. (2025) investigate convergence under data streams modeled by non-stationary Markov processes. From a data sampling perspective, ODE (Gong et al., 2023) introduces a selection method based on a data evaluation metric under storage constraints, but also relies on the stationary assumption. Wang et al. (2023a) assume the existence of a true long-term distribution in the local data stream of clients and proposes the cache update strategy to align the data distribution in the local cache to the underlying long-term distribution. Fed-HIST (Zhang et al., 2024) avoids the problem of raw data storage by retrieving model-based historical representations through similarity comparison. DYNAMITE (Liu et al., 2023) optimizes batch size and aggregation frequency under dynamic conditions, but it uses reservoir sampling and does not adjust data selection based on distribution changes. On the system side, Jin et al. (2021) formulate a latency-minimizing FL scheduling problem with online and bandit algorithms under budget and network constraints. Hu et al. (2024) develop a Lyapunov-based resource management scheme for streaming FL with adaptive control over computation and communication under long-term energy constraints. FedStream (Wang et al., 2023b) tackles dual heterogeneity in data and arrival patterns with asynchronous aggregation and local adaptation, but lacks theoretical justification. Different from the above studies, our work considers temporally evolving data distributions and develops a distribution-aware data sampling strategy grounded in theoretical analysis.

**Concept Drift in Federated Learning.**  While concept drift (Gama et al., 2014; Lu et al., 2018; Tahmasbi et al., 2021) has been widely studied in traditional machine learning, existing solutions often cannot be directly applied to FL due to the inherent heterogeneity across clients. Several works focus on detecting and responding to drift at the client level. FedConD (Chen et al., 2021) detects local drift based on historical model performance and adapts by adjusting the local regularization parameters. Flash (Panchal et al., 2023) detects drift via the magnitude of client updates and adapts the learning rate accordingly, while AdapFedAvg (Canonaco et al., 2021) passively adjusts learning rates to improve model plasticity under drift. Manias et al. (2021) detect drifted clients using dimensionality reduction and clustering on model updates, primarily aiming at client isolation rather than adaptation. Other approaches address drift through clustering-based strategies. Fielding (Li et al., 2024) detects concept drift via label distribution changes and selectively re-clusters clients to preserve cluster quality under heterogeneity. FedDrift (Jothimurugesan et al., 2023) formalizes staggered drift adaptation as a time-varying clustering problem and proposes hierarchical clustering algorithms guided by local drift detection. FedRC (Guo et al., 2024) proposes a bi-level optimization framework based on a clustering principle to address simultaneous feature, label distribution shifts, and concept shift in FL. FedCCFA (Chen et al., 2024) aligns client feature spaces under distributed concept drift by combining classifier clustering and entropy-based adaptive feature alignment. Most concept drift adaptation methods typically modify the model architecture (Chen et al., 2021), optimization strategy (Panchal et al., 2023; Canonaco et al., 2021), or client clustering (Li et al., 2024; Jothimurugesan et al., 2023; Chen et al., 2024) in response to detected shifts. In contrast, we propose a theoretically grounded client-specific data sampling strategy and model aggregation algorithm in streaming FL.

**Federated Continual Learning.**  Federated continual learning (FCL) (Yang et al., 2024; Wang et al., 2024) addresses evolving data but primarily focuses on mitigating catastrophic forgetting across sequential tasks. Guo et al. (2021) propose a FCL framework based on approximating prior

local objectives, while FedWeIT (Yoon et al., 2021) decomposes model weights into global and task-specific components to enable selective inter-client knowledge transfer. AF-FCL (Wuerkaixi et al., 2024) introduces a selective generative replay method for FCL that emphasizes accurate forgetting to discard biased knowledge across heterogeneous clients. FedCIL (Qi et al., 2023) introduces model consolidation and consistency enforcement to stabilize training on non-IID streaming tasks without storing historical data. Huang et al. (2022) and Usmanova et al. (2021) both incorporate knowledge distillation strategies to enhance generalization and mitigate forgetting in federated settings. GLFC (Dong et al., 2022) further proposes class-aware gradient compensation and proxy-based global model selection to handle class-incremental learning with dynamic client participation. While these FCL approaches focus on mitigating catastrophic forgetting across sequential tasks or enhancing model generalization under domain shifts, they do not explicitly model or predict the temporal evolution of data distributions. In contrast, our work leverages a predictive oracle to guide data sampling and optimization in streaming FL.

# C  DISCUSSIONS

## C.1  DISCUSSION ON THE PREDICTION ORACLE

In the main body of our work, we assume the existence of a prediction oracle that provides each client $n$ with an estimated state distribution $\hat{\boldsymbol{\pi}}_n$. In this section, we further discuss the applicability and plausibility of this assumption, clarifying under what conditions such an oracle is reasonable and how it can be instantiated in practice.

The prediction oracle assumption becomes reasonable when two conditions are satisfied: (i) the client data naturally evolves over time, leading to state transitions in the underlying distribution, and (ii) these transitions exhibit structural patterns that can be learned from historical observations. As examples, we present three representative application scenarios:

- **Mobility-driven environments.** In UAV networks or mobile sensing platforms, client data distributions shift due to physical movement across regions. In such settings, the underlying state can be naturally defined by spatial regions (e.g., grid cells or points of interest). Moreover, mobility patterns such as constrained flight paths, periodic patrol routes, or Markovian movement models provide structured trajectories from which the state distribution can be effectively inferred. As a result, the prediction oracle can leverage historical mobility traces to estimate future state distributions with reasonable accuracy.

- **Periodicity-driven environments.** In many domains, data distributions evolve according to recurring temporal patterns. Typical examples include transportation systems, where traffic intensity varies between rush hours and off-peak periods, and environmental monitoring, where sensor readings change with daily or seasonal cycles. In such cases, states can be naturally defined based on temporal segments. Since these periodic structures repeat over time, state distributions can be reliably estimated using historical data through statistical models.

- **Interaction-driven environments.** In recommendation systems, user–item interactions evolve in real time, producing non-stationary feedback streams. Here, the state can be defined as a representation of the user's latent preference profile, which evolves as the user interacts with new items. For instance, preference shifts may correspond to transitions between clusters of interaction features (e.g., genres of movies, categories of products, or communities of social content). Although individual actions are often noisy, the aggregated behavior of users tends to reveal structural dynamics such as preference drifts, trending items, or temporal co-occurrence patterns. By clustering interaction features and modeling transitions across preference states, these dynamics can be captured using online learning or probabilistic estimators.

In all these scenarios, the oracle does not need to be perfectly accurate: even approximate estimates of the state distribution, obtained through lightweight predictors or Bayesian updates from past data, are sufficient for guiding the DDS and SAW mechanisms in our framework. This demonstrates that the prediction oracle is not a restrictive abstraction but rather a broadly applicable tool in streaming federated learning settings. Moreover, our experimental results (Section 6 and Appendix H) further confirm that the framework remains robust under various levels of prediction inaccuracies. To make this abstraction more concrete, we next illustrate how such an oracle can be instantiated in practice through a Bayesian estimation framework.

**Bayesian Estimation Framework.** Suppose that at time $t$, client $n$ has observed a sequence of states $\mathcal{S}_{n,t} = \{s_1, s_2, \ldots, s_t\}$, where each $s_i \in \{1, 2, \ldots, M\}$. We model the state generation process as a categorical distribution parameterized by $\boldsymbol{\pi}_n$. To estimate $\boldsymbol{\pi}_n$, we adopt a Bayesian approach by placing a Dirichlet prior:

$$\boldsymbol{\pi}_n \sim \mathrm{Dir}(\boldsymbol{\alpha}_0),$$

where $\boldsymbol{\alpha}_0 = (\alpha_0^1, \alpha_0^2, \ldots, \alpha_0^M)$ is a prior belief.

Given the observed state counts $\mathbf{c}_t = (c_t^1, c_t^2, \ldots, c_t^M)$, where $c_t^m = \sum_{i=1}^{t} \mathbb{I}[s_i = m]$, the posterior distribution becomes:

$$\boldsymbol{\pi}_n \mid \mathcal{S}_{n,t} \sim \mathrm{Dir}(\boldsymbol{\alpha}_0 + \mathbf{c}_t),$$

and the posterior mean is:

$$\hat{\pi}_{n,m} = \mathbb{E}[\pi_{n,m} \mid \mathcal{S}_{n,t}] = \frac{\alpha_0^m + c_t^m}{\sum_{j=1}^{M}(\alpha_0^j + c_t^j)}.$$

The posterior variance of each component $\pi_n^m$ under the Dirichlet distribution is given by:

$$\mathrm{Var}[\pi_n^m \mid \mathcal{S}_{n,t}] = \frac{(\alpha_0^m + c_t^m)(S_t - \alpha_0^m - c_t^m)}{S_t^2(S_t + 1)},$$

where $S_t = \sum_{j=1}^{M}(\alpha_0^j + c_t^j) = \sum_{j=1}^{M}\alpha_0^j + t$.

This expression reveals that the estimation error decreases as the number of observed samples increases (i.e., as $c_t^m$ grows), leading to more confident and accurate estimates over time. The variance scales approximately as $\mathcal{O}(1/t)$.

The above procedure provides a concrete and theoretically grounded method for constructing a prediction oracle. It enables dynamic tracking of local state distributions with low overhead, and its estimation error diminishes with increasing historical information. This property is especially suitable for streaming federated learning settings, where clients accumulate observations over time and require increasingly accurate guidance for data sampling and model aggregation.

For datasets with no clear prior structure, practitioners can adopt adaptive latent-state modeling. A lightweight solution is to monitor whether new observations are poorly explained by all existing states; when their distribution significantly deviates, a new state is created using concept-drift detection or distance-based criteria. This requires only minimal modification to our framework and enables the latent state space to expand dynamically as needed. Moreover, the prior of each state in our Bayesian oracle can be updated according to its empirical frequency.

### C.2 DISCUSSION ON THE HETEROGENEITY BOUNDS

While our theoretical framework defines $d_m$ as an upper bound on the gradient variance under state $m$, estimating gradient variance directly is often impractical in streaming scenarios with limited and dynamic local data.

Inspired by prior works such as Fed-CBS (Zhang et al., 2023) and FedDisco (Ye et al., 2023), we approximate $d_m$ using distance metrics of the class distribution, based on the intuition that skewed class distributions often induce unstable local updates, which in turn imply higher gradient variance. This approximation enables a practical and data-accessible estimation of $d_m$.

We further provide a theoretical insight to support this approximation. Specifically, the local objective gradient $\nabla F_m(x)$ on client $m$ can be decomposed over its label distribution (Zhang et al., 2023):

$$\nabla F_m(x) = \sum_{c=1}^{C} p_m^c \nabla F_m^c(x),$$

where $p_m^c$ is the proportion of class $c$ on client $m$, and $\nabla F_m^c(x)$ is the gradient of the loss conditioned on class $c$. Then, we note that:

$$||\nabla F_m(x) - \nabla F(x)||^2 = ||\sum_{c=1}^{C}(p_m^c - \frac{1}{C})\nabla F_m^c(x)||^2 \leq \sum_{c=1}^{C}(p_m^c - \frac{1}{C})^2 \cdot \sum_{c=1}^{C}||\nabla F_m^c(x)||^2.$$

Here, $\sum_{c=1}^{C}(p_m^c - \frac{1}{C})^2$ can be viewed as a kind of distance between the class distribution and the uniform distribution. Therefore, we have a strong intuition that there is a linear relationship between $d_m$ and the distance.

To support this approximation, we present an empirical analysis in Appendix H comparing four metrics—L1, L2, KL, and JS divergence—and find a consistent correlation between distributional divergence and update instability. These results validate the suitability of our approximation in realistic settings.

### C.3 DISCUSSION ON THE THEOREMS

To clarify Theorem 1, it provides an upper bound on the expected squared gradient norm under a fixed data sampling strategy $\alpha$ and aggregation weights $\{p_n\}$. This bound characterizes the optimization error of our FL framework in a streaming setting with non-stationary data.

Key terms and their interpretations are as follows:

- $\alpha_n$ and $\alpha_{n,m}$ capture the sampling budget of client $n$ and its sampling ratio in state $m$, representing the impact of data sampling on convergence.

- $\beta_n$ and $\beta'_n$ measure the heterogeneity of local updates and sampling behaviors, which increases the variance of gradient estimates.

- $\gamma_n$ reflects the cumulative influence of retained old data across time steps.

- The term involving $\pi_{n,m}$ quantifies the state distribution; for details on its impact on convergence, see Theorem 2 regarding the effect of the prediction oracle error.

Compared with the static-data setting, our bound explicitly accounts for time-varying and partially predictable client data, which does not appear in standard FL analyses. This provides a theoretical guarantee that bridges classical static FL and fully adversarial online FL.

Theorem 2 characterizes the impact of imperfect predictions from the oracle on the convergence of our FL framework. Specifically, it shows that if the data sampling and aggregation strategies are constructed using estimated state distributions $\hat{\boldsymbol{\pi}}_n$, the resulting convergence degradation compared to using the true distributions $\boldsymbol{\pi}_n$ is bounded linearly by the total prediction error $\sum_{n=1}^{N} \delta_n$, where $\delta_n = \|\hat{\boldsymbol{\pi}}_n - \boldsymbol{\pi}_n\|_1$. This theorem quantifies the robustness of our method to imperfect oracles: small estimation errors lead to proportionally small degradation in convergence. Compared to prior FL analyses, which assume either static distributions or fully adversarial online data, Theorem 2 provides a practical guarantee for partially predictable, streaming data, showing that our framework remains effective even when the oracle is imperfect.

## C.4 Discussion on the Algorithms

The sampling strategy in Algorithm 2 has complexity $\mathcal{O}(M^2)$ per client, and computing aggregation weights on the server costs $\mathcal{O}(N)$, where $M$ is the number of latent states and $N$ is the number of clients. Regarding the overall computational overhead during federated learning rounds, we consider the following two cases:

- **Offline setting.** When the oracle is fixed, $\alpha_{n,m}$ and $\{p_n\}$ are computed once before training. The added cost during FL rounds is therefore negligible compared to local computation and communication.

- **Online setting.** When the oracle is updated, each client updates its estimate after uploading its model. The sampling ratios used in the current round come from the previous estimate. This update is lightweight and can be overlapped with communication, imposing almost no extra wall-clock time.

Overall, the computation of sampling ratios and aggregation weights adds minimal overhead in both settings, and in our experiments (offline case), the additional cost is essentially zero relative to standard FL training.

## C.5 Discussion on the Distribution Update Modeling

In our framework, the convex update in Eq. (1) is directly aligned with the sampling mechanism: each client retains a portion of historical data and replaces the rest with newly arrived samples, naturally resulting in a linear mixture of the corresponding state distributions.

Alternative formulations are possible. For example, we can introduce kernel-based blending, where each client's local distribution is represented as a kernel mean embedding (KME) in a reproducing kernel Hilbert space (RKHS). Specifically, for client $n$ at time $t$ in round $r$, the local distribution can be represented as $\mu_{n,t}^{(r)} = \mathbb{E}_{x \sim \mathcal{D}_{n,t}^{(r)}}[\phi(x)]$, where $\phi(\cdot)$ denotes the feature mapping corresponding to a positive-definite kernel $k(\cdot, \cdot)$. The local KME update can then be written as a convex combination in the RKHS:

$$\mu_{n,t}^{(r)} = (1 - \alpha_{n,m_{n,t}^{(r)}}) \cdot \mu_{n,t-1}^{(r)} + \alpha_{n,m_{n,t}^{(r)}} \cdot \mu_{m_{n,t}^{(r)}}, \tag{17}$$

where $\mu_{m_{n,t}^{(r)}} = \mathbb{E}_{x \sim \mathcal{D}_{m_{n,t}^{(r)}}}[\phi(x)]$ is the KME of the newly observed state. This formulation preserves the original sampling strategy while mapping distributions into a high-dimensional feature space, enabling the representation to capture non-linear or multi-modal characteristics.

# D NOTATIONS AND TECHNICAL LEMMAS

## D.1 NOTATIONS

Table 4 summarizes the notations appearing in this paper.

Table 4: Summary of key notations.

| Symbol | Description |
|--------|-------------|
| $R, r$ | number, index of training rounds |
| $N, n$ | number, index of clients |
| $M, m$ | number, index of latent states |
| $T, t$ | number, index of local update steps |
| $S_r$ | set of participating clients in round $r$ |
| $\mathcal{D}_m$ | the data distribution of latent state $m$ |
| $\mathcal{D}_{n,t}^{(r)}$ | the data distribution of client $n$ at time $t$ in round $r$ |
| $\boldsymbol{\pi}_n$ | $(\pi_{n,1}, \ldots, \pi_{n,M})$ is a round-truth state distribution of client $n$ |
| $\pi_{n,m}$ | probability that client $n$ is in state $m$ |
| $\hat{\boldsymbol{\pi}}_n$ | $(\hat{\pi}_{n,1}, \ldots, \hat{\pi}_{n,M})$ is a prediction oracle of client $n$ |
| $\hat{\pi}_{n,m}$ | estimate of $\pi_{n,m}$ |
| $\alpha_{n,m}$ | the fraction of new samples from state $m$ that client $n$ samples |
| $p_n$ | the aggregation weight |
| $q_n$ | the available probability |
| $\eta$ | learning rate (or stepsize) |
| $L$ | $L$-smoothness constant (Asm. 1) |
| $\sigma^2$ | upper bound on variance of stochastic gradients at each client (Asm. 2) |
| $G$ | constant in Asm. 3 to bound the overall gradient heterogeneity across all latent states |
| $d_m$ | constants in Asm. 4 to bound the state-specific gradient heterogeneity |
| $F/F_{n,t}^{(r)}$ | global objective/local objective of client $n$ at time $t$ in the $r$-th round |
| $F_m$ | expected loss function on the distribution $\mathcal{D}_m$ of latent state $m$ |
| $w_m$ | the weight of $F_m$ in the global objective |
| $\bar{\mathbf{x}}^{(r)}$ | global model parameters in the $r$-th round |
| $\mathbf{x}_{n,t}^{(r)}$ | local model parameters of client $n$ after $t$ local steps in the $r$-th round |
| $\mathbf{g}_{n,t}^{(r)}$ | $\mathbf{g}_{n,t}^{(r)} \triangleq \nabla f(\mathbf{x}_{n,t-1}^{(r)}; \xi_{n,t})$ denotes the stochastic gradients of $F_{n,t}^{(r)}$ regarding $\mathbf{x}_{n,t-1}^{(r)}$ |

## D.2 LEMMAS

**Jensen's inequality.** Let $h : \mathbb{R}^d \to \mathbb{R}$ be a convex function. For any vectors $\boldsymbol{x}_1, \ldots, \boldsymbol{x}_n \in \mathbb{R}^d$ and any non-negative weights $\lambda_1, \ldots, \lambda_n$ satisfying $\sum_{i=1}^n \lambda_i = 1$, it holds that

$$h\Big( \sum_{i=1}^n \lambda_i \boldsymbol{x}_i \Big) \leq \sum_{i=1}^n \lambda_i h(\boldsymbol{x}_i). \tag{18}$$

As a special case with $h(\boldsymbol{x}) = \|\boldsymbol{x}\|^2$, we obtain

$$\Big\| \frac{1}{n} \sum_{i=1}^n \boldsymbol{x}_i \Big\|^2 \leq \frac{1}{n} \sum_{i=1}^n \|\boldsymbol{x}_i\|^2. \tag{19}$$

**Lemma 1.** *Let $\{\mathbf{z}_t\}_{t=1}^T$ be a sequence of random vectors adapted to the underlying filtration $\{\mathcal{F}_t\}$ such that for all $t$, $\mathbb{E}[\mathbf{z}_t | \mathcal{F}_{t-1}] = 0$ and $\mathbb{E}[\|\mathbf{z}_t\|^2 | \mathcal{F}_{t-1}] \leq \sigma^2$. Then, the following identity holds:*

$$\mathbb{E}\Big[ \| \sum_{t=1}^T \mathbf{z}_t \|^2 \Big] \leq T\sigma^2. \tag{20}$$

This lemma provides a standard bound on the variance of a martingale difference sequence and will be used to control the accumulation of stochastic gradient noise over time.

**Lemma 2.** *Suppose we have a sequence $\{a_t\}$ satisfying the recursive inequality $a_t \leq (1-\alpha)a_{t-1}+\beta$, where $\alpha \in (0,1)$ and $\beta > 0$ are constants. Then, for any $t \geq 1$, the sequence is bounded as:*

$$a_t \leq a_0(1-\alpha)^t + \frac{\beta}{\alpha}\big(1 - (1-\alpha)^t\big). \tag{21}$$

*Proof.* We prove this by iteratively unfolding the recursion. Divide both sides of the inequality by $(1-\alpha)^t$:

$$\frac{a_t}{(1-\alpha)^t} \leq \frac{a_{t-1}}{(1-\alpha)^{t-1}} + \frac{\beta}{(1-\alpha)^t} \tag{22}$$

$$\leq \frac{a_{t-2}}{(1-\alpha)^{t-2}} + \frac{\beta}{(1-\alpha)^{t-1}} + \frac{\beta}{(1-\alpha)^t} \tag{23}$$

$$\vdots \tag{24}$$

$$\leq a_0 + \sum_{i=1}^{t} \frac{\beta}{(1-\alpha)^i}. \tag{25}$$

Multiplying both sides by $(1-\alpha)^t$ yields:

$$a_t \leq (1-\alpha)^t \Big(a_0 + \sum_{i=1}^{t} \frac{\beta}{(1-\alpha)^i}\Big) \tag{26}$$

$$= a_0(1-\alpha)^t + \beta \sum_{i=1}^{t}(1-\alpha)^{t-i} \tag{27}$$

$$= a_0(1-\alpha)^t + \frac{\beta}{\alpha}\big(1 - (1-\alpha)^t\big). \tag{28}$$

$\square$

## E  PROOF OF THEOREM 1

The global objective function is $F(\mathbf{x}) = \sum_{m=1}^{M} w_m F_m(\mathbf{x})$. We begin by analyzing the global model update across one communication round, which can be expressed as:

$$\Delta \mathbf{x} := \bar{\mathbf{x}}^{(r+1)} - \bar{\mathbf{x}}^{(r)} = -\eta \sum_{n \in S_r} p_n \sum_{t=1}^{T} \mathbf{g}_{n,t}^{(r)}, \tag{29}$$

where $\mathbf{g}_{n,t}^{(r)}$ denotes the stochastic gradient computed by client $n$ at time step $t$ in round $r$. According to Assumption 2, these gradients are unbiased estimators of the local loss, i.e., $\mathbb{E}[\mathbf{g}_{n,t}^{(r)}] = \nabla F_{n,t}^{(r)}(\mathbf{x}_{n,t}^{(r)})$. In the following, we focus on a single training round, and hence we drop the superscript $(r)$ for clarity. For example, we write $\mathbf{x}_{n,t}$ for $\mathbf{x}_{n,t}^{(r)}$ and use $F_{n,t}$ to replace $F_{n,t}^{(r)}$. Moreover, let $\mathbf{x}$ denote the initial local model $\mathbf{x}_{n,0}^{(r)}$. Unless otherwise stated, the expectation is conditioned on the global model $\bar{\mathbf{x}}^{(r)}$ and the participating client set $S_r$. Since $F(\cdot)$ is $L$-smooth, we get:

$$\mathbb{E}\big[F(\mathbf{x} + \Delta \mathbf{x}) - F(\mathbf{x})\big] \leq \mathbb{E}\big[\langle \nabla F(\mathbf{x}), \Delta \mathbf{x} \rangle\big] + \frac{L}{2} \mathbb{E}\|\Delta \mathbf{x}\|^2. \tag{30}$$

We now proceed to bound the two terms in the RHS of (30) separately.

**Bounding $\mathbb{E}\big[\langle \nabla F(\mathbf{x}), \Delta \mathbf{x} \rangle\big]$ in (30).**

$$A := \mathbb{E}\big[\langle \nabla F(\mathbf{x}), \Delta \mathbf{x} \rangle\big] = -\eta \sum_{n \in S_r} p_n \sum_{t=1}^{T} \mathbb{E}\big[\langle \nabla F(\mathbf{x}), \nabla F_{n,t}(\mathbf{x}_{n,t-1}) \rangle\big] \tag{31}$$

$$= -\frac{\eta}{2} \sum_{n \in S_r} p_n \sum_{t=1}^{T} \|\nabla F(\mathbf{x})\|^2 - \frac{\eta}{2} \sum_{n \in S_r} p_n \sum_{t=1}^{T} \mathbb{E}\big\|\nabla F_{n,t}(\mathbf{x}_{n,t-1})\big\|^2$$

$$+ \frac{\eta}{2} \sum_{n \in S_r} p_n \sum_{t=1}^{T} \mathbb{E}\big\|\nabla F(\mathbf{x}) - \nabla F_{n,t}(\mathbf{x}_{n,t-1})\big\|^2 \tag{32}$$

$$\leq -\frac{\eta}{2} \sum_{n \in S_r} p_n \sum_{t=1}^{T} \|\nabla F(\mathbf{x})\|^2 - \frac{\eta}{2} \sum_{n \in S_r} p_n \sum_{t=1}^{T} \mathbb{E}\big\|\nabla F_{n,t}(\mathbf{x}_{n,t-1})\big\|^2$$

$$+ \frac{\eta}{2} \sum_{n \in S_r} p_n \sum_{t=1}^{T} \Big(2\mathbb{E}\big\|\nabla F_{n,t}(\mathbf{x}_{n,t-1}) - \nabla F_{n,t}(\mathbf{x})\big\|^2 + 2\mathbb{E}\big\|\nabla F_{n,t}(\mathbf{x}) - \nabla F(\mathbf{x})\big\|^2\Big) \tag{33}$$

$$\leq -\frac{\eta}{2} \sum_{n \in S_r} p_n \sum_{t=1}^{T} \|\nabla F(\mathbf{x})\|^2 - \frac{\eta}{2} \sum_{n \in S_r} p_n \sum_{t=1}^{T} \mathbb{E}\big\|\nabla F_{n,t}(\mathbf{x}_{n,t-1})\big\|^2$$

$$+ \frac{\eta}{2} \sum_{n \in S_r} p_n \sum_{t=1}^{T} (2L^2 \mathbb{E}\|\mathbf{x}_{n,t-1} - \mathbf{x}\|^2 + 2\mathbb{E}\big\|\nabla F_{n,t}(\mathbf{x}) - \nabla F(\mathbf{x})\big\|^2) \tag{34}$$

$$= -\frac{T\eta}{2} \sum_{n \in S_r} p_n \|\nabla F(\mathbf{x})\|^2 - \frac{\eta}{2} \sum_{n \in S_r} p_n \sum_{t=1}^{T} \mathbb{E}\big\|\nabla F_{n,t}(\mathbf{x}_{n,t-1})\big\|^2$$

$$+ L^2 \eta \sum_{n \in S_r} p_n \sum_{t=1}^{T} \mathbb{E}\|\mathbf{x}_{n,t-1} - \mathbf{x}\|^2 + \eta \sum_{n \in S_r} p_n \sum_{t=1}^{T} \mathbb{E}\big\|\nabla F_{n,t}(\mathbf{x}) - \nabla F(\mathbf{x})\big\|^2, \tag{35}$$

where (32) applies the identity $\langle a, b \rangle = \frac{1}{2}\|a\|^2 + \frac{1}{2}\|b\|^2 - \frac{1}{2}\|a - b\|^2$, (33) uses the inequality $\|a - b\|^2 \leq 2\|a - c\|^2 + 2\|b - c\|^2$ by inserting the intermediate term $\nabla F_{n,t}(\mathbf{x})$, and then (34) uses $L$-smoothness.

**Bounding $\frac{L}{2}\mathbb{E}\|\Delta\mathbf{x}\|^2$ in (30).**

$$B := \frac{L}{2}\mathbb{E}\|\Delta\mathbf{x}\|^2 = \frac{L}{2}\eta^2\mathbb{E}\Big\|\sum_{n\in S_r} p_n \sum_{t=1}^{T} \mathbf{g}_{n,t}\Big\|^2 \tag{36}$$

$$\leq L\eta^2\mathbb{E}\Big\|\sum_{n\in S_r} p_n \sum_{t=1}^{T}\big(\mathbf{g}_{n,t}-\nabla F_{n,t}(\mathbf{x}_{n,t-1})\big)\Big\|^2 + L\eta^2\mathbb{E}\Big\|\sum_{n\in S_r} p_n \sum_{t=1}^{T}\nabla F_{n,t}(\mathbf{x}_{n,t-1})\Big\|^2 \tag{37}$$

$$\leq L\eta^2 \sum_{n\in S_r} p_n^2 \cdot T\sigma^2 + L\eta^2|S_r|\sum_{n\in S_r} p_n^2 \cdot T\sum_{t=1}^{T}\mathbb{E}\big\|\nabla F_{n,t}(\mathbf{x}_{n,t-1})\big\|^2, \tag{38}$$

where (37) uses the inequality $\|a+b\|^2 \leq 2\|a\|^2 + 2\|b\|^2$. The first term in (38) is based on Lemma 1 and uses the fact that clients are independent of each other. The second term in (38) uses the Jensen's inequality. Combining bounds for $A$ and $B$, and substituting back into (30), we obtain:

$$\mathbb{E}\big[F(\mathbf{x}+\Delta\mathbf{x})-F(\mathbf{x})\big]$$

$$\leq -\frac{T\eta}{2}\sum_{n\in S_r} p_n\|\nabla F(\mathbf{x})\|^2 + LT\eta^2\sigma^2\sum_{n\in S_r} p_n^2 + \sum_{n\in S_r}\big(LT\eta^2|S_r|p_n^2 - \frac{\eta}{2}p_n\big)\sum_{t=1}^{T}\mathbb{E}\big\|\nabla F_{n,t}(\mathbf{x}_{n,t-1})\big\|^2$$

$$+ L^2\eta\sum_{n\in S_r} p_n \sum_{t=1}^{T}\mathbb{E}\|\mathbf{x}_{n,t-1}-\mathbf{x}\|^2 + \eta\sum_{n\in S_r} p_n \sum_{t=1}^{T}\mathbb{E}\big\|\nabla F_{n,t}(\mathbf{x})-\nabla F(\mathbf{x})\big\|^2 \tag{39}$$

$$\leq -\frac{T\eta}{2}\sum_{n\in S_r} p_n\|\nabla F(\mathbf{x})\|^2 + LT\eta^2\sigma^2\sum_{n\in S_r} p_n^2 + L^2\eta\sum_{n\in S_r} p_n \sum_{t=1}^{T}\mathbb{E}\|\mathbf{x}_{n,t-1}-\mathbf{x}\|^2$$

$$+ \eta\sum_{n\in S_r} p_n \sum_{t=1}^{T}\mathbb{E}\|\nabla F_{n,t}(\mathbf{x})-\nabla F(\mathbf{x})\|^2, \tag{40}$$

where (40) holds under the condition that for all $n\in S_r$ we have $LT|S_r|\eta^2 p_n^2 \leq \frac{\eta}{2}p_n$, which is naturally satisfied when $LT\eta \leq \frac{1}{2M}$.

**Bounding $\sum_{t=1}^{T}\mathbb{E}\|\mathbf{x}_{n,t-1}-\mathbf{x}\|^2$ in (40).**

$$C := \sum_{t=1}^{T}\mathbb{E}\|\mathbf{x}_{n,t-1}-\mathbf{x}\|^2 = \sum_{t=1}^{T}\mathbb{E}\Big\|-\eta\sum_{i=1}^{t-1}\mathbf{g}_{n,i}\Big\|^2 \tag{41}$$

$$\leq 4\eta^2\sum_{t=1}^{T}\mathbb{E}\Big[\Big\|\sum_{i=1}^{t-1}\mathbf{g}_{n,i}-\sum_{i=1}^{t-1}\nabla F_{n,i}(\mathbf{x}_{n,i-1})\Big\|^2 + \Big\|\sum_{i=1}^{t-1}\nabla F_{n,i}(\mathbf{x}_{n,i-1})-\sum_{i=1}^{t-1}\nabla F_{n,i}(\mathbf{x})\Big\|^2$$

$$+ \Big\|\sum_{i=1}^{t-1}\nabla F_{n,i}(\mathbf{x})-\sum_{i=1}^{t-1}\nabla F(\mathbf{x})\Big\|^2 + \Big\|\sum_{i=1}^{t-1}\nabla F(\mathbf{x})\Big\|^2\Big] \tag{42}$$

$$\leq 4\eta^2\sum_{t=1}^{T}(t-1)\sigma^2 + 4\eta^2\sum_{t=1}^{T}(t-1)L^2\sum_{i=1}^{t-1}\mathbb{E}\|\mathbf{x}_{n,i-1}-\mathbf{x}\|^2$$

$$+ 4\eta^2\sum_{t=1}^{T}(t-1)\sum_{i=1}^{t-1}\mathbb{E}\big\|\nabla F_{n,i}(\mathbf{x})-\nabla F(\mathbf{x})\big\|^2 + 4\eta^2\sum_{t=1}^{T}(i-1)^2\|\nabla F(\mathbf{x})\|^2 \tag{43}$$

$$\leq 2T^2\eta^2\sigma^2 + 2L^2T^2\eta^2\cdot C + 2T^2\eta^2\sum_{t=1}^{T}\mathbb{E}\big\|\nabla F_{n,t}(\mathbf{x})-\nabla F(\mathbf{x})\big\|^2 + \frac{4}{3}T^3\eta^2\|\nabla F(\mathbf{x})\|^2, \tag{44}$$

where (42) uses the Jensen's inequality. The first term in (43) is based on Lemma 1. The second term in (38) uses the Jensen's inequality and $L$-smoothness. The third term in (43) uses the Jensen's inequality. (44) uses the fact that $\sum_{t=1}^{T}(t-1) \leq \frac{T^2}{2}$ and $\sum_{t=1}^{T}(t-1)^2 \leq \frac{T^3}{3}$.

After rearranging the preceding inequality and using $2L^2T^2\eta^2 < 1$, we get

$$C \leq \frac{1}{1-2L^2T^2\eta^2}\Big[2T^2\eta^2\sigma^2 + 2T^2\eta^2\sum_{t=1}^{T}\mathbb{E}\big\|\nabla F_{n,i}(\mathbf{x})-\nabla F(\mathbf{x})\big\|^2 + \frac{4}{3}T^3\eta^2\|\nabla F(\mathbf{x})\|^2\Big]. \tag{45}$$

We denote that $c = LT\eta$. Then, plugging the bound of $C$ into (40), we get

$$\mathbb{E}\big[F(\mathbf{x}+\Delta\mathbf{x}) - F(\mathbf{x})\big] \leq \Big(\frac{4c^2}{3(1-2c^2)} - \frac{1}{2}\Big)T\eta \sum_{n \in S_r} p_n\|\nabla F(\mathbf{x})\|^2 + LT\eta^2\sigma^2 \sum_{n \in S_r} p_n^2$$

$$+ \frac{2L^2T^2\eta^3\sigma^2}{1-2c^2} \sum_{n \in S_r} p_n + \Big(\eta + \frac{2L^2T^2\eta^3}{1-2c^2}\Big) \sum_{n \in S_r} p_n \sum_{t=1}^{T} \mathbb{E}\big\|\nabla F_{n,t}(\mathbf{x}) - \nabla F(\mathbf{x})\big\|^2. \quad (46)$$

**Bounding $\|\nabla F_{n,t}(\mathbf{x}) - \nabla F(\mathbf{x})\|^2$ in (46).**

Recall that the local data distribution evolves according to $\mathcal{D}_{n,t} = (1 - \alpha_{n,m_{n,t}}) \cdot \mathcal{D}_{n,t-1} + \alpha_{n,m_{n,t}} \cdot \mathcal{D}_{m_{n,t}}$, where $m_{n,t} \in \mathcal{M}$ denotes the latent state of client $n$ at time step $t$, and $\alpha_{n,m_{n,t}} \in [0,1]$ is the sampling ratio defined in Section 3. Due to the linearity of expectation, the corresponding loss function also follows a similar recursive relationship:

$$F_{n,t} = (1 - \alpha_{n,m_{n,t}})F_{n,t-1} + \alpha_{n,m_{n,t}}F_{m_{n,t}}, \quad (47)$$

where $F_{m_{n,t}}$ denotes the loss associated with $\mathcal{D}_{m_{n,t}}$. We apply the Jensen's inequality and obtain:

$$\|\nabla F_{n,t}(\mathbf{x}) - \nabla F(\mathbf{x})\|^2 \leq (1 - \alpha_{n,m_{n,t}})\|\nabla F_{n,t-1}(\mathbf{x}) - \nabla F(\mathbf{x})\|^2 + \alpha_{n,m_{n,t}}\|\nabla F_{m_{n,t}}(\mathbf{x}) - \nabla F(\mathbf{x})\|^2. \quad (48)$$

We now take expectation on both sides of (48) with respect to the latent state $m_{n,t}$, under the ground-truth state distribution. By using the Assumption 4, we get:

$$\mathbb{E}\big\|\nabla F_{n,t}(\mathbf{x}) - \nabla F(\mathbf{x})\big\|^2 \quad (49)$$

$$\leq \Big(1 - \sum_{m=1}^{M} \pi_{n,m}\alpha_{n,m}\Big)\mathbb{E}\big\|\nabla F_{n,t-1}(\mathbf{x}) - \nabla F(\mathbf{x})\big\|^2 + \sum_{m=1}^{M} \pi_{n,m}\alpha_{n,m}\mathbb{E}\big\|\nabla F_m(\mathbf{x}) - \nabla F(\mathbf{x})\big\|^2 \quad (50)$$

$$\leq \Big(1 - \sum_{m=1}^{M} \pi_{n,m}\alpha_{n,m}\Big)\mathbb{E}\big\|\nabla F_{n,t-1}(\mathbf{x}) - \nabla F(\mathbf{x})\big\|^2 + \sum_{m=1}^{M} \pi_{n,m}\alpha_{n,m}d_m. \quad (51)$$

We now apply Lemma 2 to the recursive bound derived in (51). Let $a_t = \mathbb{E}\big\|\nabla F_{n,t}(\mathbf{x}) - \nabla F(\mathbf{x})\big\|^2$, $\alpha = \alpha_n \triangleq \sum_{m=1}^{M} \pi_{n,m}\alpha_{n,m}$, and $\beta = \beta_n \triangleq \sum_{m=1}^{M} \pi_{n,m}\alpha_{n,m}d_m$. Then, we get

$$\sum_{t=1}^{T} \mathbb{E}\big\|\nabla F_{n,t}(\mathbf{x}) - \nabla F(\mathbf{x})\big\|^2 \leq \frac{\beta_n}{\alpha_n}\Big(T - \sum_{t=1}^{T}(1-\alpha_n)^t\Big) + \sum_{t=1}^{T}(1-\alpha_n)^t \mathbb{E}\big\|\nabla F_{n,0}(\mathbf{x}) - \nabla F(\mathbf{x})\big\|^2. \quad (52)$$

Plugging this result into the bound in (46), we obtain:

$$\mathbb{E}\big[F(\mathbf{x} + \Delta\mathbf{x}) - F(\mathbf{x})\big]$$

$$\leq \Big(\frac{4c^2}{3(1-2c^2)} - \frac{1}{2}\Big)T\eta \sum_{n \in S_r} p_n\|\nabla F(\mathbf{x})\|^2 + LT\eta^2\sigma^2 \sum_{n \in S_r} p_n^2 + \frac{2L^2T^2\eta^3\sigma^2}{1-2c^2} \sum_{n \in S_r} p_n$$

$$+ \Big(T\eta + \frac{2L^2T^3\eta^3}{1-2c^2}\Big) \sum_{n \in S_r} p_n\Big[\frac{\beta_n}{\alpha_n}(1-\gamma_n) + \gamma_n\mathbb{E}\big\|\nabla F_{n,0}(\mathbf{x}) - \nabla F(\mathbf{x})\big\|^2\Big]. \quad (53)$$

where $\gamma_n = \frac{1}{T}\sum_{t=1}^{T}(1-\alpha_n)^t \in (0,1)$.

**Bounding $\mathbb{E}\big\|\nabla F_{n,0}(\mathbf{x}) - \nabla F(\mathbf{x})\big\|^2$ in (53).**

We observe that the local loss function $F_{n,t}^{(r)}$ can be expressed as a weighted mixture of loss functions associated with latent states $m \in \mathcal{M}$. Specifically, we suppose $F_{n,t}^{(r)} = \langle \vec{\boldsymbol{w}}_{n,r,t}, \vec{F} \rangle$, where $\vec{\boldsymbol{w}}_{n,r,t} = (w_{n,r,t}^{(1)}, w_{n,r,t}^{(2)}, \ldots, w_{n,r,t}^{(M)})$ denotes the mixture weights over the latent states and $\vec{F} = (F_1, F_2, \ldots, F_M)$ represents the vector of corresponding state-specific loss functions. Based on the recursive update rule of the local loss function in (47), the weight vector $\vec{\boldsymbol{w}}_{n,r,t}$ evolves as

$$\vec{\boldsymbol{w}}_{n,r,t} = (1 - \alpha_{n,m_{n,t}})\vec{\boldsymbol{w}}_{n,r,t-1} + \alpha_{n,m_{n,t}}\vec{\boldsymbol{e}}_{m_{n,t}}, \quad (54)$$

where $\vec{\boldsymbol{e}}_m$ denotes the one-hot unit vector corresponding to latent state $m$, with its $m$-th component equal to 1 and all others set to 0. By applying the Cauchy-Schwarz inequality, we get:

$$\|\nabla F_{n,0}^{(r)}(\bar{\mathbf{x}}^{(r)}) - \nabla F(\bar{\mathbf{x}}^{(r)})\|^2 \leq \|\vec{\boldsymbol{w}}_{n,r,0} - \vec{\boldsymbol{w}}\|^2 \cdot \sum_{m=1}^{M} \|\nabla F_m(\bar{\mathbf{x}}^{(r)})\|^2 \leq G\|\vec{\boldsymbol{w}}_{n,r,0} - \vec{\boldsymbol{w}}\|^2. \quad (55)$$

where $\vec{\boldsymbol{w}} = (w_1, w_2, \cdots, w_M)$ denotes the weight vector in the global objective, and Assumption 3 is applied.

To further bound the discrepancy between the client-specific mixture weights and the global target weights, we decompose the $\|\vec{\boldsymbol{w}}_{n,r,t} - \vec{\boldsymbol{w}}\|^2$ into two parts using the Jensen's inequality:

$$\|\vec{\boldsymbol{w}}_{n,r,t} - \vec{\boldsymbol{w}}\|^2 \le 2\Big\|\vec{\boldsymbol{w}}_{n,r,t} - \frac{1}{\alpha_n}\sum_{m=1}^{M}\pi_{n,m}\alpha_{n,m}\vec{\boldsymbol{e}}_m\Big\|^2 + 2\Big\|\frac{1}{\alpha_n}\sum_{m=1}^{M}\pi_{n,m}\alpha_{n,m}\vec{\boldsymbol{e}}_m - \vec{\boldsymbol{w}}\Big\|^2. \tag{56}$$

The first term in equation (56) quantifies the deviation between the actual weight vector and a fixed weight vector, which is fully determined by the distribution-aware data sampling strategy. The second term describes the inherent mismatch between the weights determined by the strategy and the global target weights. We deal with the first term in the following.

**Bounding $\big\|\vec{\boldsymbol{w}}_{n,r,t} - \frac{1}{\alpha_n}\sum_{m=1}^{M}\pi_{n,m}\alpha_{n,m}\vec{\boldsymbol{e}}_m\big\|^2$ in (56).**

$$\mathbb{E}\Big\|\vec{\boldsymbol{w}}_{n,r,t} - \frac{1}{\alpha_n}\sum_{m=1}^{M}\pi_{n,m}\alpha_{n,m}\vec{\boldsymbol{e}}_m\Big\|^2 \tag{57}$$

$$=\mathbb{E}\|\vec{\boldsymbol{w}}_{n,r,t}\|^2 - \frac{2}{\alpha_n}\mathbb{E}\Big[\big\langle\vec{\boldsymbol{w}}_{n,r,t}, \sum_{m=1}^{M}\pi_{n,m}\alpha_{n,m}\vec{\boldsymbol{e}}_m\big\rangle\Big] + \frac{1}{\alpha_n^2}\sum_{m=1}^{M}\pi_{n,m}^2\alpha_{n,m}^2 \tag{58}$$

$$=\mathbb{E}\big\|(1-\alpha_{n,m_{n,t}})\vec{\boldsymbol{w}}_{n,r,t-1} + \alpha_{n,m_{n,t}}\vec{\boldsymbol{e}}_{m_{n,t}}\big\|^2 - \frac{2}{\alpha_n}\mathbb{E}\Big[\big\langle(1-\alpha_{n,m_{n,t}})\vec{\boldsymbol{w}}_{n,r,t-1} + \alpha_{n,m_{n,t}}\vec{\boldsymbol{e}}_{m_{n,t}},$$

$$\sum_{m=1}^{M}\pi_{n,m}\alpha_{n,m}\vec{\boldsymbol{e}}_m\big\rangle\Big] + \frac{1}{\alpha_n^2}\sum_{m=1}^{M}\pi_{n,m}^2\alpha_{n,m}^2 \tag{59}$$

$$=\sum_{m=1}^{M}\pi_{n,m}(1-\alpha_{n,m})^2\mathbb{E}\|\vec{\boldsymbol{w}}_{n,r,t-1}\|^2 + 2\sum_{m=1}^{M}\pi_{n,m}\alpha_{n,m}(1-\alpha_{n,m})\mathbb{E}\big[\langle\vec{\boldsymbol{w}}_{n,r,t-1}, \vec{\boldsymbol{e}}_m\rangle\big] + \sum_{m=1}^{M}\pi_{n,m}\alpha_{n,m}^2$$

$$-\frac{2(1-\alpha_n)}{\alpha_n}\sum_{m=1}^{M}\pi_{n,m}\alpha_{n,m}\mathbb{E}\big[\langle\vec{\boldsymbol{w}}_{n,r,t-1}, \vec{\boldsymbol{e}}_m\rangle\big] - \frac{2}{\alpha_n}\sum_{m=1}^{M}\pi_{n,m}^2\alpha_{n,m}^2 + \frac{1}{\alpha_n^2}\sum_{m=1}^{M}\pi_{n,m}^2\alpha_{n,m}^2 \tag{60}$$

$$=\sum_{m=1}^{M}\pi_{n,m}(1-\alpha_{n,m})^2\mathbb{E}\|\vec{\boldsymbol{w}}_{n,r,t-1}\|^2 + 2\sum_{m=1}^{M}\pi_{n,m}\alpha_{n,m}\Big(1-\alpha_{n,m} - \frac{1-\alpha_n}{\alpha_n}\Big)\mathbb{E}\big[\langle\vec{\boldsymbol{w}}_{n,r,t-1}, \vec{\boldsymbol{e}}_m\rangle\big]$$

$$+\sum_{m=1}^{M}\pi_{n,m}\alpha_{n,m}^2 + \Big(\frac{1}{\alpha_n^2} - \frac{2}{\alpha_n}\Big)\sum_{m=1}^{M}\pi_{n,m}^2\alpha_{n,m}^2 \tag{61}$$

$$=(1-\alpha_n)^2\mathbb{E}\|\vec{\boldsymbol{w}}_{n,r,t-1}\|^2 - \frac{2(1-\alpha_n)^2}{\alpha_n}\sum_{m=1}^{M}\pi_{n,m}\alpha_{n,m}\mathbb{E}\big[\langle\vec{\boldsymbol{w}}_{n,r,t-1}, \vec{\boldsymbol{e}}_m\rangle\big] + \frac{(1-\alpha_n)^2}{\alpha_n^2}\sum_{m=1}^{M}\pi_{n,m}^2\alpha_{n,m}^2$$

$$+\sum_{m=1}^{M}\pi_{n,m}\alpha_{n,m}^2 - \sum_{m=1}^{M}\pi_{n,m}^2\alpha_{n,m}^2 + 2\sum_{m=1}^{M}\pi_{n,m}\alpha_{n,m}(\alpha_n - \alpha_{n,m})\mathbb{E}\big[\langle\vec{\boldsymbol{w}}_{n,r,t-1}, \vec{\boldsymbol{e}}_m\rangle\big]$$

$$+\Big(\sum_{m=1}^{M}\pi_{n,m}\alpha_{n,m}^2 - \alpha_n^2\Big)\mathbb{E}\|\vec{\boldsymbol{w}}_{n,r,t-1}\|^2 \tag{62}$$

$$=(1-\alpha_n)^2\mathbb{E}\Big\|\vec{\boldsymbol{w}}_{n,r,t-1} - \frac{1}{\alpha_n}\sum_{m=1}^{M}\pi_{n,m}\alpha_{n,m}\vec{\boldsymbol{e}}_m\Big\|^2 + \sum_{m=1}^{M}\pi_{n,m}\alpha_{n,m}^2 - \sum_{m=1}^{M}\pi_{n,m}^2\alpha_{n,m}^2$$

$$+\sum_{m=1}^{M}\pi_{n,m}\alpha_{n,m}(\alpha_{n,m} - \alpha_n)\mathbb{E}\Big[\|\vec{\boldsymbol{w}}_{n,r,t-1}\|^2 - 2\langle\vec{\boldsymbol{w}}_{n,r,t-1}, \vec{\boldsymbol{e}}_m\rangle\Big] \tag{63}$$

$$=(1-\alpha_n)^2\mathbb{E}\Big\|\vec{\boldsymbol{w}}_{n,r,t-1} - \frac{1}{\alpha_n}\sum_{m=1}^{M}\pi_{n,m}\alpha_{n,m}\vec{\boldsymbol{e}}_m\Big\|^2 - \sum_{m=1}^{M}\pi_{n,m}^2\alpha_{n,m}^2$$

$$+\sum_{m=1}^{M}\pi_{n,m}\alpha_{n,m}(\alpha_{n,m} - \alpha_n)\mathbb{E}\big[\|\vec{\boldsymbol{w}}_{n,r,t-1} - \vec{\boldsymbol{e}}_m\|^2\big] + \alpha_n^2 \tag{64}$$

$$\le(1-\alpha_n)^2\mathbb{E}\Big\|\vec{\boldsymbol{w}}_{n,r,t-1} - \frac{1}{\alpha_n}\sum_{m=1}^{M}\pi_{n,m}\alpha_{n,m}\vec{\boldsymbol{e}}_m\Big\|^2 + 2\sum_{m=1}^{M}\pi_{n,m}\alpha_{n,m}^2 - \sum_{m=1}^{M}\pi_{n,m}^2\alpha_{n,m}^2 + \alpha_n^2. \tag{65}$$

Here, (62) splits the first term in (61) by applying the identity $\sum_{m=1}^{M} \pi_{n,m}(1-\alpha_{n,m})^2 = (1-\alpha_n)^2 + \sum_{m=1}^{M} \pi_{n,m}\alpha_{n,m}^2 - \alpha_n^2$, and splits the second term in (61) by applying the identity $1-\alpha_{n,m}-\frac{1-\alpha_n}{\alpha_n} = (\alpha_n - \alpha_{n,m}) - \frac{(1-\alpha_n)^2}{\alpha_n}$. (65) uses the fact that $\|\vec{w}_{n,r,t-1} - \vec{e}_m\|^2 \le 2$.

Then, we set $a_t = \mathbb{E}\|\vec{w}_{n,r,t} - \frac{1}{\alpha_n} \sum_{m=1}^{M} \pi_{n,m}\alpha_{n,m}\vec{e}_m\|^2$, and apply Lemma 2 to this sequence. Let $\alpha = \alpha_n' \triangleq 1 - (1-\alpha_n)^2$ and $\beta = \beta_n' \triangleq 2\sum_{m=1}^{M} \pi_{n,m}\alpha_{n,m}^2 - \sum_{m=1}^{M} \pi_{n,m}^2\alpha_{n,m}^2 + \alpha_n^2$.

We consider the number of times client $n$ has participated in training up to round $r$, denoted as $\tau_n(r)$. Let $r' < r$ denote the last round that client $n$ participated before round $r$. Then, since the local weight vector is only updated when the client participates, we have:

$$\mathbb{E}\left\|\vec{w}_{n,r,0} - \frac{1}{\alpha_n}\sum_{m=1}^{M} \pi_{n,m}\alpha_{n,m}\vec{e}_m\right\|^2 = \mathbb{E}\left\|\vec{w}_{n,r',T} - \frac{1}{\alpha_n}\sum_{m=1}^{M} \pi_{n,m}\alpha_{n,m}\vec{e}_m\right\|^2$$

$$\le \frac{\beta_n'}{1-(1-\alpha_n)^2} + (1-\alpha_n)^{2T\cdot\tau_n(r-1)} \cdot \mathbb{E}\left\|\vec{w}_{n,0,0} - \frac{1}{\alpha_n}\sum_{m=1}^{M} \pi_{n,m}\alpha_{n,m}\vec{e}_m\right\|^2 \tag{66}$$

$$\le \frac{\beta_n'}{1-(1-\alpha_n)^2} + 2(1-\alpha_n)^{2T\cdot\tau_n(r-1)}, \tag{67}$$

where the second inequality uses the fact that both $\vec{w}_{n,0,0}$ and $\frac{1}{\alpha_n}\sum_{m=1}^{M} \pi_{n,m}\alpha_{n,m}\vec{e}_m$ are probability vectors supported on the simplex, and hence their squared distance is upper bounded by 2.

Take expectation into the two sides of (53) about the selected client set in the round $r$, and combining the bounds in (56) and (67), we get

$$\mathbb{E}\left[F(\mathbf{x} + \Delta\mathbf{x}) - F(\mathbf{x})\right]$$

$$\le \left(\frac{4c^2}{3(1-2c^2)} - \frac{1}{2}\right)T\eta \sum_{n=1}^{N} p_n q_n \|\nabla F(\mathbf{x})\|^2 + LT\eta^2\sigma^2 \sum_{n=1}^{N} p_n^2 q_n + \frac{2L^2T^2\eta^3\sigma^2}{1-2c^2}\sum_{n=1}^{N} p_n q_n$$

$$+ \frac{T\eta}{1-2c^2}\sum_{n=1}^{N} p_n q_n\left[\frac{\beta_n}{\alpha_n}(1-\gamma_n) + \gamma_n G\|\vec{w}_{n,r,0} - \vec{w}\|^2\right] \tag{68}$$

$$\le \left(\frac{4c^2}{3(1-2c^2)} - \frac{1}{2}\right)T\eta \sum_{n=1}^{N} p_n q_n \|\nabla F(\mathbf{x})\|^2 + LT\eta^2\sigma^2 \sum_{n=1}^{N} p_n^2 q_n + \frac{2L^2T^2\eta^3\sigma^2}{1-2c^2}\sum_{n=1}^{N} p_n q_n$$

$$+ \frac{T\eta}{1-2c^2}\sum_{n=1}^{N} p_n q_n\left[\frac{\beta_n}{\alpha_n}(1-\gamma_n) + \gamma_n \cdot 2G\left(\frac{\beta_n'}{1-(1-\alpha_n)^2} + 2(1-\alpha_n)^{2T\tau_n(r-1)}\right.\right.$$

$$+ \left.\left.\left\|\frac{1}{\alpha_n}\sum_{m=1}^{M} \pi_{n,m}\alpha_{n,m}\vec{e}_m - \vec{w}\right\|^2\right)\right]. \tag{69}$$

Taking the expectation over the randomness in previous training rounds, conditioned on $\bar{\mathbf{x}}^{(r)}$, we obtain:

$$\mathbb{E}\left[F(\bar{\mathbf{x}}^{(r+1)}) - F(\bar{\mathbf{x}}^{(r)})\right] = \mathbb{E}\left[\mathbb{E}\left[F(\bar{\mathbf{x}}^{(r)} + \Delta\mathbf{x}) - F(\bar{\mathbf{x}}^{(r)})|\bar{\mathbf{x}}^{(r)}\right]\right]$$

$$\le \left(\frac{4c^2}{3(1-2c^2)} - \frac{1}{2}\right)T\eta \sum_{n=1}^{N} p_n q_n \mathbb{E}\|\nabla F(\bar{\mathbf{x}}^{(r)})\|^2 + LT\eta^2\sigma^2 \sum_{n=1}^{N} p_n^2 q_n + \frac{2L^2T^2\eta^3\sigma^2}{1-2c^2}\sum_{n=1}^{N} p_n q_n$$

$$+ \frac{T\eta}{1-2c^2}\sum_{n=1}^{N} p_n q_n\left[\frac{\beta_n}{\alpha_n}(1-\gamma_n) + \gamma_n \cdot 2G\left(\frac{\beta_n'}{1-(1-\alpha_n)^2} + 2\mathbb{E}\left[(1-\alpha_n)^{2T\cdot\tau_n(r-1)}\right]\right.\right.$$

$$+ \left.\left.\left\|\frac{1}{\alpha_n}\sum_{m=1}^{M} \pi_{n,m}\alpha_{n,m}\vec{e}_m - \vec{w}\right\|^2\right)\right]. \tag{70}$$

We suppose $r \geq 2$ and denote $\tau_n(0) = 0$, we get:

$$\mathbb{E}\big[(1-\alpha_n)^{2T \cdot \tau_n(r-1)}\big] = \mathbb{E}\Big[\mathbb{E}\big[(1-\alpha_n)^{2T\tau_n(r-1)}\big|\tau_n(r-2)\big]\Big] \tag{71}$$

$$=\mathbb{E}\big[(1-p_n)(1-\alpha_n)^{2T\tau_n(r-2)} + p_n(1-\alpha_n)^{2T(\tau_n(r-2)+1)}\big] \tag{72}$$

$$=\big(1 - p_n + p_n(1-\alpha_n)^{2T}\big)\mathbb{E}[(1-\alpha_n)^{2T\tau_n(r-2)}] \tag{73}$$

$$=\big(1 - p_n + p_n(1-\alpha_n)^{2T}\big)^{r-1}\mathbb{E}[(1-\alpha_n)^{2T\tau_n(0)}] \tag{74}$$

$$=\big(1 - p_n + p_n(1-\alpha_n)^{2T}\big)^{r-1}. \tag{75}$$

Finally, we conclude that

$$\min_r \mathbb{E}\|\nabla F(\bar{\mathbf{x}}^{(r)})\|^2 \leq \frac{1}{R}\sum_{r=1}^R \|\nabla F(\bar{\mathbf{x}}^{(r)})\|^2$$

$$\leq \frac{1}{(\frac{1}{2} - \frac{4c^2}{3(1-2c^2)})\sum_{n=1}^N p_n q_n}\Big[\frac{1}{T\eta R}F(\bar{\mathbf{x}}^{(1)}) + L\eta\sigma^2 \sum_{n=1}^N p_n^2 q_n + \frac{2cL\eta\sigma^2}{1-2c^2}\sum_{n=1}^N p_n q_n + \frac{1}{1-2c^2}\sum_{n=1}^N p_n q_n$$

$$\Big(\frac{\beta_n}{\alpha_n}(1-\gamma_n) + 2G\gamma_n \cdot \frac{\beta_n'}{1-(1-\alpha_n)^2} + 4G\gamma_n \cdot \frac{1}{R}\sum_{r=1}^R\big(1 - p_n + p_n(1-\alpha_n)^{2T}\big)^{r-1}$$

$$+ 2G\gamma_n \sum_{m=1}^M (\frac{1}{\alpha_n}\pi_{n,m}\alpha_{n,m} - w_m)^2\Big)\Big] \tag{76}$$

$$\leq \frac{1}{(\frac{1}{2} - \frac{4c^2}{3(1-2c^2)})\sum_{n=1}^N p_n q_n}\Big[\frac{1}{T\eta R}F(\bar{\mathbf{x}}^{(1)}) + L\eta\sigma^2 \sum_{n=1}^N p_n^2 q_n + \frac{2cL\eta\sigma^2}{1-2c^2}\sum_{n=1}^N p_n q_n + \frac{1}{1-2c^2}\sum_{n=1}^N p_n q_n$$

$$\Big(\frac{\beta_n}{\alpha_n}(1-\gamma_n) + 2G\gamma_n \cdot \frac{\beta_n'}{1-(1-\alpha_n)^2} + 4G\gamma_n \cdot \frac{1}{R} \cdot \frac{1}{p_n(1-(1-\alpha_n)^{2T})}$$

$$+ 2G\gamma_n \sum_{m=1}^M (\frac{1}{\alpha_n}\pi_{n,m}\alpha_{n,m} - w_m)^2\Big)\Big] \tag{77}$$

$$\leq \frac{18}{\sum_{n=1}^N p_n q_n}\Big[\frac{1}{T\eta R}F(\bar{\mathbf{x}}^{(1)}) + L\eta\sigma^2 \sum_{n=1}^N p_n^2 q_n + \frac{5L\eta\sigma^2}{3}\sum_{n=1}^N p_n q_n + \frac{5}{3}\sum_{n=1}^N p_n q_n$$

$$\Big(\frac{\beta_n}{\alpha_n}(1-\gamma_n) + 2G\gamma_n \cdot \frac{\beta_n'}{1-(1-\alpha_n)^2} + 4G\gamma_n \cdot \frac{1}{R} \cdot \frac{1}{p_n(1-(1-\alpha_n)^{2T})}$$

$$+ 2G\gamma_n \sum_{m=1}^M (\frac{1}{\alpha_n}\pi_{n,m}\alpha_{n,m} - w_m)^2\Big)\Big], \tag{78}$$

where

$$\alpha_n = \sum_{m=1}^M \pi_{n,m}\alpha_{n,m}, \quad \beta_n = \sum_{m=1}^M \pi_{n,m}\alpha_{n,m}d_m, \quad \gamma_n = \sum_{t=1}^T (1-\alpha_n)^t, \tag{79}$$

$$\beta_n' = 2\sum_{m=1}^M \pi_{n,m}\alpha_{n,m}^2 - \sum_{m=1}^M \pi_{n,m}^2\alpha_{n,m}^2 + \alpha_n^2, \tag{80}$$

(78) holds when $c \leq \min\{\frac{1}{2M}, \sqrt{\frac{1}{5}}\}$.

By choosing a sufficiently small learning rate $\eta = \frac{1}{\sqrt{R}}$, we derive the following convergence bound:

$$\min_r \mathbb{E}\|\nabla F(\bar{\mathbf{x}}^{(r)})\|^2 \leq \mathcal{O}(\frac{1}{\sqrt{R}}) + \mathcal{O}(\frac{1}{R}) + \epsilon, \tag{81}$$

where $R$ is the number of global rounds, and $\epsilon$ is a non-vanishing residual term induced by the distribution shift from streaming data, i.e., the objective inconsistency (Wang & Ji, 2022).

The $\mathcal{O}(1/\sqrt{R})$ term is consistent with the convergence rate of several federated learning baselines under smooth non-convex assumptions, such as FedAdam (Reddi et al., 2020), FLASH (Panchal et al., 2023), and FedDisco (Ye et al., 2023).

# F  UPPER BOUND MINIMIZATION

Generally, a tighter bound corresponds to a better result. Thus, we optimize the upper bound in (78) from two aspects: the client-specific data sampling strategy $\{\alpha_{n,m}\}_{n\in\mathcal{N},m\in\mathcal{M}}$ and the server-side aggregation algorithm $\{p_n\}_{n\in\mathcal{N}}$. Directly solving the minimization of the upper bound results in a complicated expression. To simplify the expression, we treat $\alpha_n$ as a fixed hyperparameter that governs the average update ratio of the client buffer, referred to as the *sampling budget*.

## F.1  DISTRIBUTION-GUIDED DATA SAMPLING STRATEGY

Our client-specific data sampling strategy is obtained through solving the following optimization problem:

$$\min_{\alpha_{n,m}} \frac{\beta_n}{\alpha_n}(1-\gamma_n) + 2G\gamma_n \cdot \frac{\beta_n'}{1-(1-\alpha_n)^2} + 2G\gamma_n \sum_{m=1}^{M}\left(\frac{1}{\alpha_n}\pi_{n,m}\alpha_{n,m} - w_m\right)^2,$$

$$\text{s.t. } \sum_{m=1}^{M}\pi_{n,m}\alpha_{n,m} = \alpha_n, \ \alpha_{n,m}\in[0,1]. \tag{82}$$

To solve this optimization problem, one condition of the optimal solution is that the derivative of the following function equals zero:

$$Q(\alpha_{n,m}) = \frac{\beta_n}{\alpha_n}(1-\gamma_n) + 2G\gamma_n \cdot \frac{\beta_n'}{1-(1-\alpha_n)^2} + 2G\gamma_n \sum_{m=1}^{M}\left(\frac{1}{\alpha_n}\pi_{n,m}\alpha_{n,m} - w_m\right)^2 \tag{83}$$

$$- \lambda_1\left(\sum_{m=1}^{M}\pi_{n,m}\alpha_{n,m} - \alpha_n\right) + \sum_{m=1}^{M}\mu_m\alpha_{n,m} - \nu_m\left(\sum_{m=1}^{M}\alpha_{n,m} - 1\right), \tag{84}$$

where $\lambda_1$, $\mu_m$ and $\nu_m$ are multipliers. Then, we have

$$\frac{1-\gamma_n}{\alpha_n}\pi_{n,m}d_m + \frac{2G\gamma_n}{1-(1-\alpha_n)^2}(4\pi_{n,m} - 2\pi_{n,m}^2)\alpha_{n,m} +$$

$$\frac{4G\gamma_n\pi_{n,m}}{\alpha_n}\left(\frac{1}{\alpha_n}\pi_{n,m}\alpha_{n,m} - w_m\right) - \lambda_1\pi_{n,m} + \mu_m - \nu_m = 0 \tag{85}$$

To derive the sampling ratio $\alpha_{n,m}$, we only consider the active states with $\pi_{n,m} > 0$. Therefore, we obtain

$$\alpha_{n,m} \propto \frac{\lambda\alpha_n + \frac{\nu_m - \mu_m}{\pi_{n,m}} + 4G\gamma_n w_m - (1-\gamma_n)d_m}{1 + \frac{1-\alpha_n}{\alpha_n}\pi_{n,m}} \tag{86}$$

To simplify the solution and derive a closed-form, interpretable expression, we relax the KKT conditions by eliminating the dual variables, setting $\mu_m = 0$ and $\nu_m = 0$, under the assumption that the optimal values of $\alpha_{n,m}$ lie strictly within the open interval $(0,1)$. This assumption is practically reasonable. In realistic streaming environments, it is neither desirable nor feasible for a client to completely discard previously stored data ($\alpha_{n,m} = 1$) or to entirely ignore new samples from a given state ($\alpha_{n,m} = 0$). Either extreme leads to inefficient storage usage and poor adaptability to evolving data distributions. Under this relaxed setting, we define the following score function:

$$\text{score}_n(m) = \frac{w_m - a_1 d_m + b_1}{1 + \frac{1-\alpha_n}{\alpha_n}\pi_{n,m}}, \tag{87}$$

where $a_1 = \frac{1-\gamma_n}{4G\gamma_n}$ and $b_1 = \frac{\lambda_1}{4G\gamma_n}$ are tunable constants. Intuitively, the score increases with the weight in the global objective function $w_m$, and decreases with the state-specific heterogeneity bound $d_m$ or the probability $\pi_{n,m}$ of client $n$ being in state $m$.

To ensure feasibility and numerical stability, we apply a ReLU operation to filter out negative scores, and normalize the resulting values with respect to the state distribution $\boldsymbol{\pi}_n$:

$$\alpha_{n,m} = \frac{\alpha_n \cdot \text{ReLu}(\text{score}_n(m))}{\sum_{m'=1}^{M}\pi_{n,m'} \cdot \text{ReLu}(\text{score}_n(m'))} \tag{88}$$

## F.2 AGGREGATION WEIGHT DETERMINATION

We now describe how to determine the client aggregation weights $\{p_n\}_{n=1}^N$ by minimizing the dominant terms in the convergence upper bound (78), discarding constant factors irrelevant to $p_n$. Specifically, we formulate the following constrained optimization:

$$\min_{p_n} \frac{1}{\sum_{n=1}^N p_n q_n} \Big[ L\eta\sigma^2 \sum_{n=1}^N p_n^2 q_n + \frac{5}{3} \sum_{n=1}^N p_n q_n \Big( L\eta\sigma^2 + \frac{\beta_n}{\alpha_n}(1-\gamma_n)$$

$$+ 2G\gamma_n \cdot \frac{\beta_n'}{1-(1-\alpha_n)^2} + 2G\gamma_n \sum_{m=1}^M \big( \frac{1}{\alpha_n} \pi_{n,m} \alpha_{n,m} - w_m \big)^2 \Big) \Big]$$

$$\text{s.t.} \sum_{n=1}^N p_n = 1, \; p_n \geq 0. \tag{89}$$

To simplify the expression, we denote

$$s_n = L\eta\sigma^2 + \frac{\beta_n}{\alpha_n}(1-\gamma_n) + 2G\gamma_n \cdot \frac{\beta_n'}{1-(1-\alpha_n)^2} + 2G\gamma_n \sum_{m=1}^M \big( \frac{1}{\alpha_n} \pi_{n,m} \alpha_{n,m} - w_m \big)^2 \tag{90}$$

$$T_0 = \sum_{n=1}^N p_n q_n, \; T_1 = L\eta\sigma^2 \sum_{n=1}^N p_n^2 q_n, \; T_2 = \frac{5}{3} \sum_{n=1}^N p_n q_n s_n. \tag{91}$$

We apply a standard surrogate objective (Ye et al., 2023), transforming from minimizing $(T_1+T_2)/T_0$ to minimizing $T_1 + T_2 - \lambda_2 T_0$, where $\lambda_2 > 0$ is a balancing hyper-parameter. Therefore, the optimization is

$$\min_{p_n} L\eta\sigma^2 \sum_{n=1}^N p_n^2 q_n + \frac{5}{3} \sum_{n=1}^N p_n q_n s_n - \lambda_2 \sum_{n=1}^N p_n q_n$$

$$\text{s.t.} \sum_{n=1}^N p_n = 1, \; p_n \geq 0, \tag{92}$$

Let $\nu$ and $\mu_n$ be the Lagrangian multipliers associated with the equality and inequality constraints, respectively. The Lagrangian is:

$$L(p_n) = L\eta\sigma^2 \sum_{n=1}^N p_n^2 q_n + \frac{5}{3} \sum_{n=1}^N p_n q_n s_n - \lambda_2 \sum_{n=1}^N p_n q_n + \sum_{n=1}^N \mu_n p_n - \nu \Big( \sum_{n=1}^N p_n - 1 \Big). \tag{93}$$

Taking the derivative of $L(\cdot)$ with respect to $p_n$ and setting it to zero yields:

$$2L\eta\sigma^2 p_n q_n + \frac{5}{3} q_n s_n - \lambda_2 q_n + \mu_n - \nu = 0 \tag{94}$$

Solving this, we obtain:

$$p_n = \frac{1}{2L\eta\sigma^2} \Big( \lambda - \frac{5}{3} s_n + \frac{\nu - \mu_n}{q_n} \Big). \tag{95}$$

If we relax the KKT multipliers and absorb them into constants $a_2$ and $b_2$, we arrive at the interpretable approximation:

$$p_n \propto \frac{1}{q_n} - a_2 \cdot s_n + b_2, \tag{96}$$

where $a_2$ and $b_2$ are tunable constants. Intuitively, $s_n$ is a metric to measure the heterogeneity of client-specific data distribution. It takes into account a client-specific data sampling strategy. Clients with smaller $s_n$ are assigned higher aggregation weights. The term $1/q_n$ ensures fairness with respect to availability.

### F.3 THE SFEDPO ALGORITHM

This section provides the complete pseudocode of our proposed SFedPO framework, as described in Section 5. Algorithm 1 outlines the end-to-end federated optimization process in the streaming setting with a prediction oracle and dynamic data sampling.

We observe that although Eq. (11) preserves the proportionality between sampling ratios and utility scores, it does not inherently ensure that $\hat{\alpha}_{n,m} \in [0,1]$. To resolve this, Algorithm 2 employs a budget-aware projection strategy: it iteratively clips any $\hat{\alpha}_{n,m}$ exceeding 1 and adjusts the remaining budget allocation across the other states. This iterative procedure guarantees that all sampling ratios remain feasible and that the total allocation strictly satisfies the sampling budget constraint $\alpha_n$.

---

**Algorithm 1:** `SFedPO`: Streaming Federated Learning with Prediction Oracle

---

**Require:** Initial global model $\bar{\mathbf{x}}^{(1)}$, learning rate $\eta$, number of rounds $R$, local steps $T$, prediction oracle $\mathcal{O}$, sampling budget $\alpha_n$

1: **for** each round $r = 1$ to $R$ **do**
2:     Server selects a client set $S_r$
3:     **for** each selected client $n \in S_r$ **in parallel do**
4:         $\hat{\boldsymbol{\pi}}_n \leftarrow \mathcal{O}(n, r)$                              *// predicted state distribution*
5:         Compute $\hat{\alpha}_{n,m}$ using Algorithm 2.
6:         Initialize $\mathbf{x}_{n,0}^{(r)} \leftarrow \bar{\mathbf{x}}^{(r)}$
7:         **for** each local step $t = 1$ to $T$ **do**
8:             Receive new data distribution $\{\mathcal{D}_m\}_{m=1}^M$ from streaming source.
9:             Update local dataset with a ratio of $\alpha_{n,m_{n,t}^{(r)}}$
10:            Sample minibatch $\xi_{n,t} \sim \mathcal{D}_{n,t}^{(r)}$, and train the model: $\mathbf{x}_{n,t}^{(r)} = \mathbf{x}_{n,t-1}^{(r)} - \eta \cdot \mathbf{g}_{n,t}^{(r)}$
11:         **end for**
12:         Send update $\Delta\mathbf{x}_n^{(r)} = \mathbf{x}_{n,E}^{(r)} - \bar{\mathbf{x}}^{(r)}$ to server
13:     **end for**
14:     Server computes aggregation weights $\hat{p}_n$ using:

$$\hat{p}_n = \frac{\text{ReLU}\left(\frac{1}{q_n} - a_2 \cdot \hat{s}_n + b_2\right)}{\sum_{j \in S_r} \text{ReLU}\left(\frac{1}{q_j} - a_2 \cdot \hat{s}_j + b_2\right)}$$

15:     Update global model:

$$\bar{\mathbf{x}}^{(r+1)} = \bar{\mathbf{x}}^{(r)} + \eta \sum_{n \in S_r} \hat{p}_n \cdot \Delta\mathbf{x}_n^{(r)}$$

16: **end for**

---

**Algorithm 2:** Computation of $\hat{\alpha}_{n,m}$

---

**Require:** Number of clients $N$ and states $M$, sampling budget $\alpha_n$, estimated state distribution $\hat{\pi}_{n,m}$, $\widehat{\text{score}}_n(m)$

1: Initialize remaining set $\mathcal{R} \leftarrow \{m \mid \hat{\pi}_{n,m} > 0\}$, residual budget $\tilde{\alpha}_n \leftarrow \alpha_n$
2: Set found_one $\leftarrow$ **True**
3: **while** found_one and $\mathcal{R} \neq \emptyset$ **do**
4:    found_one $\leftarrow$ **False**
5:    $S \leftarrow \sum_{m \in \mathcal{R}} \hat{\pi}_{n,m} \cdot \text{ReLU}(\widehat{\text{score}}_n(m))$
6:    **if** $S = 0$ **then**
7:      **for** each $m \in \mathcal{R}$ **do**
8:        $\hat{\alpha}_{n,m} \leftarrow \frac{\tilde{\alpha}_n}{\sum_{m \in \mathcal{R}} \hat{\pi}_{n,m}}$
9:      **end for**
10:     **break**
11:   **else**
12:      **for** each $m \in \mathcal{R}$ **do**
13:        $\hat{\alpha}_{n,m} \leftarrow \min\left(\frac{\tilde{\alpha}_n \cdot \text{ReLU}(\widehat{\text{score}}_n(m))}{S}, 1\right)$
14:        **if** $\hat{\alpha}_{n,m} = 1$ **then**
15:          $\tilde{\alpha}_n \leftarrow \tilde{\alpha}_n - \hat{\pi}_{n,m}$
16:          remove $m$ from $\mathcal{R}$, set found_one $\leftarrow$ **True**
17:          **break**
18:        **end if**
19:      **end for**
20:   **end if**
21: **end while**

---

# G    PROOF OF THEOREM 2

Based on the convergence analysis in Appendix E, we develop a theoretical strategy under the ground-truth state distribution $\boldsymbol{\pi}_n = (\pi_{n,1}, \ldots, \pi_{n,M})$ for each client $n \in \mathcal{N}$. In practical deployments, however, this is not directly observable and must be approximated using a prediction oracle, which provides an estimated state distribution $\hat{\boldsymbol{\pi}}_n = (\hat{\pi}_{n,1}, \ldots, \hat{\pi}_{n,M})$. This naturally raises the question:

*How does the prediction error affect the convergence behavior of the overall algorithm?*

To address this question, we conduct a robustness analysis that quantifies the impact of prediction error on the convergence guarantee derived in Appendix E. Specifically, we define the prediction error for client $n$ as $\delta_n := \|\hat{\boldsymbol{\pi}}_n - \boldsymbol{\pi}_n\|_1$, and study how this error propagates into the convergence bound through the data sampling strategy $\boldsymbol{\alpha}$ and aggregation weights $\{p_n\}_{n \in \mathcal{N}}$, both of which are functions of $\boldsymbol{\pi}_n$ in theory but implemented based on $\hat{\boldsymbol{\pi}}_n$ in practice.

Recall the convergence bound in Theorem 1, we focus on the following term:

$$\mathcal{B}(\alpha_{n,m}, p_n) := \frac{1}{\sum_{n=1}^{N} p_n q_n} \Big[ L\eta\sigma^2 \sum_{n=1}^{N} p_n^2 q_n + \frac{5}{3} \sum_{n=1}^{N} p_n q_n s_n \Big] \tag{97}$$

where

$$s_n = L\eta\sigma^2 + \frac{\beta_n}{\alpha_n}(1 - \gamma_n) + \frac{2G_n\gamma_n\beta_n'}{1 - (1 - \alpha_n)^2} + 2G\gamma_n \sum_{m=1}^{M} \Big( \frac{1}{\alpha_n}\pi_{n,m}\alpha_{n,m} - w_m \Big)^2. \tag{98}$$

In the realistic setting, these ratios are implemented as $\hat{\alpha}_{n,m}$ using the estimated distribution $\hat{\boldsymbol{\pi}}_n$. Following the relaxed design in Section 5, the sampling ratios are determined by:

$$\widehat{\text{score}}_n(m) = \frac{w_m - a_1 d_m + b_1}{1 + \frac{\alpha_n}{1 - \alpha_n}\hat{\pi}_{n,m}}, \text{ and } \hat{\alpha}_{n,m} = \frac{\text{ReLU}(\widehat{\text{score}}_n(m))}{\sum_{m'=1}^{M} \hat{\pi}_{n,m'} \cdot \text{ReLU}(\widehat{\text{score}}_n(m'))}. \tag{}$$

Our goal is to analyze the difference between the convergence bound under the theoretical strategy $(\alpha_{n,m}, p_n)$ and the practical strategy $(\hat{\alpha}_{n,m}, \hat{p}_n)$ computed using prediction oracle, and to establish a robustness bound in terms of the prediction error $\delta_n$.

## G.1    IMPACT ON SAMPLING STRATEGY.

We now analyze how the prediction error $\delta_n = \|\hat{\boldsymbol{\pi}}_n - \boldsymbol{\pi}_n\|_1$ affects the resulting data sampling strategy. Let $\boldsymbol{\alpha}_n = \{\alpha_{n,m}\}_{m=1}^{M}$ and $\hat{\boldsymbol{\alpha}}_n = \{\hat{\alpha}_{n,m}\}_{m=1}^{M}$ be the sampling ratios computed using the true and estimated state distributions, respectively. Our goal is to bound the $\ell_1$ deviation between the two vectors, i.e., $\|\hat{\boldsymbol{\alpha}}_n - \boldsymbol{\alpha}_n\|_1$.

Let $S_n := \sum_{j=1}^{M} \pi_{n,j} \cdot \text{ReLU}(\text{score}_n(j))$ and $\hat{S}_n := \sum_{j=1}^{M} \hat{\pi}_{n,j} \cdot \text{ReLU}(\widehat{\text{score}}_n(j))$. We suppose that the score function is $L_s$-Lipschitz in $\pi_{n,m}$, i.e.,

$$|\widehat{\text{score}}_n(m) - \text{score}_n(m)| \leq L_s |\hat{\pi}_{n,m} - \pi_{n,m}|. \tag{99}$$

We can derive the following bound:

$$\begin{aligned}
&|\hat{\alpha}_{n,m} - \alpha_{n,m}| \\
&\leq \left| \frac{\text{ReLU}(\widehat{\text{score}}_n(m))}{\hat{S}_n} - \frac{\text{ReLU}(\text{score}_n(m))}{\hat{S}_n} \right| + \left| \frac{\text{ReLU}(\text{score}_n(m))}{\hat{S}_n} - \frac{\text{ReLU}(\text{score}_n(m))}{S_n} \right| \\
&\leq \frac{L_s |\hat{\pi}_{n,m} - \pi_{n,m}|}{\hat{S}_n} + \text{ReLU}(\text{score}_n(m)) \cdot \left| \frac{1}{\hat{S}_n} - \frac{1}{S_n} \right|.
\end{aligned} \tag{100}$$

We further estimate the deviation between the $\hat{S}_n$ and $S_n$:

$$|\hat{S}_n - S_n| = \left| \sum_{j=1}^{M} \hat{\pi}_{n,j} \cdot \text{ReLU}(\widehat{\text{score}}_n(j)) - \sum_{j=1}^{M} \pi_{n,j} \cdot \text{ReLU}(\text{score}_n(j)) \right|$$

$$\leq \sum_{j=1}^{M} \left| \hat{\pi}_{n,j} \cdot \text{ReLU}(\widehat{\text{score}}_n(j)) - \pi_{n,j} \cdot \text{ReLU}(\text{score}_n(j)) \right|$$

$$\leq \sum_{j=1}^{M} \left( |\hat{\pi}_{n,j} - \pi_{n,j}| \cdot \text{ReLU}(\text{score}_n(j)) + \hat{\pi}_{n,j} \cdot |\text{ReLU}(\widehat{\text{score}}_n(j)) - \text{ReLU}(\text{score}_n(j))| \right)$$

$$\leq \delta_n \cdot \|\text{ReLU}(\text{score}_n)\|_\infty + L_s \cdot \delta_n. \tag{101}$$

Thus, we obtain

$$\left| \frac{1}{\hat{S}_n} - \frac{1}{S_n} \right| = \frac{|\hat{S}_n - S_n|}{\hat{S}_n \cdot S_n} \leq \frac{\delta_n \cdot (L_s + \|\text{ReLU}(\text{score}_n)\|_\infty)}{\hat{S}_n \cdot S_n}, \tag{102}$$

and hence,

$$|\hat{\alpha}_{n,m} - \alpha_{n,m}| \leq \frac{L_s |\hat{\pi}_{n,m} - \pi_{n,m}|}{\hat{S}_n} + \text{ReLU}(\text{score}_n(m)) \cdot \frac{\delta_n \cdot (L_s + \|\text{ReLU}(\text{score}_n)\|_\infty)}{\hat{S}_n \cdot S_n}. \tag{103}$$

Summing over all $m \in \mathcal{M}$, and using the fact that ReLU scores are bounded and $S_n, \hat{S}_n \geq \epsilon$ for some small constant, we obtain:

$$\|\hat{\boldsymbol{\alpha}}_n - \boldsymbol{\alpha}_n\|_1 \leq C_1 \cdot \delta_n, \tag{104}$$

where $C_1$ depends on the Lipschitz constant $L_s$, the upper bound of the ReLU scores, and the lower bounds of the $S_n$ and $\hat{S}_n$.

### G.2 Impact on Aggregation Weights.

Recall the client-specific heterogeneity score:

$$s_n = L\eta\sigma^2 + \frac{\beta_n}{\alpha_n}(1 - \gamma_n) + \frac{2G\gamma_n\beta'_n}{1 - (1 - \alpha_n)^2} + 2G\gamma_n \sum_{m=1}^{M} \left( \frac{1}{\alpha_n}\pi_{n,m}\alpha_{n,m} - w_m \right)^2.$$

The first term is a constant and unaffected by prediction error. We now analyze the impact of prediction error $\delta_n := \|\hat{\boldsymbol{\pi}}_n - \boldsymbol{\pi}_n\|_1$ on the remaining three terms.

**Bounding** $\frac{\beta_n}{\alpha_n}(1 - \gamma_n)$. Let

$$\beta_n = \sum_{m=1}^{M} \pi_{n,m}\alpha_{n,m}d_m, \quad \hat{\beta}_n = \sum_{m=1}^{M} \hat{\pi}_{n,m}\hat{\alpha}_{n,m}d_m.$$

Then, using triangle inequality, we have

$$|\hat{\beta}_n - \beta_n| \leq \sum_{m=1}^{M} |\hat{\pi}_{n,m}\hat{\alpha}_{n,m} - \pi_{n,m}\alpha_{n,m}| \cdot |d_m|$$

$$\leq \sum_{m=1}^{M} \left( |\hat{\pi}_{n,m} - \pi_{n,m}| \cdot |\hat{\alpha}_{n,m}| + \pi_{n,m} \cdot |\hat{\alpha}_{n,m} - \alpha_{n,m}| \right) d_m.$$

$$\leq D_{\max} \cdot \left( \|\hat{\boldsymbol{\alpha}}_n - \boldsymbol{\alpha}_n\|_1 + \delta_n \right), \tag{105}$$

where $D_{\max} := \max_m d_m$. This gives:

$$\left| \frac{\hat{\beta}_n}{\alpha_n}(1 - \gamma_n) - \frac{\beta_n}{\alpha_n}(1 - \gamma_n) \right| \leq \frac{(1 - \gamma_n)D_{\max}}{\alpha_n} \cdot (C_1 + 1) \cdot \delta_n \quad =: C_2\delta_n. \tag{106}$$

**Bounding $\frac{2G\gamma_n\beta'_n}{1-(1-\alpha_n)^2}$.** Recall

$$\beta'_n = 2\sum_{m=1}^{M}\pi_{n,m}\alpha_{n,m}^2 - \sum_{m=1}^{M}\pi_{n,m}^2\alpha_{n,m}^2 + \alpha_n^2.$$

Let $\hat{\beta}'_n$ be the corresponding quantity computed under $\hat{\pi}_{n,m}$ and $\hat{\alpha}_{n,m}$:

$$\hat{\beta}'_n = 2\sum_{m=1}^{M}\hat{\pi}_{n,m}\hat{\alpha}_{n,m}^2 - \sum_{m=1}^{M}\hat{\pi}_{n,m}^2\hat{\alpha}_{n,m}^2 + \alpha_n^2.$$

Note that $\alpha_n$ is assumed to be fixed, so it cancels in the subtraction. Then we have:

$$|\hat{\beta}'_n - \beta'_n| \le 2\sum_{m=1}^{M}|\hat{\pi}_{n,m}\hat{\alpha}_{n,m}^2 - \pi_{n,m}\alpha_{n,m}^2| + \sum_{m=1}^{M}|\hat{\pi}_{n,m}^2\hat{\alpha}_{n,m}^2 - \pi_{n,m}^2\alpha_{n,m}^2|. \tag{107}$$

We now bound each term separately. First, observe that

$$\left|\hat{\pi}_{n,m}\hat{\alpha}_{n,m}^2 - \pi_{n,m}\alpha_{n,m}^2\right| \le \left|\hat{\pi}_{n,m}(\hat{\alpha}_{n,m}^2 - \alpha_{n,m}^2)\right| + \left|(\hat{\pi}_{n,m} - \pi_{n,m})\alpha_{n,m}^2\right|$$
$$\le \hat{\pi}_{n,m}\cdot 2\alpha_{\max}|\hat{\alpha}_{n,m} - \alpha_{n,m}| + \alpha_{\max}^2|\hat{\pi}_{n,m} - \pi_{n,m}|, \tag{108}$$

where we assume that $\alpha_{n,m}, \hat{\alpha}_{n,m} \le \alpha_{\max} \le 1$.

Similarly, for the second term:

$$\left|\hat{\pi}_{n,m}^2\hat{\alpha}_{n,m}^2 - \pi_{n,m}^2\alpha_{n,m}^2\right| \le \left|\hat{\pi}_{n,m}^2(\hat{\alpha}_{n,m}^2 - \alpha_{n,m}^2)\right| + \left|(\hat{\pi}_{n,m}^2 - \pi_{n,m}^2)\alpha_{n,m}^2\right|$$
$$\le \hat{\pi}_{n,m}^2\cdot 2\alpha_{\max}|\hat{\alpha}_{n,m} - \alpha_{n,m}| + 2\pi_{\max}\alpha_{\max}^2|\hat{\pi}_{n,m} - \pi_{n,m}|, \tag{109}$$

where we assume that $\pi_{n,m}, \hat{\pi}_{n,m} \le \pi_{\max} \le 1$.

Summing over all $m$, and applying the triangle inequality:

$$|\hat{\beta}'_n - \beta'_n| \le 4\pi_{\max}\alpha_{\max}\|\hat{\boldsymbol{\alpha}}_n - \boldsymbol{\alpha}_n\|_1 + 2\alpha_{\max}^2\|\hat{\boldsymbol{\pi}}_n - \boldsymbol{\pi}_n\|_1 \tag{110}$$
$$+ 2\pi_{\max}^2\alpha_{\max}\|\hat{\boldsymbol{\alpha}}_n - \boldsymbol{\alpha}_n\|_1 + 2\pi_{\max}\alpha_{\max}^2\|\hat{\boldsymbol{\pi}}_n - \boldsymbol{\pi}_n\|_1$$
$$\le (4\pi_{\max}\alpha_{\max} + 2\pi_{\max}^2\alpha_{\max})C_1\delta_n + (2\alpha_{\max}^2 + 2\pi_{\max}\alpha_{\max}^2)\delta_n. \tag{111}$$

Hence, we conclude:

$$\left|\frac{2G\gamma_n\hat{\beta}'_n}{1-(1-\alpha_n)^2} - \frac{2G\gamma_n\beta'_n}{1-(1-\alpha_n)^2}\right|$$
$$\le \frac{2G\gamma_n}{1-(1-\alpha_n)^2}[(4\pi_{\max}\alpha_{\max} + 2\pi_{\max}^2\alpha_{\max})C_1 + (2\alpha_{\max}^2 + 2\pi_{\max}\alpha_{\max}^2)]\cdot\delta_n \quad =: C_3\cdot\delta_n. \tag{112}$$

**Bounding $2G\gamma_n\sum_{m=1}^{M}\left(\frac{1}{\alpha_n}\pi_{n,m}\alpha_{n,m} - w_m\right)^2$.** Let

$$z_m = \frac{1}{\alpha_n}\pi_{n,m}\alpha_{n,m}, \quad \hat{z}_m = \frac{1}{\alpha_n}\hat{\pi}_{n,m}\hat{\alpha}_{n,m}.$$

Then, we have

$$|\hat{z}_m - z_m| = \left|\frac{1}{\alpha_n}(\hat{\pi}_{n,m}\hat{\alpha}_{n,m} - \pi_{n,m}\alpha_{n,m})\right| \le \frac{1}{\alpha_n}(|\hat{\pi}_{n,m} - \pi_{n,m}|\cdot\hat{\alpha}_{n,m} + \pi_{n,m}\cdot|\hat{\alpha}_{n,m} - \alpha_{n,m}|)$$
$$\le \frac{1}{\alpha_n}(\alpha_{\max}|\hat{\pi}_{n,m} - \pi_{n,m}| + \pi_{\max}|\hat{\alpha}_{n,m} - \alpha_{n,m}|). \tag{113}$$

Summing over all $m$, and applying the triangle inequality:

$$\left|\sum_{m=1}^{M}(\hat{z}_m - w_m)^2 - \sum_{m=1}^{M}(z_m - w_m)^2\right| \le \sum_{m=1}^{M}|\hat{z}_m^2 - z_m^2| + 2\sum_{m=1}^{M}|w_m||\hat{z}_m - z_m|$$
$$\le \sum_{m=1}^{M}|\hat{z}_m - z_m|\cdot|\hat{z}_m + z_m| + 2\sum_{m=1}^{M}|w_m|\cdot|\hat{z}_m - z_m|$$
$$\le \sum_{m=1}^{M}\frac{4}{\alpha_n}(\alpha_{\max}|\hat{\pi}_{n,m} - \pi_{n,m}| + \pi_{\max}|\hat{\alpha}_{n,m} - \alpha_{n,m}|)$$
$$\le \frac{4(\alpha_{\max} + C_1\pi_{\max})}{\alpha_n}\cdot\delta_n, \tag{114}$$

where we use the fact that $\hat{z}_m, z_m \in [0, 1]$ and $w_m \leq 1$, for all $m$.

Therefore, we have:

$$\left| 2G\gamma_n \sum_{m=1}^{M} (\hat{z}_m - w_m)^2 - 2G\gamma_n \sum_{m=1}^{M} (z_m - w_m)^2 \right| \leq 2G\gamma_n \cdot \frac{4(\alpha_{\max} + C_1\pi_{\max})}{\alpha_n} \cdot \delta_n \quad =: C_4 \cdot \delta_n. \tag{115}$$

Combining bounds (106), (112), and (115), we obtain the total deviation of $s_n$:

$$|\hat{s}_n - s_n| \leq (C_2 + C_3 + C_4) \cdot \delta_n \quad =: C_s \cdot \delta_n, \tag{116}$$

We now analyze how the prediction error $\delta_n$ affects the final aggregation weights. Given the aggregation weights:

$$p_n = \frac{\text{ReLU}(\frac{1}{q_n} - a_2 \cdot s_n + b_2)}{\sum_{i=1}^{N} \text{ReLU}(\frac{1}{q_i} - a_2 \cdot s_i + b_2)}, \quad \hat{p}_n = \frac{\text{ReLU}(\frac{1}{q_n} - a_2 \cdot \hat{s}_n + b_2)}{\sum_{i=1}^{N} \text{ReLU}(\frac{1}{q_i} - a_2 \cdot \hat{s}_i + b_2)}.$$

Let us denote:

$$\psi_n := \text{ReLU}(\tfrac{1}{q_n} - a_2 s_n + b_2), \quad \hat{\psi}_n := \text{ReLU}(\tfrac{1}{q_n} - a_2 \hat{s}_n + b_2),$$

$$Z := \sum_{i=1}^{N} \psi_i, \quad \hat{Z} := \sum_{i=1}^{N} \hat{\psi}_i.$$

Then, the deviation between $\hat{p}_n$ and $p_n$ can be bounded by:

$$|\hat{p}_n - p_n| = \left| \frac{\hat{\psi}_n}{\hat{Z}} - \frac{\psi_n}{Z} \right|$$

$$\leq \left| \frac{\hat{\psi}_n - \psi_n}{\hat{Z}} \right| + \left| \psi_n \cdot \left( \frac{1}{\hat{Z}} - \frac{1}{Z} \right) \right|$$

$$\leq \frac{a_2 |\hat{s}_n - s_n|}{\hat{Z}} + \psi_n \cdot \frac{|Z - \hat{Z}|}{Z\hat{Z}}, \tag{117}$$

where we used the fact that ReLU is 1-Lipschitz.

Note that $|Z - \hat{Z}| \leq \sum_{i=1}^{N} |\psi_i - \hat{\psi}_i| \leq a_2 \sum_{i=1}^{N} |s_i - \hat{s}_i| \leq a_2 C_s \cdot \sum_{i=1}^{N} \delta_i$.

Thus,

$$|\hat{p}_n - p_n| \leq \frac{a_2 C_s \delta_n}{\hat{Z}} + \frac{\psi_n \cdot a_2 C_s \sum_{i=1}^{N} \delta_i}{Z\hat{Z}}, \tag{118}$$

where $C_s$ is a bound such that $|s_n - \hat{s}_n| \leq C_s\delta_n$ as shown in (116).

Therefore, under mild conditions on $\psi_n \geq \epsilon$ and $Z, \hat{Z} \geq N\epsilon$ for some $\epsilon > 0$, we conclude:

$$|\hat{p}_n - p_n| \leq C_5 \cdot \delta_n + C_6 \cdot \sum_{i=1}^{N} \delta_i, \tag{119}$$

for constants $C_5, C_6$ depending on $a_2$, $C_s$, and bounds on $\psi_n$ and $Z$.

### G.3 IMPACT ON CONVERGENCE BOUND.

We now analyze how the prediction error $\delta_n = \|\hat{\pi}_n - \pi_n\|_1$ propagates to the convergence upper bound through sampling strategy $\{\alpha_{n,m}\}$ and aggregation weights $\{\hat{p}_n\}$.

Recall

$$\mathcal{B}(\alpha_{n,m}, p_n) = \frac{1}{\sum_{n=1}^{N} p_n q_n} \left[ L\eta\sigma^2 \sum_{n=1}^{N} p_n^2 q_n + \frac{5}{3} \sum_{n=1}^{N} p_n q_n s_n \right],$$

$$\mathcal{B}(\hat{\alpha}_{n,m}, \hat{p}_n) = \frac{1}{\sum_{n=1}^{N} \hat{p}_n q_n} \left[ L\eta\sigma^2 \sum_{n=1}^{N} \hat{p}_n^2 q_n + \frac{5}{3} \sum_{n=1}^{N} \hat{p}_n q_n \tilde{s}_n \right],$$

where

$$\tilde{s}_n = L\eta\sigma^2 + \frac{\tilde{\beta}_n}{\alpha_n}(1 - \gamma_n) + \frac{2G\gamma_n\tilde{\beta}'_n}{1 - (1 - \alpha_n)^2} + 2G\gamma_n \sum_{m=1}^{M} \left(\frac{1}{\alpha_n}\pi_{n,m}\alpha_{n,m} - w_m\right)^2, \quad (120)$$

$$\tilde{\beta}_n = \sum_{m=1}^{M} \pi_{n,m}\hat{\alpha}_{n,m}d_m \quad (121)$$

$$\tilde{\beta}'_n = 2\sum_{m=1}^{M} \pi_{n,m}\hat{\alpha}_{n,m}^2 - \sum_{m=1}^{M} \pi_{n,m}^2\hat{\alpha}_{n,m}^2 + \alpha_n^2. \quad (122)$$

Then, the deviation can be bounded as:

$$|\mathcal{B}(\hat{\alpha}_{n,m}, \hat{p}_n) - \mathcal{B}(\alpha_{n,m}, p_n)| \leq \left|\frac{1}{\sum_{n=1}^{N} \hat{p}_n q_n} - \frac{1}{\sum_{n=1}^{N} p_n q_n}\right| \cdot \mathcal{B}_{\max} + \frac{1}{\sum_{n=1}^{N} p_n q_n} \cdot \Delta_{\text{num}}, \quad (123)$$

where $\mathcal{B}_{\max}$ denotes a uniform upper bound on the numerator, and

$$\Delta_{\text{num}} = L\eta\sigma^2 \cdot \left|\sum_{n=1}^{N} (\hat{p}_n^2 - p_n^2)q_n\right| + \frac{5}{3} \cdot \left|\sum_{n=1}^{N} \hat{p}_n q_n \tilde{s}_n - \sum_{n=1}^{N} p_n q_n s_n\right|. \quad (124)$$

We now bound each part of $\Delta_{\text{num}}$:

**Bounding** $\sum_{n=1}^{N} (\hat{p}_n^2 - p_n^2)q_n$. Let $\delta_n^p := \hat{p}_n - p_n$. Then

$$\left|\sum_{n=1}^{N} (\hat{p}_n^2 - p_n^2)q_n\right| = \left|\sum_{n=1}^{N} \left[(p_n + \delta_n^p)^2 - p_n^2\right]q_n\right| = \left|\sum_{n=1}^{N} \left[2p_n\delta_n^p + (\delta_n^p)^2\right]q_n\right|$$

$$\leq 2\sum_{n=1}^{N} p_n q_n|\delta_n^p| + \sum_{n=1}^{N} q_n(\delta_n^p)^2$$

$$\leq 2C_5 \sum_{n=1}^{N} p_n q_n \delta_n + 2C_6 \left(\sum_{n=1}^{N} p_n q_n\right) \cdot \left(\sum_{n=1}^{N} \delta_n\right) + \mathcal{O}(\delta_n^2). \quad (125)$$

**Bounding** $\sum_{n=1}^{N} \hat{p}_n q_n \tilde{s}_n - \sum_{n=1}^{N} p_n q_n s_n$. We fist bound $|\tilde{s}_n - s_n|$. Similar to calculating the bound of $|\hat{s}_n - s_n|$, we have

$$|\tilde{\beta}_n - \beta_n| \leq \sum_{m=1}^{M} \pi_{n,m}|\hat{\alpha}_{n,m} - \alpha_{n,m}| \cdot |d_m| \leq D_{\max} \cdot \|\hat{\boldsymbol{\alpha}}_n - \boldsymbol{\alpha}_n\|_1, \quad (126)$$

$$|\tilde{\beta}'_n - \beta'_n| \leq 2\sum_{m=1}^{M} \pi_{n,m}|\hat{\alpha}_{n,m}^2 - \alpha_{n,m}^2| + \sum_{m=1}^{M} \pi_{n,m}^2|\hat{\alpha}_{n,m}^2 - \alpha_{n,m}^2|$$

$$\leq (4\pi_{\max}\alpha_{\max} + 2\pi_{\max}^2\alpha_{\max})\|\hat{\boldsymbol{\alpha}}_n - \boldsymbol{\alpha}_n\|_1. \quad (127)$$

Therefore, there exists a const $\tilde{C}_s$ such that

$$|\tilde{s}_n - s_n| \leq \tilde{C}_s \cdot \delta_n. \quad (128)$$

We decompose the difference:

$$\left|\sum_{n=1}^{N} \hat{p}_n q_n \tilde{s}_n - \sum_{n=1}^{N} p_n q_n s_n\right| = \left|\sum_{n=1}^{N} (\hat{p}_n - p_n)q_n s_n + \sum_{n=1}^{N} \hat{p}_n q_n(\tilde{s}_n - s_n)\right|$$

$$\leq \sum_{n=1}^{N} |\delta_n^p| q_n s_n + \sum_{n=1}^{N} \hat{p}_n q_n \tilde{C}_s \delta_n \quad (129)$$

$$\leq \sum_{n=1}^{N} (C_5 \cdot \delta_n + C_6 \cdot \sum_{i=1}^{N} \delta_i)q_n s_n + \sum_{n=1}^{N} \hat{p}_n q_n \tilde{C}_s \delta_n. \quad (130)$$

**Bounding the Denominator Term.**    We now analyze the denominator difference:

$$\left| \frac{1}{\sum_{n=1}^{N} \hat{p}_n q_n} - \frac{1}{\sum_{n=1}^{N} p_n q_n} \right| = \left| \frac{Y - \hat{Y}}{Y \cdot \hat{Y}} \right|, \quad \text{where} \quad Y := \sum_{n=1}^{N} p_n q_n, \ \hat{Y} := \sum_{n=1}^{N} \hat{p}_n q_n.$$

Assuming $Y, \hat{Y} \geq \epsilon > 0$, we have

$$\left| \frac{1}{\hat{Y}} - \frac{1}{Y} \right| \leq \frac{1}{\epsilon^2} \cdot |Y - \hat{Y}| = \frac{1}{\epsilon^2} \cdot \left| \sum_{n=1}^{N} (p_n - \hat{p}_n) q_n \right|. \tag{131}$$

Using the previously derived bound $|\hat{p}_n - p_n| \leq C_5 \cdot \delta_n + C_6 \cdot \sum_{i=1}^{N} \delta_i$, we obtain

$$|Y - \hat{Y}| \leq \sum_{n=1}^{N} |p_n - \hat{p}_n| \cdot q_n \leq \sum_{n=1}^{N} (C_5 \cdot \delta_n + C_6 \cdot \sum_{i=1}^{N} \delta_i) q_n$$

$$= C_5 \sum_{n=1}^{N} q_n \delta_n + C_6 \left( \sum_{i=1}^{N} \delta_i \right) \cdot \sum_{n=1}^{N} q_n. \tag{132}$$

Hence,

$$\left| \frac{1}{\hat{Y}} - \frac{1}{Y} \right| \leq \frac{C_5 \cdot \|q\|_1 + C_6 \cdot \|q\|_1}{\epsilon^2} \cdot \sum_{n=1}^{N} \delta_n, \tag{133}$$

Finally, substituting (125), (130), and (133) into (123), we obtain:

$$|\mathcal{B}(\hat{\alpha}_{n,m}, \hat{p}_n) - \mathcal{B}(\alpha_{n,m}, p_n)| \leq \left( \frac{\mathcal{B}_{\max}(C_5 + C_6)\|q\|_1}{\epsilon^2} \right) \cdot \sum_{n=1}^{N} \delta_n$$

$$+ \frac{1}{\epsilon} \Big[ L\eta\sigma^2 \Big( 2C_5 \sum_{n=1}^{N} p_n q_n \delta_n + 2C_6 \sum_{n=1}^{N} p_n q_n \cdot \sum_{n=1}^{N} \delta_n \Big)$$

$$+ \frac{5}{3} \Big( \sum_{n=1}^{N} C_5 \delta_n q_n s_n + C_6 \sum_{i=1}^{N} \delta_i \sum_{n=1}^{N} q_n s_n + \sum_{n=1}^{N} \hat{p}_n q_n \tilde{C}_s \delta_n \Big) \Big] + \mathcal{O}(\delta_n^2)$$

$$\leq \mathcal{O}\Big( \sum_{n=1}^{N} \delta_n \Big). \tag{134}$$

# H MORE EXPERIMENTAL DETAILS

## H.1 EXPERIMENTAL SETUP

**Datasets and models.** We conduct comprehensive experiments on four public benchmark datasets, including Fashion-MNIST (Xiao et al., 2017), CIFAR-10 (Krizhevsky et al., 2009), CINIC-10 (Darlow et al., 2018), and HAM10000 (Tschandl et al., 2018). Fashion-MNIST comprises $28 \times 28$ grayscale images of 70,000 fashion products with 10 classes, and there are 60,000 training images and 10,000 testing images. The CIFAR-10 dataset consists of 50,000 training images and 10,000 testing images, each with a size of $3 \times 32 \times 32$. CINIC-10 is an extension of CIFAR-10 via the addition of downsampled ImageNet (Deng et al., 2009) images. The HAM10000 dataset is an image dataset used for skin lesion classification in the medical field. For Fashion-MNIST, We use a LeNet-5 (LeCun et al., 1998) with two 5×5 convolutional layers, each followed by ReLU activation and max pooling, and three fully connected layers. For CIFAR-10 and CINIC-10, we use an 8-layer AlexNet (Krizhevsky et al., 2012) with a size of 136 MB. For HAM10000, we use a customized CNN consisting of three 3 × 3 convolutional layers (with ReLU activation and max pooling) and two fully connected layers with dropout regularization.

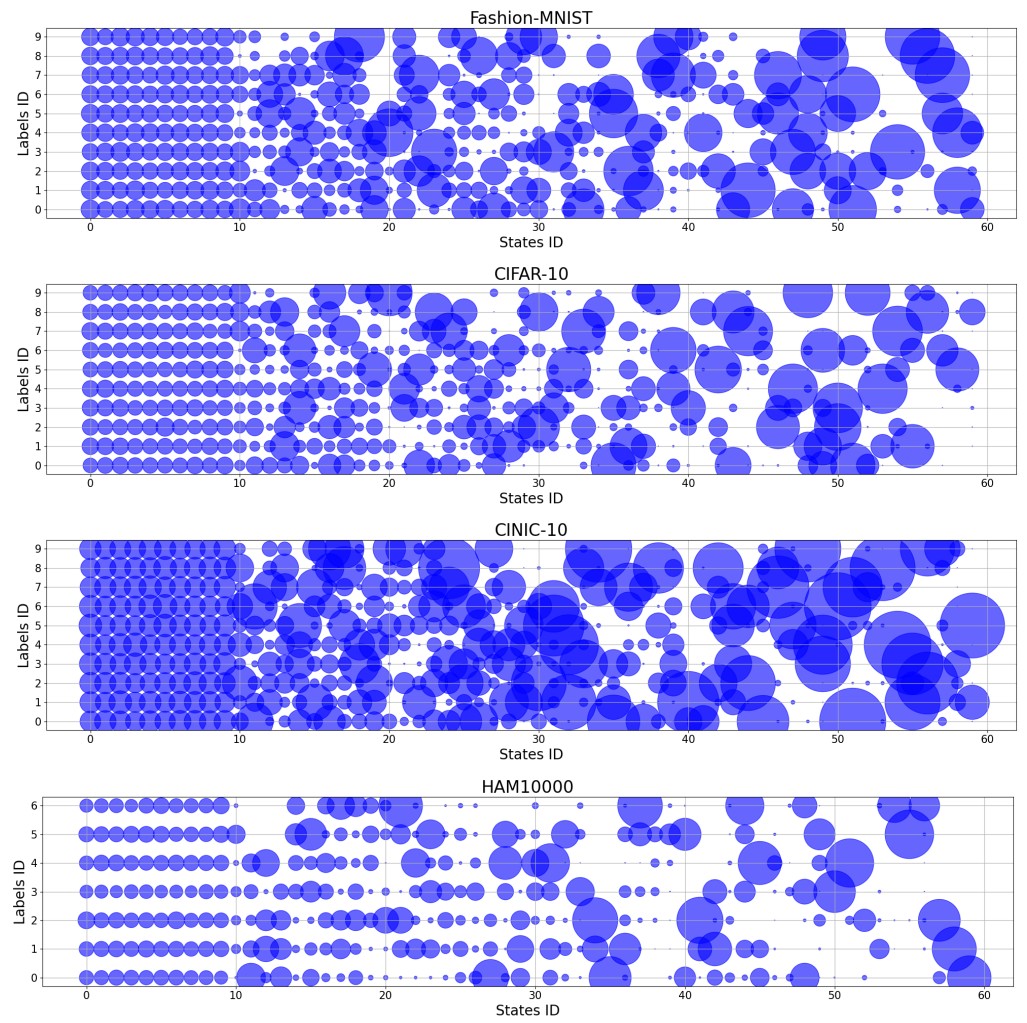

Figure 4: Visualization of the class distribution under stratified state space on the four datasets, where the bubble size represents the number of class samples per state.

**Intra-state heterogeneity.** We organize the 60 latent states into 6 clusters (10 states per cluster), each associated with a Dirichlet partitioning strategy (Wang et al., 2020a) with distinct concentration

parameters $\alpha \in \{0.05, 0.1, 0.2, 0.5, 1.0, 100.0\}$. We visualize the class distribution under stratified state space on the four datasets in Fig. 4.

**Two heterogeneity scenarios.** We consider two heterogeneity scenarios. (1) Full-access (mild heterogeneity): Each client has non-zero access probability $\pi_{n,m} > 0$ for all states $m \in \{1, \ldots, M\}$. (2) Partial-access (extreme heterogeneity): Each client is restricted to a randomly sampled subset of 10 states, with $\pi_{n,m} = 0$ for others. Moreover, 50% of clients are initialized with latent states drawn from high-heterogeneity clusters ($\alpha \in \{0.05, 0.1\}$). We visualize the state probability $\pi_{n,m}$ of all clients under two scenarios in Fig. 5.

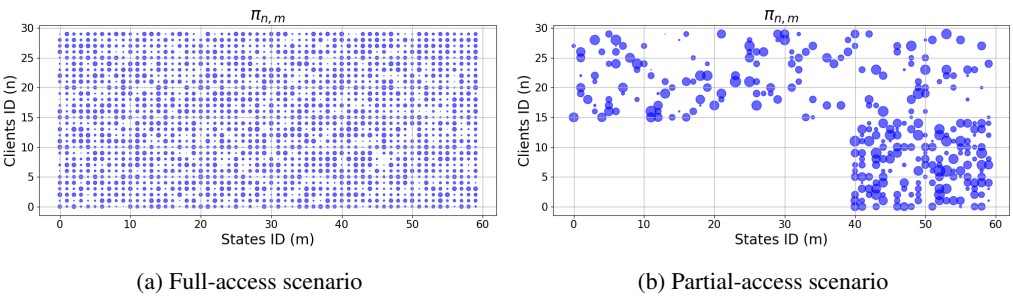

(a) Full-access scenario          (b) Partial-access scenario

Figure 5: Visualization of the state probability of all clients under two scenarios, where the bubble size reflects the probability of a client reaching that state.

**Implementation details.** The overall framework of SFedPO is implemented with Pytorch (Paszke et al., 2019), and all experiments are conducted on an Intel(R) Xeon(R) Platinum 8352V CPU and an NVIDIA A40 (48GB) GPU, and 256GB RAM. We run federated learning for 100 rounds. The number of time steps and batch size are 5 and 64, respectively. We use an SGD optimizer with a 0.01 learning rate, and the weight decay is set to 1e-4. For each client $n$, we set its sampling budget $\alpha_n$ to 0.5 and allocate a data capacity $D$ of 500 samples. While our theoretical framework defines $d_m$ as the upper bound on the gradient variance under state $m$, estimating such variance is often impractical in streaming scenarios with limited and evolving local data. Inspired by (Ye et al., 2023), we adopt a practical surrogate by assuming that $d_m$ is proportional to the discrepancy between the class distribution and a uniform distribution. Accordingly, we use the KL divergence as a proxy measure for $d_m$ in all experiments.

**Client-specific sampling ratios.** We visualize the client-specific sampling ratios $\{\alpha_{n,m}\}_{m=1}^{M}$ under two scenarios in Fig. 6. We observe that the client-specific sampling ratios exhibit a clustering pattern aligned with the clustered structure of the state space. Moreover, as the distribution of the state space becomes more imbalanced, the client-specific sampling ratios tend to decrease accordingly.

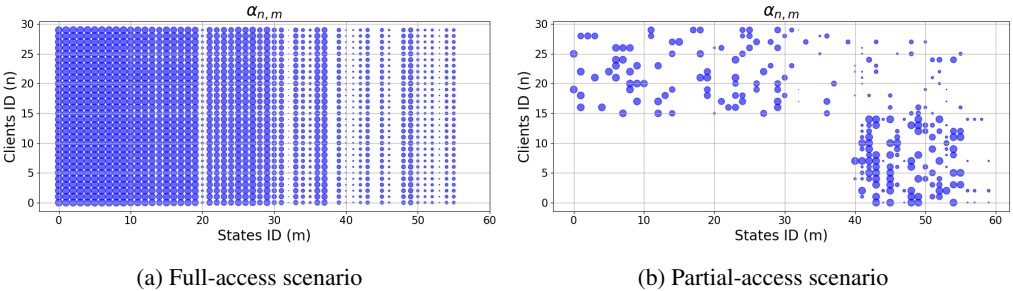

(a) Full-access scenario          (b) Partial-access scenario

Figure 6: Visualization of the client-specific sampling ratios of all clients under two scenarios, where the bubble size reflects the value of sampling ratios.

**Baselines.** We compare SFedPO with the following methods:

- **FedOGD:** FedOGD is a vanilla online federated Learning method that processes sequential data and performs online optimization per round.
- **FedOMD:** FedOMD is an online federated Learning method, which performs online mirror descent and achieves sublinear regret.

- **Adaptive-FedAvg:** Adaptive-FedAvg adjusts local learning rates to improve model plasticity under concept drift in federated Learning.

- **FedDrift:** FedDrift uses a hierarchical clustering method to address the problem of concept drift adaptation in federated Learning.

- **Flash:** Flash synergizes client-side early-stopping to detect concept drifts with server-side drift-aware adaptive optimization to effectively adjust the global learning rate.

- **FedEWC:** EWC is a classic regularization-based method for continual learning that uses the Fisher information matrix to estimate the importance of parameters. In our streaming scenario, each client leverages EWC to retain information from the previous state.

- **FLwF-2T:** FLwF-2T is a distillation-based method that deals with catastrophic forgetting in federated continual learning. FLwF-2T enables the current model to distill knowledge from both the model trained in the previous state and the global model from the last round.

- **IS:** Important Sampling (IS) uses a higher gradient norm to reflect the informativeness of the data. In our streaming scenario, it selects the optimal data from the new state based on the metric and discards the old data accordingly.

- **ODE:** ODE introduces an online data selection method based on a data valuation metric: the projection of local gradients onto the global gradient. In our streaming scenario, it selects the optimal data from the new state based on the metric and discards the old data accordingly.

- **DRSR:** DRSR updates the local data of each client according to the distribution discrepancy between the long-term data distribution and the client's local data. In our streaming scenario, it selects informative samples from both the existing and incoming data based on the update rule.

Since some methods (i.e., FedOGD, FedOMD, Adaptive-FedAvg, FedDrift, Flash, FedEWC, FLwF-2T) do not incorporate any data sampling mechanism, we assume that each client fully replaces its local dataset with new data drawn from the current state distribution, thereby aligning with the streaming FL setting.

**FL methods in modularity experiment.** We apply SFedPO's data sampling and aggregation strategies to several representative FL methods: **FedAvg** (McMahan et al., 2017), which is the pioneering FL method; **FedProx** (Li et al., 2020a), a classic regularization-based FL method; **FedCurv** (Shoham et al., 2019), a FL method based on curvature adjustment regularization; **FedNTD** (Lee et al., 2022), which preserves the global knowledge by not-true distillation in FL; **FedEXP** (Jhunjhunwala et al., 2023), which tune the global learning rate via extrapolation to speed up the global convergence. In our streaming data scenario, these methods sample data based on state transitions in each round and train local models accordingly.

**Algorithm-specific hyperparameters.** The above baselines and FL methods adopt the same experimental setup as SFedPO, while the algorithm-specific hyperparameters are configured as follows:

- **Flash:** we tune the global learning rate $\eta_g \in \{0.001, 0.01, 0.1, 1.0\}$, and set it to 0.01.

- **FedEWC:** we tune the penalty coefficient $\lambda \in \{0.001, 0.01, 0.1, 1.0\}$, and set it to 0.1. $\lambda$ is a scalar that balances the contribution of the regularization loss relative to the cross-entropy loss in the total objective.

- **FLwF-2T:** we tune the $\alpha \in \{0.001, 0.01, 0.1, 0.3, 0.7\}, \beta \in \{0.001, 0.01, 0.1, 0.3, 0.7\}$, and set them to $\alpha = 0.3, \beta = 0.3$.

- **FedProx:** we tune the $\mu \in \{0.001, 0.01, 0.1, 1.0\}$, and set it to 0.1.

- **FedCurv:** we tune the $\lambda \in \{0.001, 0.01, 0.1, 1.0\}$, and set it to 0.01.

- **FedNTD:** we tune the $\beta \in \{0.01, 0.1, 1.0, 2.0\}$, and set it to 1.0.

## H.2 OTHER EXPERIMENTS

**Performance comparison.** Fig. 7 presents the convergence performance of SFedPO and all baselines on four datasets under two scenarios (here we use F-MNIST for Fashion-MNIST).

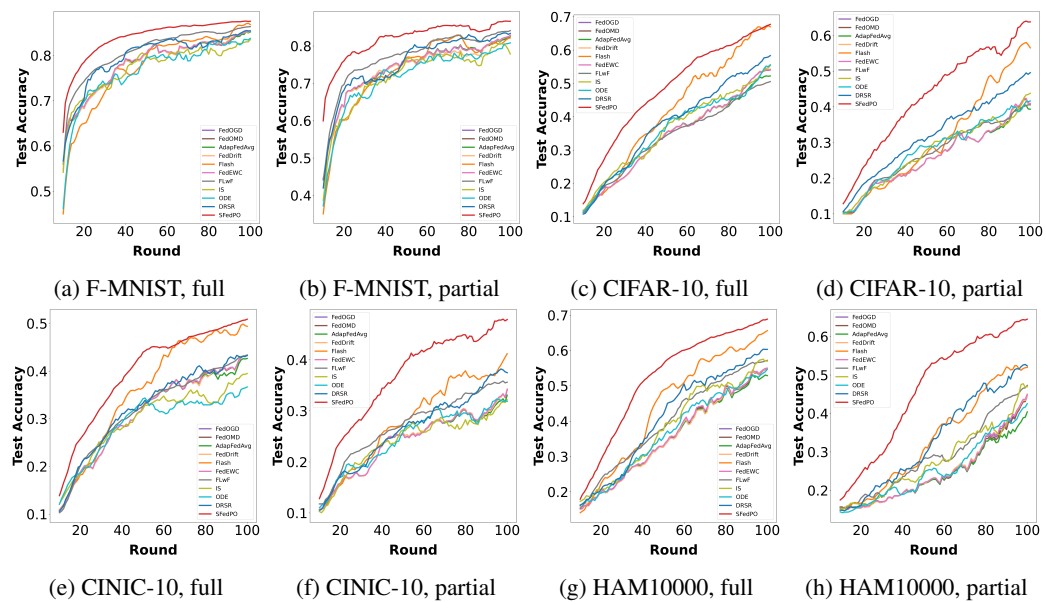

Figure 7: Performance comparison of SFedPO to other baselines in full and partial scenarios on several datasets.

**Effects of different configurations under other datasets.** We vary three core parameters, including time step ($T \in \{2, 5, 8, 10\}$), training round ($R \in \{50, 100, 150, 200\}$), and data capacity of clients ($D \in \{250, 500, 750, 1000\}$). Then, we respectively show the performance of our method and four baselines on Fashion-MNIST, CINIC-10, and HAM10000 in Fig. 8, Fig. 9, and Fig. 10. The experimental results demonstrate that our method outperforms all baseline approaches across different parameters under various datasets.

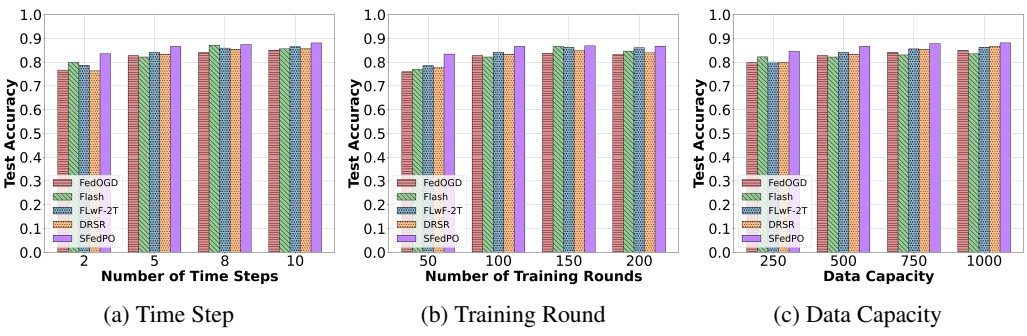

Figure 8: Fashion-MNIST.

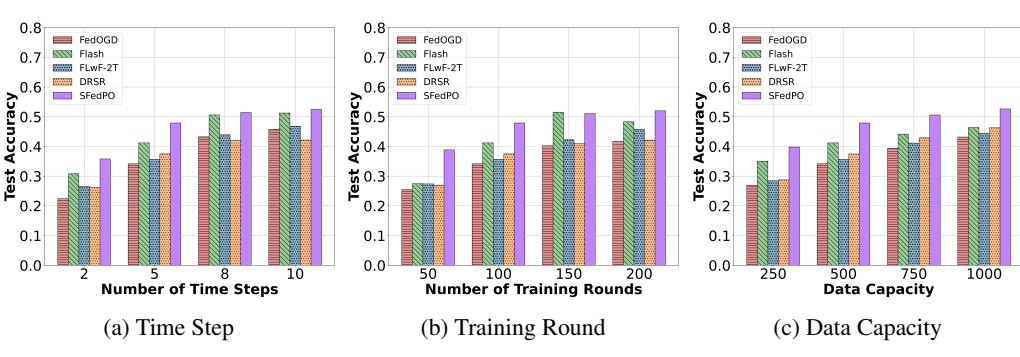

Figure 9: CINIC-10.

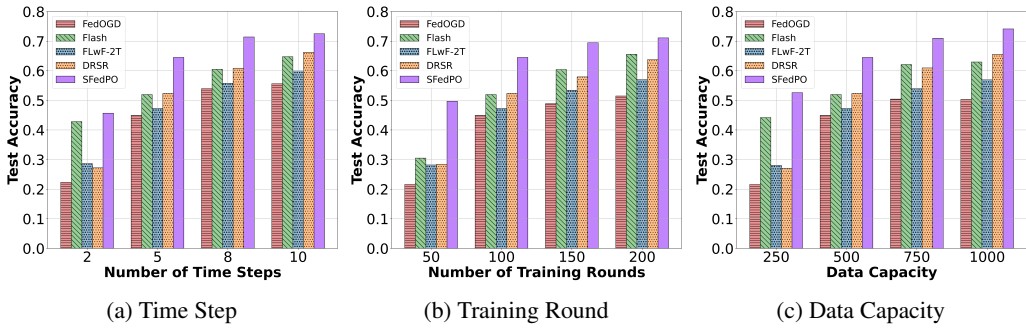

| (a) Time Step | (b) Training Round | (c) Data Capacity |

Figure 10: HAM10000.

**Effects of number of clients $N$.** In Table 5, we vary the number of clients $N \in \{10, 50, 100\}$ and compare the performance of SFedPO with several baselines in full and partial scenarios on CIFAR-10. The results show that SFedPO consistently outperforms the baselines across different $N$.

Table 5: Test accuracy (%, mean±std on 5 trials) comparison of our SFedPO framework to several baselines with different $N$ in full and partial scenarios on CIFAR-10.

| Method | CIFAR-10(full) | | | CIFAR-10(partial) | | |
|---|---|---|---|---|---|---|
| | $N = 10$ | $N = 50$ | $N = 100$ | $N = 10$ | $N = 50$ | $N = 100$ |
| FedOGD | 43.04±1.92 | 53.78±1.38 | 55.32±1.29 | 34.10±2.70 | 42.38±1.60 | 45.42±1.29 |
| Flash | 49.61±1.47 | 59.21±0.51 | 60.94±0.60 | 32.69±3.52 | 48.32±1.91 | 52.78±2.45 |
| FLwF-2T | 46.94±1.42 | 50.00±0.96 | 50.06±0.62 | 38.69±1.58 | 40.24±0.54 | 40.26±0.41 |
| DRSR | 51.45±1.99 | 58.74±0.43 | 59.98±0.45 | 43.43±1.22 | 50.70±1.06 | 53.44±0.78 |
| SFedPO | **63.51±0.90** | **67.94±0.23** | **68.62±0.26** | **55.66±1.24** | **65.57±0.34** | **66.74±0.43** |

**Effects of model architectures.** We evaluate SFedPO against several baselines in full and partial scenarios on CIFAR-10 under different model architectures, including CNN, ResNet-10 (He et al., 2016), ResNet-18 (He et al., 2016), and ResNet-34 (He et al., 2016). The results in Table 6 show that SFedPO consistently achieves the best performance, demonstrating the robustness of our method across model architectures.

Table 6: Test accuracy (%, mean±std on 5 trials) comparison of our SFedPO framework to several baselines with different model architectures in full and partial scenarios on CIFAR-10.

| Method | CIFAR-10(full) | | | | CIFAR-10(partial) | | | |
|---|---|---|---|---|---|---|---|---|
| | CNN | ResNet-10 | ResNet-18 | ResNet-34 | CNN | ResNet-10 | ResNet-18 | ResNet-34 |
| FedOGD | 65.13±0.62 | 45.46±1.05 | 47.16±1.53 | 50.05±1.32 | 56.95±1.21 | 37.18±1.45 | 40.11±2.22 | 41.67±1.07 |
| Flash | 67.29±0.56 | 49.99±1.80 | 60.14±3.41 | 55.26±2.00 | 60.59±3.15 | 42.25±1.49 | 51.74±2.035 | 47.07±1.55 |
| FLwF-2T | 66.51±0.69 | 48.60±0.20 | 49.04±1.04 | 51.42±1.53 | 59.02±1.50 | 40.60±2.33 | 42.81±1.19 | 44.74±1.28 |
| DRSR | 67.56±0.67 | 51.12±1.13 | 54.32±0.55 | 57.80±1.03 | 61.11±0.63 | 43.91±0.91 | 47.01±1.11 | 48.62±1.24 |
| SFedPO | **74.10±0.42** | **63.69±1.06** | **66.64±0.87** | **68.69±0.33** | **70.56±0.71** | **59.38±0.83** | **61.50±1.46** | **64.39±0.96** |

**Effects of different discrepancy metrics.** While our theoretical framework defines $d_m$ as the upper bound on the gradient variance under state $m$, estimating such variance is often impractical in streaming scenarios with limited and evolving local data. Inspired by (Ye et al., 2023), we adopt a practical surrogate by assuming that $d_m$ is proportional to the discrepancy between the class distribution and a uniform distribution. To investigate the impact of different discrepancy metrics, we evaluate four commonly used measures on the CIFAR-10 dataset: L1 & L2 distance, KL divergence, and Jensen–Shannon (JS) divergence (Fuglede & Topsoe, 2004). As shown in Table 7, all metrics yield comparable performance and consistently outperform baseline methods, highlighting the robustness of our framework to the choice of discrepancy measure.

Table 7: Accuracy (%) under different discrepancy metrics.

| Metric | L1 | L2 | JS | KL | Flash | DRSR |
|---|---|---|---|---|---|---|
| full | 67.56 | 67.58 | 67.56 | 68.18 | 67.00 | 58.36 |
| partial | 63.23 | 63.28 | 63.75 | 64.41 | 56.55 | 49.60 |

**Effects of Prediction Error.** To simulate prediction errors, we perturb each state probability $\hat{\pi}_n$ by a random noise uniformly drawn from $[-\epsilon, \epsilon]$, followed by renormalization to ensure $\sum_m \hat{\pi}_{n,m} = 1$. We vary $\epsilon$ from 0.00 to 0.10 to simulate increasing levels of oracle error. As shown in Figs. 11, SFedPO tends to maintain stable performance as the degree of perturbation varies on the CIFAR-10 dataset.

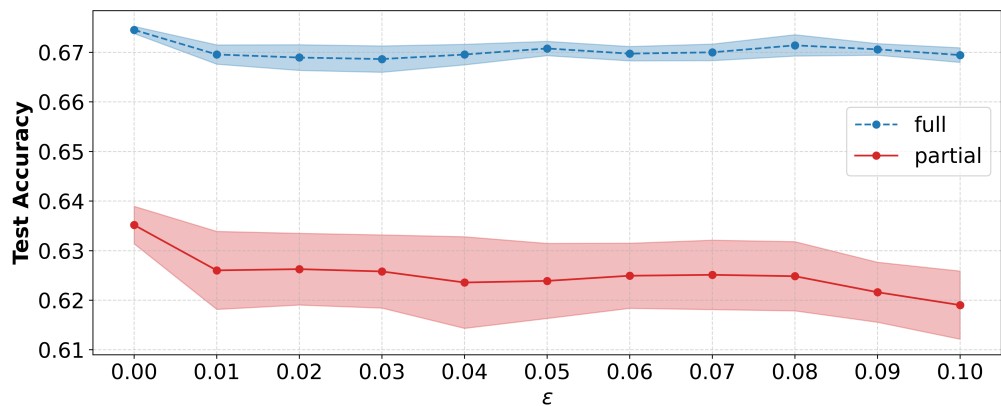

Figure 11: Performance under different degrees of perturbation.

**Effects of Client Availability.** Availability probability is directly incorporated into our aggregation mechanics as an influencing parameter, which is sampled from a normal distribution $\mathcal{N}(\mu, \sigma^2)$. We also include a modularity experiment that varies the client availability parameters. Across all settings, we observe that SFedPO consistently delivers clear performance gains, demonstrating that our module remains robust across different client availability levels. The detailed results are shown in Table 8.

Table 8: Accuracy (%) under different client availability.

| $\mathcal{N}(\mu, \sigma^2)$ | CIFAR-10(full) | | | | CIFAR-10(partial) | | | |
|---|---|---|---|---|---|---|---|---|
| | $\mu = 0.2$ | $\mu = 0.4$ | $\mu = 0.6$ | $\mu = 0.8$ | $\mu = 0.2$ | $\mu = 0.4$ | $\mu = 0.6$ | $\mu = 0.8$ |
| $\sigma = 0.01$ | 66.88(**+0.72**) | 68.92(**+2.38**) | 68.84(**+1.46**) | 69.13(**+2.05**) | 62.96(**+2.64**) | 66.56(**+5.82**) | 65.91(**+3.64**) | 66.83(**+3.89**) |
| $\sigma = 0.02$ | 67.29(**+1.15**) | 68.53(**+1.80**) | 68.65(**+1.48**) | 68.85(**+1.63**) | 63.08(**+3.18**) | 65.34(**+3.65**) | 66.58(**+4.16**) | 67.34(**+4.28**) |
| $\sigma = 0.03$ | 67.79(**+2.37**) | 68.27(**+1.58**) | 68.57(**+1.50**) | 69.16(**+1.93**) | 62.18(**+2.63**) | 64.43(**+2.48**) | 66.34(**+3.92**) | 66.69(**+4.06**) |
| $\sigma = 0.04$ | 67.33(**+1.57**) | 68.32(**+1.20**) | 68.88(**+2.11**) | 69.07(**+2.05**) | 61.88(**+2.07**) | 64.76(**+3.59**) | 66.23(**+3.30**) | 67.07(**+4.22**) |

