# OpenReview forum: "SFedPO: Streaming Federated Learning with a Prediction Oracle under Temporal Shifts"
_ICLR.cc/2026/Conference — Submitted to ICLR 2026_

### Official Review · Reviewer_CqT6 · 2025-10-16

**Soundness:** 2
**Presentation:** 2
**Contribution:** 2
**Rating:** 2
**Confidence:** 3

**Summary:**

This paper proposes SFedPO, a framework designed for FL under streaming data distributions by leveraging partial predictions about clients' data distribution shifts to guide both local data sampling and global aggregation. Based on a convergence upper bound, the authors develop two modules: Distribution-guided Data Sampling (DDS) and Shift-aware Aggregation Weights (SAW), which are claimed to jointly minimize the optimization error bound. Theoretical analysis provides convergence guarantees and robustness under prediction errors. Extensive experiments on multiple benchmarks show that SFedPO consistently improves test accuracy over existing FL baselines and can be plugged into various FL frameworks to further enhance performance.

**Strengths:**

1. This paper focuses on an important practical problem - federated learning with streaming and non-stationary data.

2. The theoretical convergence upper bound drives the development of DDS and SAW modules.

3. The experiment setup is reasonable, and the experiment results show the advantage of their proposed SFedPO.

**Weaknesses:**

W1. In lines 52-53 about the research question, the sudden transition to focusing on client sampling and server aggregation feels abrupt. From my personal perspective, the related background or explanation is not enough before coming up with the research question, which results in the research question not being convincing enough.

W2. Theorem 1 has a lack of readability because it lacks explicit interpretation and further discussion, such as convergence rate analysis, the meaning of each term, the impact of some key factors (e.g., $\pi$), difference between your theoretical results and the result with static data distribution, and so on.

W3. As for Equation (10), there are several issues:

(W3.1) In lines 268–270, the authors state that “In realistic streaming environments, it is neither desirable nor feasible for a client to completely discard previously stored data or to entirely ignore new samples from a given state.” This viewpoint is not completely convincing because there are realistic situations where it is necessary to discard newly received samples with extremely high noise. Moreover, this assumption implicitly supports the validity of Equation (10) because it ensures that $\alpha_n$ cannot be zero. However, in practice, $\alpha_n$ can indeed be zero, in which case Equation (10) becomes undefined.

(W3.2) The bounded gradient $G$ can be very large. As a very large $G$, $-a_1 d_m + b_1$ will be almost zero. Then, the score will not decrease with the state-specific heterogeneity bound $d_m$, which does not work as what they claimed.

**Typo:**
W4. Line 89: "a *date* evaluation metric" -> "a *data* evaluation metric"

**Questions:**

Q1. In the modularity experiment, does it use the same hyperparameter settings for baselines with and without SFedPO?

Q2. Can the authors provide more explanation and discussion about the Theorems 1 and 2? Moreover, what is your theoretical contribution compared to the existing works?

Q3: There are a lot of hyperparameters that need to be set in the experiment. Hence, I would like to know: how did the authors make sure that the ranges of $a_1$, $b_1$, $a_2$ and $b_2$ are reasonable? Moreover, when the heterogeneity score $s_n$ is large due to a large $G$, $a_2$ and $b_2$ may may not have a significant effect because they are dominated by $s_n$. Did the authors observe any related phenomena in your experiments?

See weaknesses above. I would adjusting my rating if the authors can address my concerns properly.

---

> ### Author Response · Authors · 2025-11-21
>
> > W1: In lines 52-53 about the research question, the sudden transition to focusing on client sampling and server aggregation feels abrupt.
>
> A1: We appreciate the reviewer’s feedback. Since client-side data sampling and server-side aggregation are standard and widely used components in FL algorithms, we will revise the sentence to:
>
> *This raises a natural question: Can we design a new FL framework that leverages partial predictions about distribution shifts to jointly optimize its two core components—client-side data sampling and server-side aggregation?*
>
>
> > W2 & Q2: More explanation and discussion about Theorems 1 and 2.
>
> A2: Thank you for the question. Due to space limits, we presented only the final forms of Theorems 1 and 2 in the main text. The appendix **already contains** the **convergence rate analysis** after the full proof of Theorem 1. In the revised version, we additionally clarify the meaning of key terms, the impact of some key factors, and their differences from the static-distribution setting.
>
> Theorem 1 provides an upper bound on the expected squared gradient norm under a fixed data sampling strategy $\boldsymbol{\alpha}$ and aggregation weights $\{p_n\}$.
>
> - $\alpha_n$ and $\alpha_{n,m}$  capture the sampling budget of client $n$ and its sampling ratio in state $m$, representing the impact of data sampling on convergence.
> - $\beta_n$ and $\beta_n'$  measure the heterogeneity of local updates and sampling behaviors, which increases the variance of gradient estimates.
> - $\gamma_n$ reflects the cumulative influence of retained old data across time steps.
> - The term involving $\pi_{n,m}$ quantifies the state distribution; for details on its impact on convergence, see Theorem 2 regarding the effect of the prediction oracle error.
>
> Unlike standard FL results, our bound explicitly incorporates **time-varying and partially predictable local data**, bridging classical static FL and online FL.
>
> Theorem 2 quantifies the impact of prediction errors from the oracle. If the strategies are constructed using estimated state distributions $\hat{\boldsymbol{\pi}}_n$, the degradation compared to using the true distributions ${\boldsymbol{\pi}}_n$  is bounded linearly by
>
> $$\sum_{n=1}^N \delta_n,$$
> where $\delta_n = \|\hat{\boldsymbol{\pi}}_n - \boldsymbol{\pi}_n\|_1$.
>
> Main contributions:
>
> - The analysis shows that our framework is **robust** to imperfect predictions.
> - It provides theoretical guarantees for **partially predictable streaming data**, a setting not addressed in prior FL analyses.
>
> > W3.1: the case $\alpha_n = 0$
>
> A3: We agree that, in practice, some newly received samples may have extremely high noise and should be discarded.
> This is **fully compatible** with our formulation: Eq. (10) allows **partial retention**, meaning a client may keep a fraction of old data and select only a subset of new samples—including selecting **zero** new samples in certain states.
>
> Regarding the concern that $\alpha_n$: in our framework, $\alpha_n$ denotes the **overall sampling budget** for newly arrived data. The case $\alpha_n = 0$ corresponds to a **static FL setting** where all new data are ignored, which is outside our streaming FL scope. In contrast, the **state-wise ratios** $\alpha_{n,m}$  can indeed be zero. We will explicitly clarify the assumption $\alpha_n > 0$ in the revised version.
>
> > W3.2 & Q3: the bounded gradient $G$, and the choice of $(a_1, b_1, a_2, b_2)$
>
> A4: We thank the reviewer for raising these questions.
>
> (1) On whether a large gradient bound $G$ makes the score insensitive to  $d_m$:
>
> A very large $G$ only arises when the number of states $M$ is extremely large. In such cases, both the state weight $w_m$ (e.g., $w_m = 1/M$) and the linear term $-a_1 d_m + b_1$ scale as $O(1/M)$. Since they shrink **proportionally**, the heterogeneity score does **not** become insensitive to $d_m$.
>
> (2) On the choice of $(a_1, b_1, a_2, b_2)$:
> Our sensitivity study shows that the method is **robust** **to a wide range** of these coefficients, and performance does not rely on narrow tuning. Moreover, $a_2$ and $b_2$ follow directly from standard **Lagrangian regularization**, as commonly used in FL, (e.g., FedDisco [Ye et al., 2023]).
>
> [1] Ye et al., Feddisco: Federated learning with discrepancy-aware collaboration. In International Conference on Machine Learning, 2023.
>
> > Q1. In the modularity experiment, does it use the same hyperparameter settings for baselines with and without SFedPO?
>
> A5: Yes. In the modularity experiment, all baselines use exactly the same hyperparameter settings before and after integrating SFedPO. We will make this point explicit in the revised version.

---

> > ### Comment · Reviewer_CqT6 · 2025-11-21
> >
> > Thank you very much for the detailed explanation and clarification. Most of my questions and concerns have been addressed, and I sincerely appreciate your effort. However, the response to W3.2 is not fully convincing for me.
> >
> > First, a large $G$ does not necessarily arise only when M is extremely large.
> > Second, I cannot agree with the statement "since they shrink proportionally, the heterogeneity score does not become insensitive to $d_m$." Even if they shrink proportionally, when G is very large, the term $-a_1 d_m + b_1$ becomes almost zero. For a term that is already close to zero, any proportional relationship inside becomes almost meaningless because the influence of $d_m$ effectively almost vanishes regardless of proportional scaling.
> >
> > Overall, based on the authors' first-round rebuttal, I would like to raise my rating to 4. I am also open to further timely discussion if the authors would like to elaborate more on this point.

---

> ### Author Response · Authors · 2025-11-25
>
> We sincerely appreciate the positive adjustment in our rating. To address the reviewer’s remaining concerns, we provide additional clarification from both theoretical and experimental perspectives. As stated in our initial rebuttal, by “shrink proportionally” we mean that both $a_1$ and $b_1$ scale as $\frac{1}{G}$. Specifically,
> $$a_1=\frac{1-\gamma_n}{4G\gamma_n}, b_1=\frac{\lambda_1}{4G\gamma_n}.$$
>
> First, without considering gradient explosion, a finite $G$ must exist, which is a common and widely adopted assumption. Consequently, both $a_1$ and $b_1$ remain strictly positive.
>
> Secondly, in our strategy for computing $\alpha_{n,m}$, we are not concerned with the specific value of $-a_1d_m+b_1$, but rather with the ratio between scores, i.e.,
>
> $$\frac{score_n(m)}{score_n(m^{\prime})}=\frac{w_m-a_1d_m+b_1}{w_{m^\prime}-a_1d_{m^\prime}+b_1}\cdot \frac{1+\frac{1-\alpha_n}{\alpha_n}\pi_{n,m^\prime}}{1+\frac{1-\alpha_n}{\alpha_n}\pi_{n,m}}.$$
>
> We emphasize that this ratio remains valid even when $a_1$ and $b_1$ become very small. This is because
>
> $$\frac{w_m-a_1d_m+b_1}{w_{m^\prime}-a_1d_{m^\prime}+b_1}\approx \frac{-a_1d_m+b_1}{-a_1d_{m^\prime}+b_1}=\frac{-a_1^\prime d_m+b_1^\prime}{-a_1^\prime d_{m^\prime}+b_1^\prime},$$
>
> where $a_1^\prime$ and $b_1^\prime$ denote the decayed versions of $a_1$ and $b_1$. Since $w_m=\frac{1}{60}=0.016$ is relatively small and effectively dominated by $b_1$, the ratio remains meaningful even when both coefficients shrink substantially.
>
> To further verify this, we extend the robustness range used in the paper ($a_1 \in [0.05, 0.25]$, $b_1 \in [0.15, 0.35]$) by **shrinking both $a_1$ and $b_1$ by an additional factor of 100**, and repeat the same experiment setup as in Fig. 3 (“Effectiveness and robustness of modules”). We observe that the DDS sampling module continues to deliver consistent improvements. The table below reports the accuracy with and without DDS, along with the gains. This demonstrates that the effectiveness of the module does not depend on the magnitudes of $a_1$ and $b_1$.
>
> We thank you again for the increased rating and hope this clarification fully resolves the remaining concerns.
>
> |  | a1=0.0005 | a1=0.001 | a1=0.002 | a1=0.003 | a1=0.004 | a1=0.005 |
> | --- | --- | --- | --- | --- | --- | --- |
> | a1:b1=1:1 | 63.16(+**2.00**) | 62.07(**+0.91**) | 62.44(**+1.27**) | 63.69(**+2.52**) | 62.63(**+1.46**) | 63.88(**+2.72**) |
> | a1:b1=1:2 | 61.66(**+0.50**) | 61.81(**+0.65**) | 62.46(**+1.29**) | 64.02(**+2.85**) | 62.87(**+1.70**) | 63.66(**+2.49**) |
> | a1:b1=1:3 | 61.28(**+0.11**) | 62.08(**+0.92**) | 61.78(**+0.61**) | 61.93(**+0.76**) | 63.38(**+2.22**) | 61.91(**+0.75**) |
> | a1:b1=1:4 | 61.67(**+0.51**) | 62.11(**+0.95**) | 63.06(**+1.90**) | 62.68(**+1.51**) | 61.75(**+0.59**) | 62.50(**+1.33**) |
> | a1:b1=1:5 | 61.44(+**0.28**) | 61.88(+**0.72**) | 61.95(**+0.79**) | 62.46(**+1.29**) | 61.72(**+0.55**) | 62.55(**+1.39**) |
> | a1:b1=1:6 | 61.42(+**0.26**) | 61.70(+**0.54**) | 63.13(**+1.97**) | 62.23(+**1.07**) | 62.79(**+1.62**) | 62.68(**+1.51**) |
> | a1:b1=1:7 | 61.73(+**0.57**) | 62.52(+**1.35**) | 61.98(**+0.82**) | 61.43(+**0.26**) | 62.35(**+1.18**) | 62.14(**+0.97**) |

---

### Official Review · Reviewer_v5P3 · 2025-10-31

**Soundness:** 3
**Presentation:** 3
**Contribution:** 3
**Rating:** 6
**Confidence:** 3

**Summary:**

The paper studies the problem of federated learning when clients receive stream of data. Due to having dynamic environment, the data distribution among clients may vary over time. The paper proposed adjusting data sampling for model training and obtained the convergence rate. Then to optimize the convergence, the paper proposes a client data sampling distribution and a server aggregation strategy. The proposed solution is based on the presence of reliable oracle that can predict states of clients. Experimental results show that the proposed algorithm achieves better results than other baselines.

**Strengths:**

- The problem of federated learning facing with stream of data is an interesting problem. The paper focuses on the cases where there is a possibility to predict the next state of clients.
- The paper provides convergence analysis for the proposed algorithm. Furthermore, the proposed data sampling and aggregation strategy technically sound which optimizes the convergence rate.
- The paper provides comprehensive the experimental results.

**Weaknesses:**

- The performance of the proposed algorithm depends on the quality of the predictive oracle model. However, an accurate oracle model may not always be available.
- The paper could benefit from including “Federated Learning for Data Streams” (Marfoq et al., 2023) as one of the baselines to strengthen the experimental study.
- A discussion on how the use of the oracle model can improve the convergence rate could also be added to enhance the paper.

**Questions:**

Based on my understanding, the paper focuses on scenarios where clients operate in dynamically changing environments, and the model training is adjusted accordingly to adapt to these changes. However, the paper does not appear to consider client heterogeneity. Could you elaborate on this aspect?

---

> ### Author Response · Authors · 2025-11-21
>
> > W1: The performance of the proposed algorithm depends on the quality of the predictive oracle model. However, an accurate oracle model may not always be available.
>
> A1: We appreciate the reviewer’s comment. The case of imperfect oracles has been fully considered in our analysis. **Theorem 2** provides a robustness guarantee showing that convergence degrades only linearly with prediction error. Our experiments (**Effects of Prediction Error**) further verify this behavior, demonstrating that the framework remains stable even when the oracle is inaccurate.
>
> > W2: The paper could benefit from including "Federated Learning for Data Streams" (Marfoq et al., 2023) as one of the baselines to strengthen the experimental study.
>
> A2: We are aware of ''Federated Learning for Data Streams" (Marfoq et al., 2023) and have discussed it in our related work. However, this method assumes a stationary data distribution, while our work targets distribution-shifting streaming FL, which is a fundamentally different setting. Therefore, we chose baselines from OFL, concept drift, FCL, and data selection methods, as their assumptions are aligned with our problem setup.
>
> > W3: A discussion on how the use of the oracle model can improve the convergence rate could also be added to enhance the paper.
>
> A3: Theorem 2 implies that a more accurate oracle directly improves the convergence by reducing the prediction-error term. The “best” oracle corresponds to the true state distribution and achieves the rate in Theorem 1. In the revised version, we will also briefly discuss practical ways to improve oracle accuracy.
>
> > Q1: Based on my understanding, the paper focuses on scenarios where clients operate in dynamically changing environments, and the model training is adjusted accordingly to adapt to these changes. However, the paper does not appear to consider client heterogeneity. Could you elaborate on this aspect?
>
> A4: Client heterogeneity is indeed incorporated in our framework. In our setting, clients follow **different latent state transition patterns**, which naturally create **heterogeneous local data distributions** across clients. This heterogeneity is explicitly captured through the client-specific score $s_n$, which accounts for:
>
> 1. **State distribution heterogeneity:** Clients observe different state dynamics, leading to diverse temporal data-generation patterns.
> 2. **Sampling and availability heterogeneity:** Clients have distinct sampling policies and availability probabilities, further amplifying differences in their effective data streams.
> 3. **Resulting statistical heterogeneity:** These factors jointly determine each client’s effective data distribution.
>
> Our aggregation rule directly uses these heterogeneous scores $s_n$ to assign different aggregation weights. Thus, client heterogeneity is not only considered but **actively drives** how global updates are formed in our method.

---

### Official Review · Reviewer_Qsz2 · 2025-11-04

**Soundness:** 3
**Presentation:** 3
**Contribution:** 3
**Rating:** 4
**Confidence:** 2

**Summary:**

This paper investigates federated learning under dynamic distribution shift scenarios. SFedPO avoids two extremes: traditional FL (assuming static distribution) and online FL (not using previous information at all)j by employing sampling strategy and client weighting mechanism. Both mechanisms are theoretically supported and experimental results demonstrate the superiority of SFedPO.

**Strengths:**

1. Research topic is timely and meaningful: federated learning using streaming data (dynamic data distribution).
2. Both ideas (sampling and client weighting) are theoretically supported.
3. Experimentally demonstrated performance improvement

**Weaknesses:**

1. Model architectures are too old: AlexNet and LeNet-5, meaning that updating only architectures can exceed the gain of the proposal. Utilizing at least ResNet is recommended.
2. Datasets are small and less challenging. Evaluating on CIFAR-100 is more common for federated learning. Dynamic environments can incur class changes in CIFAR-100, which makes challenges.

**Questions:**

Please address the weaknesses above.

---

> ### Author Response · Authors · 2025-11-21
>
> > W1: Model architectures are too old: AlexNet and LeNet-5, meaning that updating only architectures can exceed the gain of the proposal. Utilizing at least ResNet is recommended.
>
> A1: We agree with the reviewer. We would like to clarify that the original submission **already** includes evaluations of SFedPO under multiple architectures—ResNet-10, ResNet-18, and ResNet-34—as reported in the appendix (“**Effects of model architectures**”).
>
> > W2: Datasets are small and less challenging. Evaluating on CIFAR-100 is more common for federated learning. Dynamic environments can incur class changes in CIFAR-100, which makes challenges.
>
> A2: We appreciate the reviewer's suggestions, and we have accordingly added experiments on CIFAR-100. Since CIFAR-100 is substantially more challenging—with more classes and greater state-transition complexity in dynamic environments—we increase the average client data storage to 2500 samples to ensure sufficient learning capacity under this harder setting. The results are presented below.
>
> | **CIFAR-100(ResNet-18)** | **full** | **partial** |
> | --- | --- | --- |
> | FedOGD | 25.67±0.80 | 23.62±0.51 |
> | FedOMD | 25.31±0.99 | 23.45±0.68 |
> | AdapFedAvg | 25.76±0.89 | 23.15±0.42 |
> | Flash | 28.50±0.66 | 25.69±0.88 |
> | FedEWC | 25.60±1.01 | 23.46±0.25 |
> | FLwF-2T | 26.06±0.59 | 23.95±0.40 |
> | IS | 25.48±0.99 | 23.23±0.44 |
> | ODE | 25.24±0.49 | 23.90±0.69 |
> | DRSR | 27.41±0.42 | 25.55±0.50 |
> | **SFedPO** | **29.12±0.24** | **25.97±0.89** |
>
> Results show that our method significantly outperforms all baselines. We further evaluate the modularity experiment under the same setting as well:
>
> | **CIFAR-100(ResNet-18)** | **full** | **partial** |
> | --- | --- | --- |
> | FedAvg | 30.66(**+1.68**) | 27.83(**+0.70**) |
> | FedProx | 28.78(**+0.44**) | 27.13(**+0.21**) |
> | FedCurv | 29.88(**+0.17**) | 27.67(**+0.15**) |
> | FedNTD | 30.09(**+0.84**) | 29.31(**+0.74**) |
> | FedEXP | 20.82(**+1.58**) | 23.98(**+4.77**) |
>
> Results show that SFedPO exhibits strong modularity and can be easily integrated into a wide range of classical FL methods as a plug-and-play module to cope with streaming data scenarios.

---

### Official Review · Reviewer_uHgr · 2025-11-06

**Soundness:** 2
**Presentation:** 3
**Contribution:** 2
**Rating:** 6
**Confidence:** 3

**Summary:**

The paper introduces SFedPO, a streaming federated learning (FL) framework designed to bridge the gap between conventional static FL and fully adversarial online FL settings. The authors assume that temporally evolving client data can be modeled as transitions among a finite set of latent states, each corresponding to a stationary distribution of data. By incorporating a prediction oracle that estimates the transition probabilities among these states, SFedPO dynamically adjusts local data sampling through a distribution-guided strategy (DDS) and adapts global aggregation via shift-aware weights (SAW). The authors provide convergence guarantees under this setup, demonstrate robustness to oracle prediction errors, and show that SFedPO consistently improves accuracy over baseline FL methods in simulated streaming settings.

**Strengths:**

The paper is logically structured, demonstrating a clear flow from problem formulation to theoretical analysis and practical implementation. The authors present a comprehensive convergence analysis and include robustness guarantees with respect to prediction errors. The framework is modular, meaning it can be integrated with multiple FL algorithms, and the experimental results demonstrate measurable improvements in accuracy across several baseline methods. The work also addresses an unfilled niche in FL literature by navigating between static and adversarial formulations with partial future knowledge.

**Weaknesses:**

The primary weakness lies in the practicality and realism of the assumptions. The use of a finite latent space with a known transition model and access to a prediction oracle may not reflect real-world data characteristics, and the experiments do not validate this setup beyond simulations. The partial-access experiment design may unfairly benefit SFedPO by clustering latent states in ways that other methods are not designed to exploit. Moreover, the feasibility of estimating parameters, such as the number of states or heterogeneity bounds, remains unclear, and no computational overhead analysis is presented for the proposed sampling and weighting schemes.

**Questions:**

What is the computational overhead of computing the sampling ratios $\alpha_{n,m}$ and aggregation weights pₙ during federated rounds, and how does this compare to the cost of local model training and communication?

Are there settings, either in terms of client availability, data dynamics, or oracle error, where SFedPO may underperform relative to classical FL methods such as FedAvg?

How should practitioners estimate or determine the number of latent states M if no prior structure is available in a real dataset?

In Equation (1), the authors propose updating the local distribution through a convex combination. Are there alternative techniques (e.g., kernel-based blending or Bayesian updating) that could better capture uncertainty or non-convex transitions across states?

---

> ### Author Response · Authors · 2025-11-21
>
> We appreciate the reviewer's insightful comments.
> > Q1 & W3: computational overhead.
>
> A1: The sampling strategy in Algorithm 2 has complexity **$O(M^2)$** per client, and computing aggregation weights on the server costs **$O(N)$**, where $M$ is the number of latent states and $N$ is the number of clients. Regarding the overall overhead:
> - **Offline setting.** When the oracle is fixed, $\alpha_{n,m}$ and $\{p_n\}$ are computed **once before training**. The added cost during FL rounds is therefore **negligible** compared to local computation and communication.
> - **Online setting.** When the oracle is updated, each client updates its estimate after uploading its model. The sampling ratios used in the current round come from the previous estimate. This update is lightweight and can be **overlapped with communication**, imposing almost no extra wall-clock time.
>
> In our experiments (offline case), the additional cost is **essentially zero** relative to standard FL training.
> > Q2: data dynamics, client availability, oracle error
>
> A2: SFedPO is specifically designed for **streaming, non-stationary** FL and—both theoretically and empirically—outperforms classical methods under **data dynamics** (see Tables 1–2).
>
> Our theoretical analysis (Theorem 2) and the experiments in *Effects of Prediction Error* (Section 6 and Appendix H.2) show that SFedPO is robust to oracle inaccuracy, with performance degrading smoothly as prediction noise increases.
>
> Availability probability is directly incorporated into our aggregation mechanics as an influencing parameter, which is sampled from a normal distribution $N(\mu, {\sigma}^2)$. We further include a modularity experiment by varying the client availability parameters. Across all settings, SFedPO consistently brings clear performance gains. The details are shown below:
>
> | CIFAR-10(full) | μ=0.2 | μ=0.4 | μ=0.6 | μ=0.8 |
> | --- | --- | --- | --- | --- |
> | σ=0.01 | 66.88(**+0.72**) | 68.92(**+2.38**) | 68.84(**+1.46**) | 69.13(**+2.05**) |
> | σ=0.02 | 67.29(**+1.15**) | 68.53(**+1.80**) | 68.65(**+1.48**) | 68.85(**+1.63**) |
> | σ=0.03 | 67.79(**+2.37**) | 68.27(**+1.58**) | 68.57(**+1.50**) | 69.16(**+1.93**) |
> | σ=0.04 | 67.33(**+1.57**) | 68.32(**+1.20**) | 68.88(**+2.11**) | 69.07(**+2.05**) |
>
> | CIFAR-10(partial) | μ=0.2 | μ=0.4 | μ=0.6 | μ=0.8 |
> | --- | --- | --- | --- | --- |
> | σ=0.01 | 62.96(**+2.64**) | 66.56(**+5.82**) | 65.91(**+3.64**) | 66.83(**+3.89**) |
> | σ=0.02 | 63.08(**+3.18**) | 65.34(**+3.65**) | 66.58(**+4.16**) | 67.34(**+4.28**) |
> | σ=0.03 | 62.18(**+2.63**) | 64.43(**+2.48**) | 66.34(**+3.92**) | 66.69(**+4.06**) |
> | σ=0.04 | 61.88(**+2.07**) | 64.76(**+3.59**) | 66.23(**+3.30**) | 67.07(**+4.22**) |
> > Q3 & W1 & W3: if no prior structure is available
>
> A3: For datasets with no clear prior structure, practitioners can adopt **adaptive latent-state modeling**. A lightweight solution is to monitor whether new observations are poorly explained by all existing states; when their distribution significantly deviates, **a new state is created** using concept-drift detection or distance-based criteria. This requires only minimal modification to our framework and enables the latent state space to expand dynamically as needed. Moreover, the prior of each state in our Bayesian oracle can be updated according to its empirical frequency. We will add the full discussion to the appendix.
> > Q4: alternative techniques
>
> A4: In our framework, the convex update in Eq. (1) is directly aligned with the **sampling mechanism**: each client retains a portion of historical data and replaces the rest with newly arrived samples, naturally resulting in a linear mixture of the corresponding state distributions.
>
> Alternative formulations are possible. For example, we can introduce kernel-based blending, where each client's local distribution is represented as a kernel mean embedding (KME) in a reproducing kernel Hilbert space (RKHS). Specifically, for client $n$ at time $t$ in round $r$, the local distribution can be represented as $\mu_{n,t}^{(r)} = \mathbb{E}_{x}[\phi(x)]$,
>
> where $\phi(\cdot)$ denotes the feature mapping corresponding to a positive-definite kernel $k(\cdot, \cdot)$. The local KME update can then be written as a convex combination in the RKHS:
>
> $\mu_{n,t}^{(r)} = (1-\alpha_{n,m_{n,t}^{(r)}}) \cdot \mu_{n,t-1}^{(r)} + \alpha_{n,m_{n,t}^{(r)}} \cdot \mu_{m_{n,t}^{(r)}},$
>
> This formulation preserves the original sampling strategy while mapping distributions into a high-dimensional feature space, enabling the representation to capture non-linear characteristics. We consider this a promising extension and will investigate it in future work.
> > W2: clustering states
>
> A5: Our latent-state clustering is designed to reflect different degrees of heterogeneity. In realistic systems, heterogeneity arises not only from distributional differences but also from the degree of heterogeneity across states. The clustering mechanism is solely used to model these varying heterogeneity levels.

---

### Meta-Review · Area_Chair_j51A · 2026-01-04

**Summary:**

The manuscript considers FL on time-varying local data distributions, in which it proposes SFedPO. This streaming FL framework incorporates a prediction oracle to capture the temporal evolution of client-side data distributions.

The reviewers question the novelty, practicality & realism of the assumptions, computational overhead, and less challenging experimental datasets. The AC also noticed that the current manuscript did not carefully examine the literature of FL on time-varying local data, and did not consider more recent baselines.

After reviewing the rebuttal, the AC believes that the current form of the manuscript falls below the bar for ICLR acceptance.

**Reviewer Concerns:**

see summary above.

**Reviewer Scores:**

Reviewer uHgr may keep its score (6); Reviewer Qsz2 may keep its score (4); Reviewer v5P3 may keep its score (6); Reviewer CqT6 mentioned raising its score to 4.

---

### Decision · Program_Chairs · 2026-01-26

Reject